# GENEB: Why Genomic Models Are Hard to Compare

**Daria Ledneva** [1]  **Mikhail Nuridinov** [2]  **Denis Kuznetsov** [1]

## Abstract

Progress in genomic foundation models is difficult to assess due to fragmented benchmarks, incompatible evaluation protocols, and task-specific reporting. As a result, claims of superiority or generality across models are often not directly comparable. We introduce GENEB, a large-scale diagnostic benchmark that evaluates frozen representations from 40 genomic foundation models across 100 tasks spanning 13 functional categories under a unified probing-based protocol, including few-shot regimes. GENEB enables controlled comparison across model scale, architecture, tokenization, and pretraining data while explicitly exposing task-level trade-offs. Our analysis shows that aggregate leaderboards are unstable: model rankings vary sharply across task categories, scale provides only modest and inconsistent gains, and architectural and pretraining alignment frequently outweigh parameter count. These results highlight limitations of current evaluation practices and position GENEB as a reference framework for principled comparison and category-aware model selection in genomic machine learning.

## 1. Introduction

The genomic machine learning landscape has expanded rapidly over the past decade, producing a large and heterogeneous ecosystem of models, architectures, and training paradigms. This expansion has not been accompanied by commensurate methodological infrastructure for comparison. Figure 1 illustrates the present state of the field: models are evaluated on disjoint benchmarks, compared under incompatible protocols, and frequently reported as state-of-the-art within narrowly defined settings, leaving unclear how different models relate to one another or whether reported improvements reflect genuine progress.

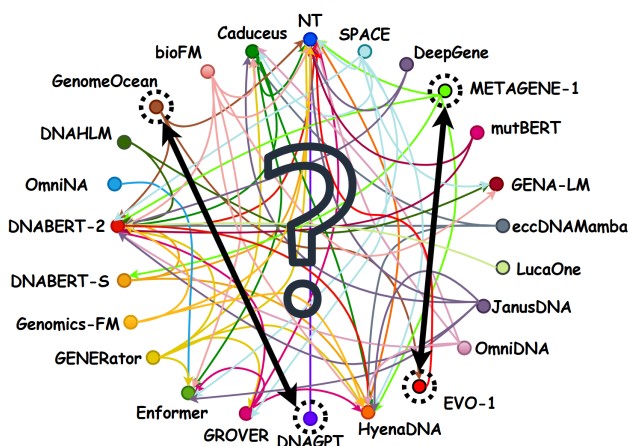

*Figure 1.* **Fragmented comparison landscape of genomic foundation models.** Each node represents a published model; directed edges denote models explicitly used as baselines or comparators in the corresponding paper. The sparse, disconnected graph reflects the absence of unified cross-model evaluation in genomic machine learning.

This fragmentation makes even basic questions difficult to answer. Principled comparison between widely discussed models such as DNA-GPT (Zhang et al., 2023), GENOMEOCEAN (Zhou et al., 2025), and EVO (Nguyen et al., 2024) is currently not feasible: each is evaluated on different task sets, preprocessing pipelines, and evaluation protocols. The same model is sometimes characterised as a major breakthrough in one context and as underperforming in another, reflecting not contradictory evidence but the absence of a common evaluation framework.

This problem is compounded by the rapid growth in scale and visibility of genomic foundation models. As models become larger and more expressive, claims of superiority and generality have grown correspondingly bolder, yet the methodological basis for adjudicating such claims has not kept pace. The result is a widening gap between what is asserted about model capability and what can be reliably established through reproducible cross-model evaluation.

We introduce **GENEB**, a large-scale benchmark evaluating 40 genomic foundation models on 100 tasks spanning 13 functional categories under a unified probing protocol. GENEB is designed to enable controlled, systematic com-

[1]Moscow Independent Research Institute of Artificial Intelligence, Moscow, Russia [2] Moscow State Institute of Steel and Alloys, Moscow, Russia. Correspondence to: Daria Ledneva <a.ledn2026@gmail.com>.

*Proceedings of the 43ʳᵈ International Conference on Machine Learning*, Seoul, South Korea. PMLR 306, 2026. Copyright 2026 by the author(s).

parison and to expose performance trade-offs obscured by fragmented evaluation practices; in spirit, it plays a role analogous to MTEB (Muennighoff et al., 2023) in natural language processing, providing a unified evaluation framework rather than a single-task leaderboard. By making results directly comparable across models and tasks, GENEB establishes a shared reference point for assessing progress in genomic machine learning.

**Conflict of Interest Disclosure.** The authors declare no financial conflicts of interest. None of the 40 evaluated models was developed by the authors or their funders.

## 2. Related Work

The rapid growth of genomic foundation models has produced a heterogeneous landscape spanning diverse architectures, tokenization schemes, and pretraining strategies.

**Architectures.** Early genomic models predominantly adopted Transformer encoders trained with masked language modeling (Zhou et al., 2024a; Dalla-Torre et al., 2023; Fishman et al., 2023; Sanabria et al., 2024). More recent work explored decoder-only and generative architectures for unified sequence modeling and long-context processing (Zhang et al., 2023; Nguyen et al., 2024; Wu et al., 2025; Zhou et al., 2025; Li et al., 2025b). To reduce attention complexity, alternative designs based on long convolutions and state-space models have been proposed, alongside hybrid architectures combining multiple paradigms (Nguyen et al., 2023; Schiff et al., 2024; Liu et al., 2025; Duan et al., 2025; Vishniakov et al., 2025b).

**Tokenization and Pretraining.** Tokenization strategies range from single-nucleotide and $k$-mer representations to learned BPE vocabularies, each offering different trade-offs between resolution and efficiency (Zhou et al., 2024a; 2025). Pretraining data similarly varies from human-only and species-specific corpora to broad multi-species and domain-focused datasets, with prior studies suggesting potential benefits of both diversity and specialization depending on the task (Dalla-Torre et al., 2023; Wu et al., 2025; Avsec et al., 2021; Zhai et al., 2025).

**Benchmarks.** Several benchmarks evaluate genomic foundation models, including Nucleotide Transformer tasks (Dalla-Torre et al., 2023), GUE/GUE+ (Zhou et al., 2024a), Genomic Benchmarks (Gresova et al., 2022), BEND (Marin et al., 2024), and DNALongBench (Cheng et al., 2025). While these resources cover important regulatory, epigenetic, and cross-species tasks, they differ in task design and evaluation protocols and typically assess only a limited subset of models, making cross-paper comparison difficult.

**Comparative Benchmarking Studies.** Recent studies have explored broader comparisons of genomic foundation models, but usually evaluate a small number of representative architectures. For example, Wang et al. (2025b) focus on approximately ten model families and predominantly human-centric tasks. Platform-based efforts such as OmniGenBench (Wang et al., 2025a) provide dynamic leaderboards, but currently include a limited and evolving set of baselines, leaving many recent DNA-specific models unevaluated.

**Positioning of GENEB.** GENEB addresses these gaps by providing a large-scale, controlled benchmark covering 40 genomic foundation models evaluated on 100 DNA classification tasks across 13 functional categories. By evaluating all models on the full task suite under a unified probing-based protocol, GENEB enables matched comparisons across architecture, tokenization, and pretraining data and yields a complete performance matrix that exposes task-dependent trade-offs. We plan to release GENEB as a public benchmark with evaluations hosted on Hugging Face, serving as a community reference analogous to MTEB in NLP (Muennighoff et al., 2023).

**Extended Related Work.** A detailed discussion of prior benchmarks, comparative studies, and architectural trends is provided in Appendix A.

## 3. Methodology

GENEB evaluates genomic foundation models (see Appendix C, Table 4) using an embedding-based *probing* protocol: frozen sequence representations are assessed with lightweight classifiers, isolating representation quality and enabling controlled comparison across architectures and training regimes. The benchmark covers diverse genomic prediction tasks spanning multiple functional categories; full task definitions are provided in Appendix B.

**Probing setup.** For each task, frozen embeddings are used as features for logistic regression (`max_iter=1000`) and evaluated in 1-shot, 10-shot, and full-data regimes (Figure 5). Results are averaged over five fixed random seeds $\{13, 17, 42, 123, 997\}$. The stability of model rankings under non-linear probing is verified empirically in Appendix E.1, and sensitivity of few-shot conclusions to the choice of regularization strength is analyzed in Appendix E.2.

**Metric and data.** We report Matthews Correlation Coefficient (MCC), which is robust to class imbalance and standard in genomic evaluation. Tasks exceeding $10^5$ sequences are subsampled. An empirical analysis using GENOMEOCEAN embeddings shows that MCC stabilizes beyond this size, motivating $10^5$ as a practical upper bound.

# 4. Aggregate Performance Analysis Across 100 Genomic Tasks

We present a systematic analysis of 40 DNA foundation models evaluated on 100 genomic prediction tasks spanning 13 functional categories. Our goal is to characterize how model scale (Figure 2), architecture, tokenization, and pretraining data interact under a unified evaluation protocol, and to extract practically relevant patterns for model selection. Unless stated otherwise, all statistics refer to MCC aggregated within the GENEB benchmark.

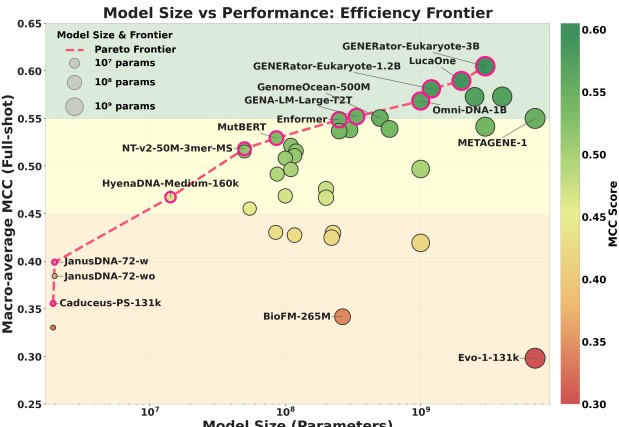

*Figure 2.* **Pareto frontier of model efficiency: macro-MCC vs. parameter count.** Each point represents one of the 40 genomic foundation models, with parameter count on a logarithmic x-axis and full-shot macro-average MCC on the y-axis. Marker size and color both encode macro-MCC. The dashed line marks the Pareto frontier of best performance–size trade-offs. Spearman correlation between $\log(\text{params})$ and macro-MCC is $\rho = 0.565$ ($p < 0.001$); excluding the prokaryotic-only outlier EVO-1-131K raises this to $\rho = 0.685$ ($p < 0.001$). While scale is a substantial predictor of aggregate performance, several large models fall below the frontier, indicating that architecture and pretraining choices can offset substantial scale differences (see Section 4, Table 1).

**The Scale–Performance Disconnect.** Model size exhibits a statistically significant and substantial association with aggregate performance ($\rho = 0.565$, $p < 0.001$; Figure 2), which strengthens further to $\rho = 0.685$ ($p < 0.001$) once the prokaryotic-only outlier EVO-1-131K is excluded (see Domain Mismatch paragraph below). Even with this strong aggregate trend, however, model selection is not reducible to parameter count: among the 36 in-domain models (excluding prokaryotic-only, microbial-only, and plant-specific pretraining), we identify 31 instances in which a model at least $5\times$ smaller outperforms a larger counterpart in aggregate MCC, with this count being identical under micro- and macro-averaging. A representative example is MUT-BERT (86M, Transformer-encoder), which exceeds ECCD-NAMAMBA (1B, Mamba-SSM) by $+0.110$ macro-MCC despite an 11.6-fold size difference, illustrating that non-scale design choices can offset substantial scale gaps in

GENEB. Category-level scaling correlations are reported in Table 1. We additionally verify that these aggregate statistics are robust to the choice of averaging scheme: macro-averaged MCC (weighting all 13 categories equally) yields rankings that correlate with the micro-averaged rankings reported here at $\rho = 0.988$ (Appendix E.4).

*Table 1.* **Per-category scaling correlations.** Spearman rank correlation $\rho$ between $\log_{10}(\text{parameter count})$ and macro-MCC within each functional category ($n = 40$ models). $\rho$ near $+1$ indicates that larger models systematically outperform smaller ones; $\rho$ near 0 indicates no monotonic relationship between size and performance. The $p$-value tests whether the observed $\rho$ differs from zero; bolded values are significant at $p < 0.05$. Rows sorted by $\rho$ descending. Scaling is significant in 11 of 13 categories, with $\rho$ ranging from 0.345 (DNA methylation) to 0.579 (histone modifications).

| Category | $\rho$ | $p$ |
|---|---|---|
| Histone modifications | **0.579** | $< 0.001$ |
| lncRNA | **0.568** | $< 0.001$ |
| Splice sites | **0.537** | $< 0.001$ |
| Enhancers | **0.490** | 0.001 |
| Promoters | **0.487** | 0.001 |
| Coding/non-coding | **0.482** | 0.002 |
| Mouse enhancers | **0.474** | 0.002 |
| Virus/phage | **0.434** | 0.005 |
| Regulatory | **0.377** | 0.017 |
| TF binding | **0.356** | 0.024 |
| DNA methylation | **0.345** | 0.030 |
| Species classification | 0.304 | 0.057 |
| Chromatin accessibility | 0.238 | 0.140 |

**Architecture Comparison Under Controlled Conditions.** To isolate architectural effects, we compare models matched by pretraining corpus (multi-species) and tokenization (BPE), and focus on pairs where the remaining configuration differences are minimized. Under these controlled conditions, Transformer models show substantial advantages over the state-space model available in this controlled setting. Specifically, OMNI-DNA-1B (Transformer-decoder) exceeds ECCDNAMAMBA (Mamba-SSM) by $+0.149$ macro-MCC (0.568 vs. 0.419), and GENOMEOCEAN-500M shows a comparable $+0.131$ gap over the same Mamba baseline (0.550 vs. 0.419). Within Transformers, we also observe an encoder advantage in one matched comparison: GENA-LM-LARGE-T2T (Transformer-encoder) exceeds OMNINA-220M (Transformer-decoder) by $+0.127$ MCC (0.552 vs. 0.425) under matched multi-species/BPE conditions. These gaps hold under both micro- and macro-averaging (Appendix E.4). The encoder-vs-decoder comparison, by contrast, is task- and setting-dependent.

**Architecture Gaps Are Largest on Cross-Species Regulatory Tasks.** Architecture-dependent gaps are particularly pronounced on tasks requiring cross-species generalization (Figure 3; Figure 4). On virus/phage, GENOMEOCEAN-500M (Transformer-decoder) exceeds ECCDNAMAMBA

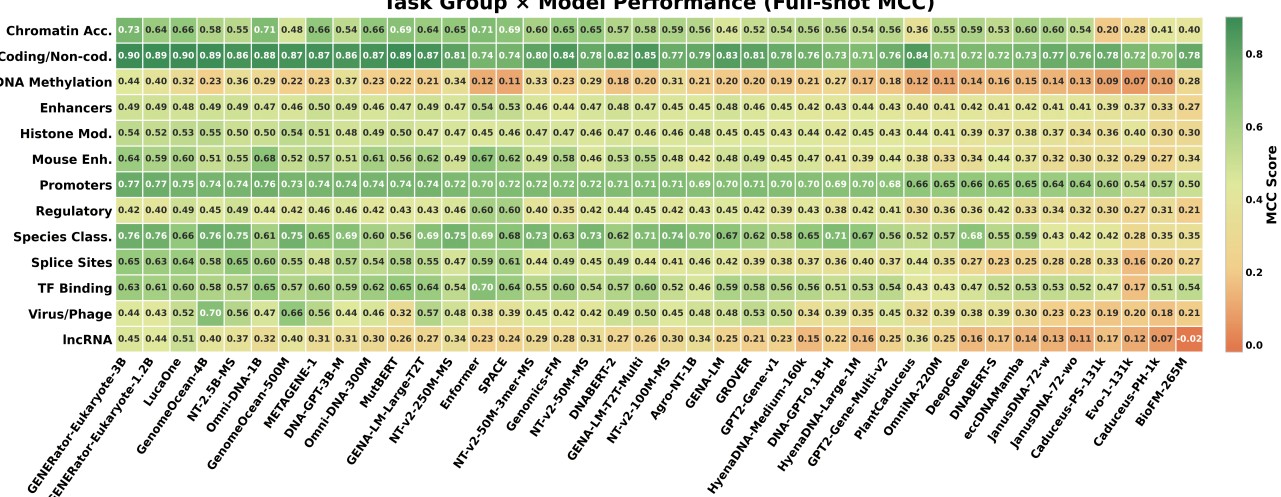

*Figure 3.* **Model performance across task groups.** Heatmap shows full-shot MCC averaged within each task group for 40 genomic foundation models, sorted by overall full-shot macro-average MCC. Cell values report category-level mean MCC, with colors ranging from red/orange for lower scores to green for higher scores. The results reveal substantial task-level heterogeneity: some categories, such as promoter, coding/non-coding, and species-classification tasks, are consistently easier, whereas DNA methylation, lncRNA, virus/phage, and regulatory tasks remain challenging. This category-specific structure shows that aggregate model rankings can hide important differences in downstream behavior.

(Mamba-SSM) by $+0.355$ macro-MCC ($0.657$ vs. $0.302$). On mouse enhancers, OMNI-DNA-1B exceeds ECCD-NAMAMBA by $+0.305$ ($0.675$ vs. $0.370$), and GENA-LM-LARGE-T2T exceeds OMNINA-220M by $+0.284$ under matched multi-species/BPE conditions. These category-level gaps are several-fold larger than the aggregate parameter-tier gain ($+0.064$ macro-MCC between models above 1B and below 200M parameters), reinforcing that architecture and pretraining alignment can exceed scale-related differences on several categories.

**Chromatin Accessibility: A Domain Where SSM Models Become Competitive.** Chromatin accessibility provides a notable exception to the general pattern of Transformer dominance. ECCDNAMAMBA (Mamba-SSM) exceeds GENOMEOCEAN-500M (Transformer-decoder) by $+0.124$ macro-MCC ($0.599$ vs. $0.475$) on this category. Both JANUSDNA-72-W and ECCDNAMAMBA also exhibit a substantial within-model advantage on chromatin accessibility relative to their overall performance ($+0.200$ and $+0.179$ MCC above their respective aggregate macro-MCC), a clear instance of category-level specialization (see Appendix F.11) within the SSM model family. The controlled pretraining-corpus analysis additionally shows a consistent $+0.062$ macro-MCC advantage of multi-species over human-only pretraining for chromatin accessibility (6/6 pairs; Table 2). Within GENEB, these results suggest that chromatin accessibility may benefit from inductive biases captured by the evaluated SSM model family when combined with taxonomically diverse pretraining.

**Tokenization Strategy Effects.** Holding architecture and pretraining corpus fixed, we isolate the effect of tokenization scheme across 12 matched controlled pairs (Appendix E.3). No global ordering emerges; the preferred scheme varies with model family and pretraining setting. Within Transformer-decoders under matched multi-species pretraining, BPE exceeds $k$-mer on average ($+0.020$ macro-MCC across 3 pairs), but with substantial pair-level variance: OMNI-DNA-300M outperforms GPT2-GENE-MULTI-V2 by $+0.071$, while OMNINA-220M underperforms the same baseline by $-0.042$. Within Transformer-encoders, BPE and $k$-mer perform comparably ($+0.006$ across 5 pairs, with all pair-level gaps below $0.02$ MCC). Under matched human pretraining, single-nucleotide tokenization (MUTBERT) exceeds BPE baselines in both available comparisons: $+0.033$ over GENA-LM and $+0.038$ over GROVER. Non-standard vocabularies extend the same pattern: $k$-mer exceeds BIOFM-265M's BioToken framework by $+0.134$ in one Transformer-decoder comparison, while LUCAONE's mixed nucleotide–amino-acid vocabulary exceeds $k$-mer by $+0.017$ over a matched Transformer-encoder (NT-2.5B-MS). Tokenization thus interacts with architecture, scale, and task structure rather than admitting a single global ordering in GENEB.

**Transfer Learning Analysis: Isolating Pretraining Corpus Effects.** To disentangle pretraining data effects from architectural confounds, we restrict analysis to controlled pairs matched on architecture, tokenization, and size (within $\pm 2\times$; Appendix E.3).

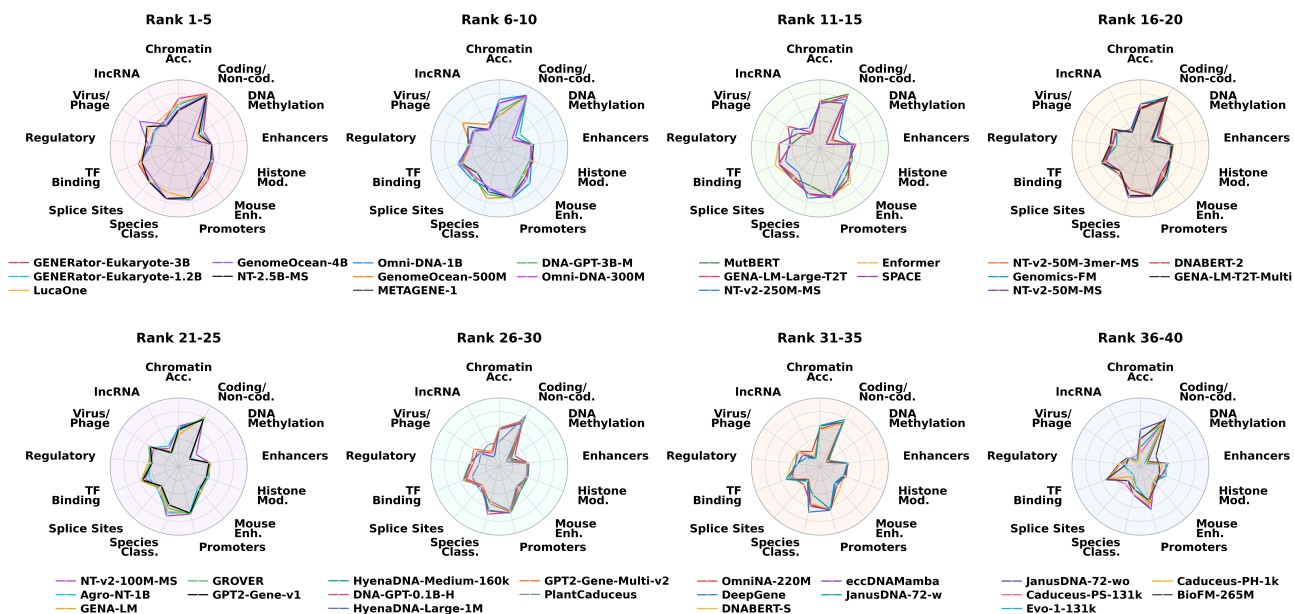

*Figure 4.* **Radar plots for category-aware model selection.** Each subplot shows full-shot macro-MCC across the 13 GENEB task categories for a group of five models, grouped by overall macro-MCC rank from strongest to weakest. The plots expose category-specific strengths not captured by aggregate rankings: ENFORMER has a moderate overall rank but leads on TF binding (0.698), enhancers (0.539), and regulatory tasks (0.604), and ranks second on mouse enhancers (0.674) and third on chromatin accessibility (0.711); the GENOMEOCEAN family is particularly strong on virus/phage tasks; and plant-oriented models such as PLANTCADUCEUS and AGRO-NT-1B show relative strength on lncRNA tasks. These profiles motivate task-specific model selection over global leaderboard position.

*Human vs. multi-species pretraining* (6 controlled pairs across Transformer-encoders/decoders and BPE/$k$-mer) shows a small aggregate effect: multi-species pretraining yields an average +0.012 macro-MCC improvement over human-only. However, the effect is structured by task category (Table 2). Multi-species pretraining shows its most consistent advantage on chromatin accessibility (6/6 pairs; $\Delta = +0.062$), with additional positive shifts on splice sites ($\Delta = +0.038$), species classification ($\Delta = +0.031$), mouse enhancers ($\Delta = +0.023$), and lncRNA ($\Delta = +0.022$). In contrast, virus/phage tasks favor human-only pretraining on average ($\Delta = -0.034$). Categories such as histone modifications, promoters, and TF binding are near parity under the $|\Delta| < 0.02$ criterion.

*Multi-species vs. multi-species-microbial* (2 controlled pairs; Transformer-encoder; $k$-mer) yields the largest corpus effect observed in GENEB controlled comparisons. Models pretrained on general multi-species data that include eukaryotic genomes (NT-v2-100M-MS, GENOMICS-FM) exceed microbial-focused pretraining (DNABERT-S) by +0.084 macro-MCC on average (2/2 pairs). The largest category gaps align with known biological differences between microbial and eukaryotic genomes: splice sites ($\Delta = +0.222$), species classification ($\Delta = +0.130$), lncRNA ($\Delta = +0.116$), and DNA methylation ($\Delta = +0.108$). In GENEB, this is consistent with microbial-focused pretrain-

ing being insufficient for eukaryotic genomic prediction tasks even under matched architectures and tokenization.

*Eukaryotic-genes vs. multi-species* (1 controlled pair; 3B Transformer-decoder; $k$-mer) shows that GENERATOR-EUKARYOTE-3B (curated eukaryotic genes) exceeds DNA-GPT-3B-M (broad multi-species) by +0.063 macro-MCC overall, with the largest advantages on chromatin accessibility (+0.191), lncRNA (+0.142), and mouse enhancers (+0.124). The only category favoring broad multi-species in this comparison is regulatory element prediction (−0.040). This conclusion rests on a single pair and may reflect model-specific training choices (Appendix E.3), but is consistent with corpus curation yielding gains beyond broad sequence diversity.

*What cannot be concluded from controlled comparisons.* Several pretraining corpus types lack matched architectural controls in the current model set and therefore cannot be attributed cleanly to data effects: plant-genome models (unique architectures and scale points), human-mouse epigenomic profile models (ENFORMER, SPACE; CNN-Transformer hybrids), and the prokaryotic model (EVO-1-131K; StripedHyena). Apparent advantages or deficits for these groups should be interpreted as potentially confounded by architecture and training procedure rather than pretraining data alone.

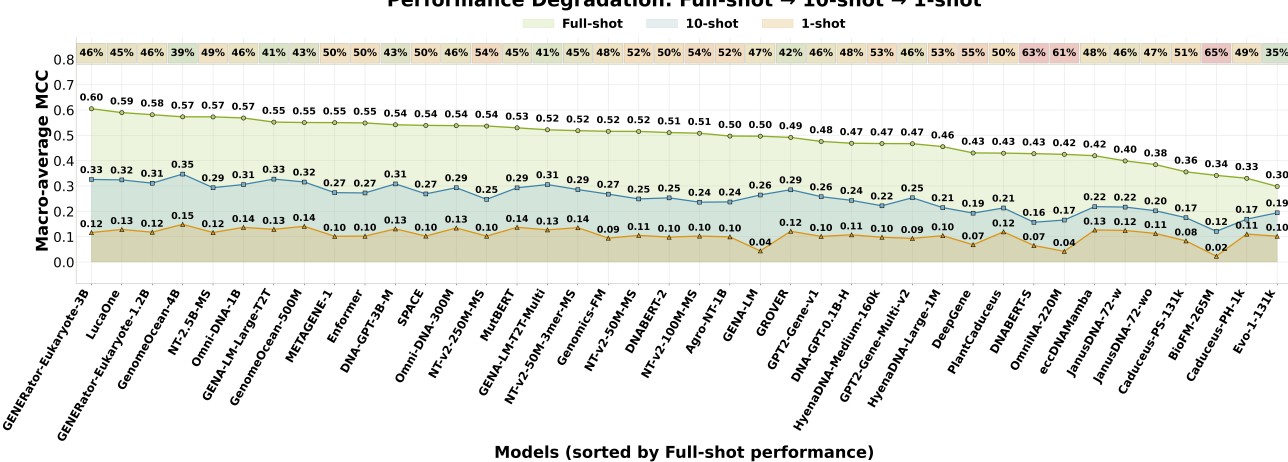

*Figure 5.* **Few-shot performance degradation.** Macro-average MCC of genomic foundation models under full-data, 10-shot, and 1-shot evaluation regimes. Models are sorted by full-data performance. The top band reports the relative performance drop from full-data to 10-shot evaluation, highlighting the sensitivity of each model to limited supervision.

*Table 2.* **Transfer learning: human vs. multi-species pretraining.** Per-category $\Delta$ MCC (multi-species − human) averaged across 6 controlled model pairs matched on architecture, tokenization, and size ($\pm 2\times$). For each pair, the within-category MCC is computed by averaging over tasks; the table reports the cross-pair average. "Wins" indicates the number of pairs in which multi-species exceeds human in that category. "Overall" is the macro-averaged aggregate ($\Delta$ MCC averaged across the 13 categories).

| Task Category | $\Delta$ MCC | Wins |
|---|---|---|
| **Overall (macro)** | **+0.012** | **4/6** |
| Chromatin Acc. | +0.062 | 6/6 |
| Splice Sites | +0.038 | 4/6 |
| Species Class. | +0.031 | 3/6 |
| Mouse Enh. | +0.023 | 4/6 |
| lncRNA | +0.022 | 5/6 |
| Histone Mod. | +0.009 | 4/6 |
| Regulatory | +0.008 | 3/6 |
| DNA Methylation | +0.005 | 2/6 |
| Coding/Non-cod. | +0.000 | 3/6 |
| Enhancers | −0.001 | 3/6 |
| Promoters | −0.001 | 2/6 |
| TF Binding | −0.005 | 2/6 |
| Virus/Phage | −0.034 | 2/6 |

Green: multi-species advantage ($\Delta > +0.02$). Gray: parity ($|\Delta| \leq 0.02$). Red: human advantage ($\Delta < -0.02$).

**Few-Shot Robustness Reveals an Inverse Performance Pattern.** Across all 40 models, mean macro-MCC degrades from 0.488 (full-shot) to 0.253 (10-shot) to 0.106 (1-shot) (Figure 5), corresponding to relative reductions of 48.2% and 78.2%, respectively. Per-model relative 10-shot drops vary substantially, from 35% for EVO-1-131K to 65% for BIOFM-265M (median 48%), reflecting both category-level and model-level effects that the aggregate trend does not capture.

*First*, the degradation is structured by task category (per-category breakdown in Appendix F). Promoter prediction retains 38.8% of full-shot macro-MCC at 1-shot and species classification retains 30.1%, consistent with broadly distributed sequence-composition signals (GC content, codon usage, $k$-mer enrichment) that pretraining captures and 1-shot supervision preserves. In contrast, three categories collapse to within 0.03 MCC of random performance at 1-shot: virus/phage ($\text{MCC}_{\text{1-shot}} = 0.027$, 93.5% drop), DNA methylation (0.015, 93.2%), and lncRNA (0.022, 91.3%). These collapses align with under-representation of viral genomes in eukaryotic pretraining corpora, the position-specific nature of DNA methylation signals, and the low-level sequence similarity of lncRNA to the broader non-coding genome.

*Second*, beyond this category-level structure, few-shot robustness is inversely aligned with full-shot performance at the model level. The five smallest absolute drops occur in EVO-1-131K ($\Delta = 0.196$), CADUCEUS-PH-1K (0.220), JANUSDNA-72-WO (0.272), CADUCEUS-PS-131K (0.272), and JANUSDNA-72-W (0.275) – all among the weakest models in full-shot. Top full-shot performers exhibit the opposite pattern: GENERATOR-EUKARYOTE-3B ($\Delta = 0.489$), GENERATOR-EUKARYOTE-1.2B (0.463), LUCAONE (0.461), and NT-2.5B-MS (0.456) all exceed 0.42 in absolute drop. This pattern does not indicate greater robustness in weaker models: a small absolute drop reflects a low full-shot ceiling that leaves limited room for further degradation, not recovery of useful signal at 1-shot. EVO-1-131K is illustrative – its low ceiling stems from domain mismatch (see Domain Mismatch paragraph below), not from few-shot regime itself.

These observations expose a structural limitation of aggregate few-shot leaderboards: they conflate category tractabil-

ity with model quality, and reward models whose absolute drop is small because their full-shot performance is already low. Category-resolved evaluation (Appendix F) is therefore the appropriate diagnostic in low-supervision regimes.

**The Hard Frontier of GENEB: Where Scaling Does Not Help.** A substantial fraction of GENEB tasks remains far from saturation: 28 of 100 tasks have mean MCC below 0.35 (Figure 3), dominated by 4mC methylation prediction (*G. subterraneus* 0.061; *E. coli* 0.103; *G. pickeringii* 0.107) and plant lncRNA identification (*S. lycopersicum* 0.221; *G. max* 0.228; *T. aestivum* 0.238). Even the strongest model on these categories, GENERATOR-EUKARYOTE-3B, reaches only 0.206–0.477 on 4mC tasks, and LUCAONE reaches 0.417–0.629 on plant lncRNA.

Category-level scaling analysis (Spearman $\rho$ between $\log(\text{params})$ and per-category macro-MCC across all 40 models; Table 1) shows that this hardness is not purely a parameter-count problem. Eleven of the 13 categories show positive scaling significant at $p < 0.05$, but with substantial variation in strength ($\rho$ ranging from 0.347 for DNA methylation to 0.579 for histone modifications). Two categories show no significant scaling: species classification ($\rho = 0.304$, $p = 0.056$) and chromatin accessibility ($\rho = 0.245$, $p = 0.128$). Even where scaling is statistically significant, the hard frontier of GENEB (4mC methylation, plant lncRNA) shows that scaling alone does not close the absolute performance gap; progress on these tasks will require complementary advances in pretraining-corpus design, inductive biases, or task-specific supervision.

**High-Variance Tasks Reveal Decisive Design Patterns.** The architectural and pretraining effects identified above are most pronounced on tasks where models disagree most: 13 GENEB tasks exhibit cross-model standard deviation above 0.12, indicating settings in which model choice has outsized impact on downstream performance (Figure 6A). Aggregating top-3 and bottom-3 placements across these tasks (39 slots in each band) reveals concentration effects (Figure 6B). By pretraining-data type, multi-species and eukaryotic-gene pretraining together capture 32/39 top-3 placements (20 and 12, respectively), while human-only pretraining is concentrated almost entirely in the bottom band (29/39 bottom-3, only 1/39 top-3). By architecture, the same imbalance holds: Transformer-decoders and Transformer-encoders together account for 33/39 top-3 placements (18 and 15, respectively), whereas Mamba-SSM (17/39 bottom-3), Hybrid-Mamba-MoE (7/39), and StripedHyena (6/39) dominate the bottom. These high-variance tasks thus operationalise the findings above into a practical diagnostic: in settings where GENEB models disagree most, pretraining scope and architectural family – not scale – predict whether a model will land in the top or bottom tier.

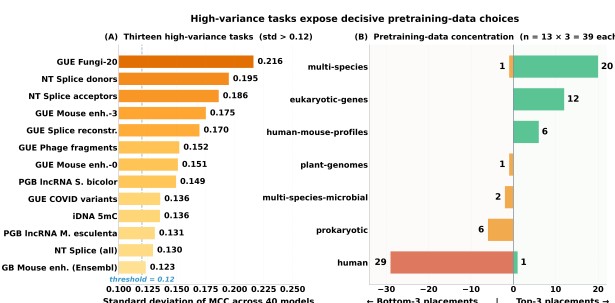

*Figure 6.* **High-variance tasks reveal the role of pretraining data.** **(A)** GENEB tasks with cross-model standard deviation above 0.12, corresponding to settings where model selection most strongly affects downstream performance. **(B)** Pretraining-data composition of top-3 and bottom-3 placements across these tasks. Multi-species and eukaryotic-gene pretraining dominate top placements, while human-only, prokaryotic, and microbial pretraining are concentrated among bottom placements. The result indicates that high-variance tasks expose biologically meaningful differences in pretraining scope that are obscured by aggregate leaderboards.

**Domain Mismatch and Out-of-Domain Models.** The poor aggregate performance of EVO-1-131K – the largest model in GENEB (7B parameters), yet ranked among the weakest on TF binding, mouse enhancers, chromatin accessibility, species classification, and splice sites (Figure 4) – is best understood as a symptom of a broader issue: the GENEB task suite is heavily skewed toward eukaryotic genomic phenomena (12 of 13 categories), creating a structural disadvantage for models pretrained on prokaryotic or microbial corpora. This disadvantage is distinct from – and should not be conflated with – the effects of scale, architecture, or tokenization. Recomputing the scale–performance correlation without EVO-1-131K raises Spearman $\rho$ from 0.565 to 0.685 ($p < 0.001$) and widens the tier gap between models above 1B and below 200M parameters from +0.064 to +0.087 macro-MCC. DNABERT-S, pretrained on multi-species microbial genomes, exhibits a milder version of the same pattern: overall macro-MCC 0.427 versus 0.512 for the two eukaryotic multi-species models under matched architecture and tokenization (NT-V2-100M-MS, GENOMICS-FM; see Transfer Learning Analysis above). We therefore treat aggregate rankings as a poor proxy for performance in prokaryotic or viral genomics, and encourage users of GENEB working in those domains to consult per-task results directly rather than relying on the overall leaderboard.

**Practitioner Recommendations.** The findings above motivate per-category model selection rather than reliance on aggregate rankings (Figure 4). The per-category recommendations below are organised around four recurring patterns. For compact deployment under tight compute budgets, MUTBERT (86M, Transformer-encoder, single-nucleotide

tokenization, human pretraining) is the top sub-100M model in 8 of 13 categories and the strongest $\leq$ 100M model on overall macro-MCC (0.529). For epigenomic-profile tasks (TF binding, regulatory, enhancers), the CNN–Transformer hybrids ENFORMER and SPACE are consistently among the top models, with ENFORMER matching the top model OMNI-DNA-1B on mouse enhancers. At the opposite end, hard regimes (DNA methylation, plant lncRNA) remain unsolved within GENEB: the best macro-MCC is 0.440 for DNA methylation (GENERATOR-EUKARYOTE-3B) and 0.508 for plant lncRNA (LUCAONE), with scaling weak for DNA methylation ($\rho = 0.347$, Table 1). Finally, few-shot evaluation reranks the best model in 8 of 13 categories, so the model winning under full supervision is not necessarily the model to deploy with only $\sim$ 10 labelled examples per task.

*Coding/non-coding.* GENERATOR-EUKARYOTE-3B (0.904) and LUCAONE (0.901) lead under full supervision; the most compact strong alternative is MUTBERT (0.894, 86M), which nearly matches the 3B-parameter leader. In 10-shot deployments, MUTBERT becomes the top performer (0.748 vs. 0.694 for GENERATOR-EUKARYOTE-3B).

*Promoter prediction.* GENERATOR-EUKARYOTE-3B (0.774) leads, with GENERATOR-EUKARYOTE-1.2B (0.768) and OMNI-DNA-1B (0.759) close behind. Under tighter compute budgets, OMNI-DNA-300M (0.740) and MUTBERT (0.739, 86M) provide strong compact options.

*Species classification.* The top three models – GENOMEOCEAN-4B (0.762), GENERATOR-EUKARYOTE-3B (0.761), and GENERATOR-EUKARYOTE-1.2B (0.757) – all exceed 0.75 and use multi-species or eukaryotic-gene pretraining. For compute-constrained deployment, the NT-V2 family (NT-V2-250M-MS 0.747, NT-V2-100M-MS 0.741, NT-V2-50M-3MER-MS 0.734) offers near-equivalent performance at one-tenth the parameter count.

*Chromatin accessibility.* GENERATOR-EUKARYOTE-3B (0.728), OMNI-DNA-1B (0.714), and ENFORMER (0.711) all exceed 0.71; ENFORMER (250M) is the strongest sub-300M option, and MUTBERT (86M, 0.691) the strongest sub-100M option. The category also stands out as a setting in which the evaluated SSM model is competitive (ECCDNAMAMBA 0.599). Under 10-shot supervision the ranking reranks substantially: GENA-LM-LARGE-T2T becomes the leader (0.567, down from 0.645 at full-shot), while GENERATOR-EUKARYOTE-3B drops to 0.540.

*TF binding.* ENFORMER (0.698) leads by a wide margin, followed by OMNI-DNA-1B (0.647) and MUTBERT (0.646, 86M). MUTBERT is therefore the recommended compact choice.

*Virus/phage classification.* GENOMEOCEAN-4B (0.697)

and GENOMEOCEAN-500M (0.657) dominate this category, with a sharp drop to GENA-LM-LARGE-T2T (0.569); below 200M parameters performance drops further (GROVER 0.532). GENOMEOCEAN-4B remains the 10-shot leader (0.377), though absolute scores collapse substantially.

*Mouse enhancers.* OMNI-DNA-1B (0.675) and ENFORMER (0.674) are effectively tied at full supervision; ENFORMER (250M) is therefore the recommended compact choice. Under 10-shot supervision, however, SPACE becomes the leader (0.379, from a full-shot score of 0.618), while OMNI-DNA-1B drops to 0.253 – a substantial rerank that practitioners working with limited cross-species labels should account for.

*Splice sites.* NT-2.5B-MS (0.652) and GENERATOR-EUKARYOTE-3B (0.648) lead under full supervision. The strongest sub-300M alternative is ENFORMER (0.586); the strongest sub-100M alternative is MUTBERT (0.579). Microbial-focused pretraining transfers poorly to this category, losing 0.222 macro-MCC to multi-species in controlled comparisons (see Transfer Learning Analysis above). Under 10-shot supervision, LUCAONE becomes the top model (0.298, from a full-shot score of 0.636), while NT-2.5B-MS drops to 0.242.

*Regulatory.* ENFORMER (0.604) and SPACE (0.598) lead substantially, reflecting their epigenomic-profile pretraining. The strongest sub-100M option is HYENADNA-MEDIUM-160K (0.432). In 10-shot, DNABERT-2 unexpectedly leads (0.216, from a full-shot score of 0.435), while ENFORMER drops to 0.184; we view this rerank as protocol-fragile given the small absolute scores in this low-data regime.

*Histone modifications.* The GENOMEOCEAN family leads (GENOMEOCEAN-4B 0.545, GENOMEOCEAN-500M 0.537), along with GENERATOR-EUKARYOTE-3B (0.537). With 30 tasks, this is the largest category in GENEB and shows the strongest within-category scaling ($\rho = 0.579$; Table 1). The strongest sub-100M option is MUTBERT (0.501). In 10-shot, MUTBERT also leads (0.300), making it the recommended choice across regimes when compute is constrained.

*Enhancers.* ENFORMER (0.539) and SPACE (0.526) lead, followed by METAGENE-1 (0.505). The strongest sub-300M alternative is ENFORMER itself. Under 10-shot, GENA-LM-LARGE-T2T becomes the top model (0.372, from 0.488 at full-shot), a switch worth noting for low-data deployments.

*lncRNA.* LUCAONE (0.508) leads by a substantial margin, followed by GENERATOR-EUKARYOTE-3B (0.453) and GENERATOR-EUKARYOTE-1.2B (0.438). The strongest sub-300M option is PLANTCADUCEUS (0.357), reflecting

its plant-specific pretraining. This is a hard regime: no per-task score exceeds 0.63 across any of the 6 plant lncRNA tasks, and 10-shot performance is at or below 0.21 for every model.

*DNA methylation.* GENERATOR-EUKARYOTE-3B (0.440) leads, with GENERATOR-EUKARYOTE-1.2B (0.397) and DNA-GPT-3B-M (0.367) following. This is the hardest category in GENEB: 6 of 8 tasks have mean MCC below 0.25; although scaling is statistically significant ($\rho = 0.347$, $p = 0.028$; Table 1), no sub-300M model exceeds 0.34. Few-shot performance collapses to near-random levels across all models, with no 1-shot score exceeding 0.04.

## 5. Conclusion

We introduced GENEB, a benchmark evaluating 40 genomic foundation models across 100 tasks from 13 functional categories under a unified linear-probing protocol with macro-MCC as the principal aggregation metric. Our results show that while model scale shows a substantial aggregate association with performance ($\rho = 0.565$), it remains an imperfect predictor of category-level outcomes: architecture and pretraining alignment frequently offset substantial scale differences, and the model that wins under full supervision reranks under 10-shot evaluation in 8 of 13 categories. Transformer-based models generally outperform the evaluated state-space alternative, though domain-specific exceptions exist (e.g., chromatin accessibility); tokenization effects interact with architecture rather than admitting a single global ordering; and microbial-only corpora transfer poorly to eukaryotic tasks.

Overall, these findings argue for category-aware, controlled evaluation rather than aggregate leaderboards. GENEB provides a reference framework to support principled model comparison and selection in genomic machine learning.

## 6. Limitations

GENEB has several limitations.

*Long-range tasks.* GENEB underrepresents tasks requiring explicit modeling of very long-range regulatory interactions ($> 10\,\mathrm{kb}$). As a result, models with explicit long-context capability (HYENADNA-LARGE-1M, CADUCEUS-PS-131K, EVO-1-131K) are not exercised on the regime where their architectural priors would most likely yield differentiating gains. A detailed enumeration of considered-but-excluded long-range datasets and model context-length constraints is provided in Appendix D.3.

*Task selection and curation.* Task selection is constrained by available datasets and existing benchmarks. Some constituent tasks may be noisy or weakly defined, particularly in

the hard regimes identified in Section 4 (DNA methylation, plant lncRNA), where label quality and supervision signal vary across sources. Further task curation and refinement is needed as genomic benchmarks mature.

*Model coverage.* Not all genomic foundation models could be included due to unavailable weights, incompatible pipelines, or computational constraints. Excluded models and inclusion criteria are discussed in Appendix D.

*Prokaryotic and viral task gap.* Of the 13 GENEB categories, only virus/phage classification reflects a non-eukaryotic domain; prokaryotic gene prediction, microbial genome assembly verification, and CRISPR system characterization are not currently represented. As a result, aggregate GENEB rankings are an unreliable proxy for performance in prokaryotic or viral genomics (see Domain Mismatch paragraph, Section 4).

*Frozen representations and pooling-tokenization interactions.* GENEB evaluates frozen representations using linear probing, which enables controlled comparison of embedding quality across the 40-model set but may underestimate the performance achievable with task-specific fine-tuning. Empirical analysis in Appendix E.1 shows that model rankings under linear probing are highly consistent with those under non-linear MLP probes; whether this stability extends to full task-specific fine-tuning remains an open question. Additionally, single-nucleotide and $k$-mer tokenizations produce substantially longer token sequences than BPE for a fixed input window, so the choice of pooling (mean, attention-weighted, or final-token) may favor different schemes. We use mean pooling throughout; pooling-tokenization interactions are not fully disentangled.

*Aggregate metric.* GENEB reports both micro- and macro-averaged MCC; histone modifications (30 tasks) and promoters (22 tasks) jointly account for over half of the per-task evaluations, so micro-averaging is structurally biased toward these two categories (see Appendix E.4). We treat macro-MCC as the principal aggregation, and recommend that category-level results, rather than any single overall ranking, drive model selection.

## 7. Use of Large Language Models

Large language models were used as writing and editing assistants during the preparation of this manuscript. Specifically, they were employed to improve clarity, organization, and phrasing of the text, as well as to assist with LaTeX formatting. All experimental design, data processing, analysis, and interpretation of results were performed by the authors, and all reported findings were verified against the underlying benchmark outputs.

## Acknowledgements

This work was supported by the Ministry of Economic Development of the Russian Federation (agreement No. 139-15-2025-013, dated June 20, 2025, subsidy identifier 000000C313925P4B0002).

## Impact Statement

GENEB is intended to improve the rigor of model comparison in genomic representation learning by replacing heterogeneous, single-paper evaluations with a unified protocol across 40 models and 100 tasks. We expect several positive effects on the field. First, category-aware evaluation reduces the risk that practitioners select models based on aggregate leaderboards that mask substantial heterogeneity across biological task types, particularly in clinically and agriculturally relevant domains (e.g., regulatory element prediction, plant lncRNA, viral classification). Second, our controlled comparisons isolate the contributions of architecture, tokenization, and pretraining corpus, which we hope will inform principled design of future genomic foundation models rather than indiscriminate scaling.

We are mindful of several potential concerns. Genomic models can in principle be applied to dual-use research, including the design of pathogenic sequences; we believe a benchmark of representation quality on standard prediction tasks does not meaningfully shift this risk surface, but we note that responsible release practices for individual models remain the responsibility of their authors. Additionally, GENEB inherits biases from its constituent datasets: the task suite is skewed toward eukaryotic and, within eukaryotes, toward human and well-studied model organisms, which may under-represent biologically and clinically important non-model-organism settings. Users of GENEB should consult per-task and per-category results when applying findings to domains beyond those directly represented in the benchmark.

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

## A. Related Work

The rapid development of foundation models for genomics has led to diverse architectural and methodological innovations (see Figure 7). We organize prior work by architectural design, tokenization, pretraining strategy, and benchmark scope.

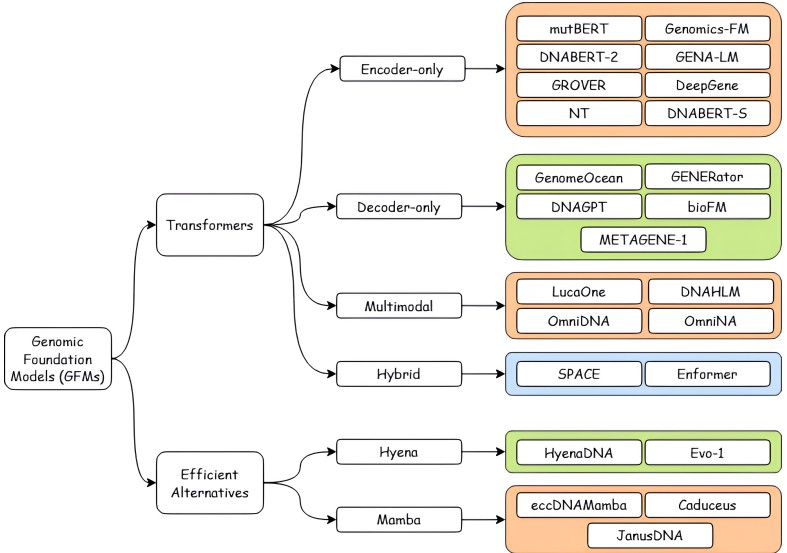

*Figure 7.* Architectural taxonomy of genomic foundation models.

**Transformer-Based Encoder Models.** Early DNA language models predominantly adopted BERT-style encoder architectures with masked language modeling (MLM) objectives. DNABERT-2 (Zhou et al., 2024a) addressed computational inefficiencies of $k$-mer tokenization by replacing it with Byte-Pair Encoding (BPE), achieving comparable performance with $21\times$ fewer parameters. The Nucleotide Transformer (Dalla-Torre et al., 2023) scaled this paradigm to 2.5B parameters, demonstrating that multi-species pretraining improves cross-species generalization. GENA-LM (Fishman et al., 2023) extended context lengths to 36kb using sparse attention mechanisms (BigBird), while GROVER (Sanabria et al., 2024) introduced frequency-balanced BPE vocabularies to mitigate the "rare word problem" in genomic sequences.

**Autoregressive and Generative Models.** Decoder-only architectures have gained prominence for their generative capabilities and unified training objectives. DNA-GPT (Zhang et al., 2023) introduced a comprehensive token language incorporating sequence, numerical, and instruction tokens for multi-task genomic analysis. Evo (Nguyen et al., 2024) achieved genome-scale sequence modeling using the StripedHyena architecture, processing up to 131kb contexts with single-nucleotide resolution, with Evo 2 (Brixi et al., 2025b) extending this to all domains of life. GENERator (Wu et al., 2025) demonstrated that gene-focused pretraining on eukaryotic sequences outperforms whole-genome approaches for both understanding and generation tasks. GenomeOcean (Zhou et al., 2025) pioneered training on metagenomic assemblies, capturing rare biosphere diversity across 645Gbp of environmental sequences. Omni-DNA (Li et al., 2025b) unified cross-modal tasks (DNA-to-text, DNA-to-image) within a single autoregressive framework through vocabulary expansion.

**State Space Models and Efficient Architectures.** To address the quadratic complexity of attention mechanisms, several works have explored sub-quadratic alternatives. HyenaDNA (Nguyen et al., 2023) employed implicit long convolutions via the Hyena operator, enabling processing of sequences up to 1M nucleotides with $\mathcal{O}(L \log L)$ complexity. Caduceus (Schiff et al., 2024) built upon the Mamba selective state space model, introducing bidirectional processing and reverse-complement (RC) equivariance to respect DNA's double-stranded nature. eccDNAMamba (Liu et al., 2025) adapted the Mamba-2 architecture specifically for extrachromosomal circular DNA, incorporating circular topology awareness through specialized augmentation strategies.

**Hybrid Architectures.** Recent work has combined the strengths of different architectural paradigms. JanusDNA (Duan et al., 2025) introduced a bidirectional Mamba-Attention-MoE hybrid that achieves both autoregressive efficiency and bidirectional understanding, processing up to 1M base pairs at single-nucleotide resolution. Gene42 (Vishniakov et al.,

2025b) extended dense attention to 192kb through incremental pretraining with adjusted RoPE frequencies.

**Multi-Modal and Multi-Species Models.** Several recent models aim for broader biological understanding through multi-modal integration. LucaOne (He et al., 2024) unified nucleic acid and protein modeling within a single 1.8B-parameter encoder using a shared 39-token vocabulary, demonstrating emergent understanding of the central dogma. OmniNA (Shen & Li, 2024) integrated nucleotide sequences with textual annotations, enabling natural language-based genomic analysis across 17 diverse tasks.

**Long-Range Genomic Modeling.** Modeling distal regulatory interactions requires extended context lengths. Enformer (Avsec et al., 2021) combined convolutional towers with 11 transformer layers to capture interactions up to 100kb, significantly improving gene expression prediction from sequence. DNALongBench (Cheng et al., 2025) provides a comprehensive benchmark for evaluating long-range dependencies up to 1M base pairs. SPACE (Yang et al., 2025c) proposed supervised pretraining on genomic profile prediction as an alternative to self-supervised approaches, employing Mixture-of-Experts (MoE) for cross-species modeling and profile-grouped decoders for capturing inter-track dependencies.

**Tokenization Strategies.** Tokenization choices significantly impact model performance and efficiency. Single-nucleotide tokenization (Nguyen et al., 2023; Duan et al., 2025) preserves fine-grained resolution critical for SNP analysis but increases sequence length. $K$-mer approaches (Dalla-Torre et al., 2023) capture local context but suffer from vocabulary explosion and sensitivity to mutations. BPE-based methods (Zhou et al., 2024a; 2025) balance efficiency and biological relevance by learning data-driven vocabularies. BioToken (Medvedev et al., 2025) introduced biologically-informed tokenization that explicitly encodes structural annotations (exon/intron boundaries) and variant information within the token vocabulary, achieving strong performance with only 265M parameters.

**Specialized Domains.** Beyond general-purpose models, specialized architectures address specific genomic applications. DNABERT-S (Zhou et al., 2024b) targets species differentiation through contrastive learning with Manifold Instance Mixup, doubling clustering performance over baselines. DeepGene (Zhang et al., 2024) incorporates pan-genome graph representations via Minigraph to capture population-level genetic diversity. MutBERT (Long et al., 2025) represents the genome as probabilistic distributions over allele frequencies, explicitly modeling population-level SNP variation. Genomics-FM (Ye et al., 2024) employs an ensemble vocabulary strategy, pretraining on multiple tokenizations simultaneously and selectively activating task-appropriate vocabularies during fine-tuning. PlantCaduceus (Zhai et al., 2025) adapts the Caduceus architecture for plant genomics, demonstrating remarkable cross-species transferability across 160My of evolutionary divergence. AgroNT (Mendoza-Revilla et al., 2023) provides a specialized foundation model for edible plant genomes.

**Benchmarks for Genomic Foundation Models.** Standardized evaluation is essential for assessing the generalization capabilities of genomic foundation models, yet existing benchmarks typically focus on limited task families or biological domains. The Nucleotide Transformer benchmark (Dalla-Torre et al., 2023) introduced a suite of DNA classification tasks spanning histone mark occupancy, regulatory element identification, and splice site prediction, establishing a common evaluation baseline for early Transformer-based models. The Genome Understanding Evaluation (GUE/GUE+) (Zhou et al., 2024a) extended this paradigm to multi-species settings, including human, mouse, virus, and yeast, and to variable sequence lengths ranging from tens to thousands of base pairs. Genomic Benchmarks (Gresova et al., 2022) further emphasized regulatory element classification, such as enhancers, promoters, coding/non-coding regions, and open chromatin, across diverse organisms.

Beyond these core resources, additional benchmarks target specific biological regimes. Plant-focused benchmarks evaluate long non-coding RNA classification and transcriptional activity in crop genomes (Yang et al., 2025b), while DNA methylation datasets assess epigenetic modifications (4mC, 5mC, 6mA) across a range of taxa (Feng et al., 2024). Collectively, these benchmarks have enabled substantial progress, but they remain fragmented in task scope, preprocessing assumptions, and evaluation protocols.

**Limitations of Existing Benchmarking Practices.** A common limitation across prior benchmarks is the restricted set of evaluated models. Most studies assess a small number of representative architectures, often selected for convenience or compatibility with a specific benchmark, rather than attempting exhaustive cross-model evaluation. As a result, comparisons across papers are difficult to interpret, and it is often unclear which models constitute appropriate baselines for a given task.

Moreover, reported state-of-the-art results frequently depend on benchmark-specific design choices, obscuring task-level trade-offs and failure modes.

Recent benchmarking efforts such as BEND (Marin et al., 2024) and DNALongBench (Cheng et al., 2025) have highlighted important evaluation issues, including biologically meaningful task design and long-range dependency modeling. However, these benchmarks still evaluate limited model subsets and do not provide a unified performance matrix across a broad landscape of genomic foundation models.

**Comparative Benchmarking Studies.** Several recent works have attempted broader comparative analysis of genomic foundation models. For example, Wang et al. (2025b) evaluate approximately ten representative model families, primarily centered on human functional annotation tasks. While such studies provide useful snapshots, the evaluated model sets remain limited and exclude many recently proposed architectures. In addition, results are typically reported in static tables without unified leaderboards, making it difficult to place new models in a broader performance context. As a consequence, researchers often lack clear guidance on which baselines are appropriate for a given task and how to position new models relative to existing work.

Platform-centric initiatives such as OmniGenBench (Wang et al., 2025a) have made important strides toward reproducibility by providing modular infrastructure and dynamic leaderboards. However, their reported baselines are often limited to a small set of established models (e.g., DNABERT-2, HyenaDNA, RNA-FM) and place substantial emphasis on RNA modalities. Consequently, a large and rapidly evolving portion of the DNA-specific foundation model landscape remains underexplored.

**Positioning of GENEB.** GENEB is designed to complement and extend prior benchmarking efforts by enforcing exhaustive and controlled evaluation across a substantially broader model and task space. GENEB aggregates 100 DNA classification tasks drawn from multiple established benchmarks, including Nucleotide Transformer tasks, Genomic Benchmarks, GUE/GUE+, plant genomics benchmarks, and DNA methylation datasets, spanning 13 functional categories. In contrast to prior work, GENEB evaluates all included models on the full task suite under a unified probing-based protocol, regardless of the model's original domain or anticipated strengths.

This exhaustive cross-evaluation yields a complete, static performance matrix over 40 genomic foundation models, including recent architectures such as BioFM, GENERator, JanusDNA, and DeepGene that are absent from many existing benchmarks. By eliminating selective reporting and enforcing matched comparisons across architecture, tokenization, and pretraining data, GENEB exposes task-dependent trade-offs and failure modes that are often obscured in partial or benchmark-specific evaluations.

Beyond the static analysis presented in this work, we intend to release GENEB as a public benchmark with model evaluations hosted on Hugging Face, enabling transparent comparison and reproducible evaluation of future genomic foundation models. In this sense, GENEB is conceptually aligned with MTEB (Muennighoff et al., 2023) in natural language processing: a community-facing benchmark that provides a shared reference point for evaluating representation quality across diverse tasks. We hope that GENEB can serve a similar role for genomic foundation models, supporting more principled model comparison and task-aware model selection.

## B. Task Taxonomy and Benchmark Composition

GENEB aggregates tasks from nine widely used genomic benchmarks to ensure broad biological coverage and comparability to prior work: **NT** (Nucleotide Transformer original, 18 tasks) and **NT-rev** (its revised release, 18 tasks) covering histone marks, enhancers, promoters, and splice sites; **GUE** (31 tasks) covering empirically-derived histone marks, TF binding, mouse enhancers, promoters, splice sites, phage fragments, and species classification; **GB** (Genomic Benchmarks, 9 tasks) covering enhancers, promoters, coding/non-coding, regulatory elements, and open chromatin; **PGB** (Plant Genomic Benchmark, 7 tasks) covering plant lncRNA and transcriptional activity; **iPro-WAEL** (8 tasks) covering bacterial, plant, and human cell-type-specific promoter recognition; **deep4mc** (6 tasks) and **iDNA-ABF** (2 tasks) covering DNA methylation (4mC, 5mC, 6mA) across taxa; and **iDHS-EL** (1 task) for chromatin accessibility.

Our benchmark comprises 100 classification tasks organized into 13 functional categories spanning regulatory element prediction, epigenomic modification detection, and evolutionary sequence analysis. This taxonomic organization reflects the diverse challenges facing DNA foundation models and enables systematic analysis across distinct biological domains. Table 3 summarizes the benchmark composition, with task counts ranging from single-task categories (Coding/Non-coding,

Chromatin Accessibility) to large multi-task groups (Histone Modifications with 30 tasks, Promoters with 22 tasks).

*Table 3.* Benchmark task organization across 13 functional categories. Tasks span regulatory element prediction, epigenetic modification detection, and evolutionary sequence analysis across bacterial, plant, and mammalian systems. Within each row, the source prefix (e.g., NT, GUE, GB) is given once and applies to all listed tasks.

| Category | Subcategory | $n$ | Representative Tasks |
|---|---|---|---|
| Histone Modifications | Original (NT) | 10 | H3, H3K14ac, H3K36me3, H3K4me1/2/3, H3K79me3, H3K9ac, H4, H4ac |
| | Empirical (GUE EMP) | 10 | H3, H3K14ac, H3K36me3, H3K4me1/2/3, H3K79me3, H3K9ac, H4, H4ac |
| | Revised (NT-rev) | 10 | H2AFZ, H3K27ac, H3K27me3, H3K36me3, H3K4me1/2/3, H3K9ac, H3K9me3, H4K20me1 |
| Promoters | Human cell-type (iPro) | 4 | iPro: GM12878, HUVEC, HeLa-S3, NHEK |
| | Plant | 3 | iPro Arabidopsis (TATA / no TATA), PGB Promoter *M. esculenta* |
| | Bacterial (iPro) | 2 | iPro: *B. amyloliquefaciens*, *R. capsulatus* |
| | General | 13 | GUE Prom 300 (all / TATA / no TATA), GUE Prom core (all / TATA / no TATA), NT Promoter (all / TATA / no TATA), NT-rev Promoter (all / TATA / no TATA), GB Promoter (no TATA) |
| Enhancers | Species-specific (GB) | 4 | GB: Drosophila enh. (STARK), Mouse enh. (Ensembl), Human enh. (Cohn), Human enh. (Ensembl) |
| | Type classification | 4 | NT Enhancers, NT Enhancers (types), NT-rev Enhancers, NT-rev Enhancers (types) |
| DNA Methylation | 5mC / 6mA (iDNA) | 2 | iDNA: 5mC, 6mA |
| | 4mC (6 species) | 6 | 4mC: *A. thaliana*, *C. elegans*, *D. melanogaster*, *E. coli*, *G. pickeringii*, *G. subterraneus* |
| Splice Sites | Site-specific | 4 | NT Splice (donors / acceptors), NT-rev Splice (donors / acceptors) |
| | Combined | 3 | NT Splice (all), NT-rev Splice (all), GUE Splice reconstr. |
| lncRNA | Plant (6 species; PGB) | 6 | PGB lncRNA: *G. max*, *M. esculenta*, *S. bicolor*, *S. lycopersicum*, *T. aestivum*, *Z. mays* |
| Mouse Enhancers | Tissue-specific (GUE) | 5 | GUE Mouse enh.: -0, -1, -2, -3, -4 |
| TF Binding | Human TFs (GUE) | 5 | GUE TF: -0, -1, -2, -3, -4 |
| Species Classification | Binary | 1 | GB Human-or-worm |
| | Multi-class (GUE) | 2 | GUE: Fungi-20, Virus-40 |
| Regulatory | Human regulatory (GB) | 2 | GB: Ensembl regulatory, OCR Ensembl |
| Virus/Phage | Viral sequences (GUE) | 2 | GUE: Phage fragments, COVID variants |
| Coding/Non-coding | Sequence type | 1 | GB Coding/Non-coding |
| Chromatin Accessibility | DNase-seq | 1 | iDHS DNase-I |

*Source prefixes:* NT = Nucleotide Transformer; NT-rev = Nucleotide Transformer revised; GUE = Genome Understanding Evaluation; GUE EMP = GUE empirically-validated subset; GB = Genomic Benchmarks; iPro = iPro-WAEL promoter dataset; PGB = Plant Genomics Benchmark; iDNA = iDNA-ABF; iDHS = iDHS-EL; 4mC = deep4mc.

**Epigenomic and Chromatin Tasks.** The largest task category, Histone Modifications ($n = 30$), encompasses prediction of post-translational modifications across distinct histone marks including H3K4me1/2/3, H3K27ac, H3K36me3, H3K9ac, and H4 acetylation states. The tasks draw from three sources: **NT** (10 tasks: H3, H4, H3K4me1/2/3, H3K9ac, H3K14ac, H3K36me3, H3K79me3, H4ac), **NT-rev** (10 tasks: revised versions of H3K4me1/2/3, H3K9ac, and H3K36me3 plus newly added H2AFZ, H3K27ac, H3K27me3, H3K9me3, and H4K20me1), and **GUE** (10 tasks: empirically derived subsets of the NT histone marks). Chromatin Accessibility ($n = 1$) is the **iDHS-EL DNase-I** task, providing a complementary measure of chromatin state.

**Regulatory Element Tasks.** Promoter recognition ($n = 22$) draws from six sources: **NT** and **NT-rev** (6 tasks total: TATA-containing, TATA-less, and combined human promoter classification), **GUE** (6 tasks: core and 300-bp promoter variants with TATA-containing, TATA-less, and combined splits), **GB** (1 task: human non-TATA promoters), **iPro-WAEL** (8 tasks spanning bacterial *B. amyloliquefaciens* and *R. capsulatus*, plant *Arabidopsis* TATA/non-TATA, and human cell-type-specific lines GM12878, HUVEC, HeLa-S3, NHEK), and **PGB** (1 task: *M. esculenta* transcriptional activity). Enhancer

prediction ($n = 8$) draws from **NT** and **NT-rev** (4 tasks: combined and type-stratified enhancer classification) and **GB** (4 tasks: *Drosophila* Stark, mouse enhancers (Ensembl), and human enhancers (Cohn and Ensembl)). General regulatory element tasks ($n = 2$) are both from **GB**: human Ensembl regulatory region prediction and human open chromatin region prediction.

**Transcription Factor and Splicing Tasks.** TF Binding prediction ($n = 5$) comprises five **GUE** human transcription factor binding tasks. Splice Site detection ($n = 7$) combines **NT** (3 tasks: donors, acceptors, combined), **NT-rev** (3 revised counterparts), and **GUE** (1 reconstructed splice site task, an artificially difficult reconstruction of splice junctions).

**Sequence Modification Tasks.** DNA Methylation prediction ($n = 8$) draws from **deep4mc** (6 tasks: 4mC detection in *A. thaliana*, *C. elegans*, *D. melanogaster*, *E. coli*, *G. pickeringii*, *G. subterraneus*) and **iDNA-ABF** (2 tasks: 5mC and 6mA modification site prediction).

**Non-coding RNA and Species Classification Tasks.** Long non-coding RNA classification ($n = 6$) comprises the six **PGB** plant lncRNA tasks: soybean (*G. max*), cassava (*M. esculenta*), sorghum (*S. bicolor*), tomato (*S. lycopersicum*), wheat (*T. aestivum*), and maize (*Z. mays*). Species Classification ($n = 3$) draws from **GB** (human-versus-worm discrimination) and **GUE** (20-way fungal species and 40-way viral species classification).

**Additional Tasks.** Mouse Enhancer prediction ($n = 5$) comprises five **GUE** mouse enhancer tasks (tissue-specific enhancer classification). Virus/Phage detection ($n = 2$) comprises two **GUE** tasks: phage fragment identification and COVID-19 variant classification. Coding/Non-coding discrimination ($n = 1$) is the **GB** coding-vs-intergenomic-sequences task. This diverse task composition enables evaluation of transfer learning across taxonomic boundaries (human to plant, eukaryotic to prokaryotic), assessment of specialization versus generalization trade-offs, and identification of task-specific model advantages. The predominance of eukaryotic tasks (12 of 13 categories) reflects current genomic research priorities while creating systematic disadvantages for prokaryotic-focused models, a limitation we address in our analysis.

## C. Model Summary

Table 4 summarizes the 40 genomic foundation models evaluated in GENEB. For each model we report its canonical name, architecture family, tokenization scheme, parameter count, and pretraining-corpus type.

*Table 4.* **Summary of the 40 genomic foundation models evaluated in GENEB.** Sorted by parameter count (descending). Tokenization labels: SN = single-nucleotide; BPE = byte-pair encoding; k-mer = fixed-length $k$-mer; mixed = mixed nucleotide/amino-acid vocabulary; BioToken = model-specific tokenizer. Architecture labels: T-enc = Transformer-encoder; T-dec = Transformer-decoder.

| Model | Architecture | Tokenization | Params | Pretraining data |
|---|---|---|---|---|
| METAGENE-1 | T-dec | BPE | 7B | multi-species |
| Evo-1-131k | StripedHyena | SN | 7B | prokaryotic |
| GenomeOcean-4B | T-dec | BPE | 4B | multi-species |
| GENERator-Eukaryote-3B | T-dec | k-mer | 3B | eukaryotic-genes |
| DNA-GPT-3B-M | T-dec | k-mer | 3B | multi-species |
| NT-2.5B-MS | T-enc | k-mer | 2.5B | multi-species |
| LucaOne | T-enc | mixed | 2B | multi-species |
| GENERator-Eukaryote-1.2B | T-dec | k-mer | 1.2B | eukaryotic-genes |
| Omni-DNA-1B | T-dec | BPE | 1B | multi-species |
| Agro-NT-1B | T-enc | k-mer | 1B | plant-genomes |
| eccDNAMamba | Mamba-SSM | BPE | 1B | multi-species |
| SPACE | CNN-Transformer-MoE | SN | 589M | human-mouse-profiles |
| GenomeOcean-500M | T-dec | BPE | 500M | multi-species |
| GENA-LM-Large-T2T | T-enc | BPE | 336M | multi-species |
| Omni-DNA-300M | T-dec | BPE | 300M | multi-species |
| BioFM-265M | T-dec | BioToken | 265M | human |
| Enformer | CNN-Transformer | SN | 250M | human-mouse-profiles |
| NT-v2-250M-MS | T-enc | k-mer | 250M | multi-species |
| PlantCaduceus | Mamba-SSM | SN | 225M | plant-genomes |
| OmniNA-220M | T-dec | BPE | 220M | multi-species |
| GPT2-Gene-Multi-v2 | T-dec | k-mer | 200M | multi-species |
| GPT2-Gene-v1 | T-dec | k-mer | 200M | human |
| Genomics-FM | T-enc | BPE+k-mer | 120M | multi-species |
| DNABERT-S | T-enc | k-mer | 117M | multi-species-microbial |
| DNABERT-2 | T-enc | BPE | 117M | multi-species |
| GENA-LM-T2T-Multi | T-enc | BPE | 110M | multi-species |
| GENA-LM | T-enc | BPE | 110M | human |
| DNA-GPT-0.1B-H | T-dec | k-mer | 100M | multi-species |
| NT-v2-100M-MS | T-enc | k-mer | 100M | multi-species |
| GROVER | T-enc | BPE | 87M | human |
| MutBERT | T-enc | SN | 86M | human |
| DeepGene | Graph-Transformer | BPE | 85M | human |
| HyenaDNA-Large-1M | Hyena | SN | 55M | human |
| NT-v2-50M-3mer-MS | T-enc | k-mer | 50M | multi-species |
| NT-v2-50M-MS | T-enc | k-mer | 50M | multi-species |
| HyenaDNA-Medium-160k | Hyena | SN | 14M | human |
| Caduceus-PS-131k | Mamba-SSM | SN | 8M | human |
| JanusDNA-72-w | Hybrid-Mamba-MoE | SN | 2M | human |
| JanusDNA-72-wo | Hybrid-Mamba-MoE | SN | 2M | human |
| Caduceus-PH-1k | Mamba-SSM | SN | 2M | human |

# D. Models Excluded from Evaluation

During our comprehensive survey of genomic foundation models, we identified several promising models that could not be included in our benchmark evaluation due to various technical and practical limitations. We document these exclusions for transparency and to inform future benchmark efforts. Table 5 summarizes the excluded models and their exclusion reasons.

*Table 5.* **Genomic foundation models excluded from GENEB evaluation.** Models are grouped by exclusion category. All models were considered for inclusion but could not be evaluated due to the stated limitations.

| Model | Category | Exclusion Reason |
|---|---|---|
| *Unavailable or Private Weights* | | |
| Gene42 (Vishniakov et al., 2025a) | Private weights | Model weights not publicly released |
| NTv3 (Boshar et al., 2025) | Private weights | Model weights not publicly released despite Hugging-Face placeholder |
| *Code/Infrastructure Issues* | | |
| NucleotideGPT (Mclaughlin et al., 2024) | Broken code | Corrupted weights and buggy code; TPU-only with no GPU support documentation |
| EpiGePT (Gao et al., 2023) | Missing extraction code | No embedding extraction interface; requires extensive custom implementation |
| ENBED (Malusare et al., 2024) | Missing extraction code | No embedding extraction code; unclear model checkpoint location |
| HAD (Yang et al., 2025a) | No code | No public code repository available |
| C.La.P. (Nisantzis et al., 2025) | No code | No public code repository available |
| HybriDNA (Ma et al., 2025) | No code | No public code repository available |
| MxDNA (Qiao et al., 2024) | Broken weights | Incompatible or corrupted model weights |
| VQDNA (Li et al., 2024) | No code | No public code repository available |
| BMFM-DNA (Li et al., 2025a) | Dependency conflicts | Library version conflicts prevent execution |
| *Computational Constraints* | | |
| Evo2 (Brixi et al., 2025a) | Hardware requirements | Requires H100/H200 GPUs exceeding our computational budget |
| *Architectural Limitations* | | |
| ChatNT (Richard et al., 2024) | Wrapper model | Uses Nucleotide Transformer as encoder; not an independent foundation model |

## D.1. Detailed Exclusion Notes

**Private or Unavailable Weights.** Two models – Gene42 and NTv3 – were excluded because their pretrained weights have not been publicly released. Gene42 (Vishniakov et al., 2025a) introduces a long-range genomic foundation model with dense attention capable of processing sequences up to 192 kbp, representing a significant architectural advancement. NTv3 (Boshar et al., 2025) extends the Nucleotide Transformer family with joint sequence-function modeling. Both models report strong benchmark results in their respective publications, but the lack of public weights prevents independent evaluation.

**Missing or Broken Code.** Several models suffer from incomplete or non-functional code releases. NucleotideGPT (Mclaughlin et al., 2024) provides weights that appear corrupted and code that only supports TPU execution without GPU alternatives. EpiGePT (Gao et al., 2023) and ENBED (Malusare et al., 2024) lack embedding extraction interfaces, making it infeasible to obtain sequence representations for downstream evaluation without substantial reverse-engineering effort. HAD (Yang et al., 2025a), C.La.P. (Nisantzis et al., 2025), and HybriDNA (Ma et al., 2025) have no public code repositories despite published papers describing their architectures.

**Runtime and Dependency Issues.** MxDNA (Qiao et al., 2024) provides code and weights but fail to execute correctly due to weight incompatibilities or undocumented runtime requirements. BMFM-DNA (Li et al., 2025a) from IBM Research encounters library version conflicts that prevent successful model loading.

**Computational Requirements.** Evo2 (Brixi et al., 2025a), the successor to the original Evo model included in our benchmark, requires H100 or H200 GPUs for inference due to its 40B parameter scale. This exceeds our available computational resources (A100 GPUs) and represents a practical barrier for many research groups.

**Wrapper Models.** ChatNT (Richard et al., 2024) was excluded because it uses a frozen Nucleotide Transformer v2 as its DNA encoder, making it a wrapper rather than an independent foundation model. Evaluating ChatNT would redundantly measure NTv2 performance while introducing confounding factors from the conversational interface.

## D.2. Implications for Reproducibility

The prevalence of excluded models (13 out of 53 initially surveyed, approximately 25%) highlights a reproducibility challenge in genomic foundation model research. We encourage future model releases to include: (i) publicly available pretrained weights, (ii) documented embedding extraction code, (iii) clear hardware requirements, and (iv) tested installation procedures across common environments.

## D.3. Excluded Long-Range Regulatory Tasks

GENEB does not include tasks that require explicit modeling of very long-range regulatory interactions ($> 10\,\text{kb}$). We considered the following candidate datasets but excluded them on the grounds outlined below:

**Enhancer–promoter interaction prediction.** Tasks based on predicting physical or functional contacts between enhancers and their target promoters at distances of 50–500 kb (e.g., ChIA-PET, BENGI, HiChIP-derived datasets). Most genomic foundation models in GENEB accept context windows below 6 kb, making fair evaluation impossible without arbitrary cropping.

**Three-dimensional chromatin contact map prediction.** Tasks based on predicting Hi-C contact frequencies or topologically associating domain (TAD) boundaries from megabase-scale sequence windows. These require context lengths that exceed the input limits of all but a few of the evaluated models.

**Distal eQTL effect prediction.** Tasks linking sequence variants to gene expression changes when the variant lies $> 100\,\text{kb}$ from the affected gene. Direct sequence-to-effect formulations require long context; reduced formulations introduce confounds that defeat the purpose of fair cross-model comparison.

**Whole-locus expression quantification.** Tasks predicting tissue-specific expression from $> 10\,\text{kb}$ genomic windows around a gene of interest, including locus-level enhancer collections.

**Models affected by these exclusions.** Models with explicit long-context capability – specifically HYENADNA-LARGE-1M (1M tokens), CADUCEUS-PS-131K (131k tokens), EVO-1-131K (131k tokens), and JANUSDNA-72-W/JANUSDNA-72-WO (Hybrid-Mamba-MoE architecture) – are not exercised on the regime where their architectural priors would most plausibly yield differentiating gains. We treat this as a known limitation of the current benchmark snapshot; extending GENEB with long-range regulatory tasks under a unified protocol is an important direction for future work.

# E. Probe Stability and Protocol Sensitivity Analysis

GENEB evaluates frozen representations with linear probing, which provides a controlled and interpretable measure of representation quality but in principle could obscure information accessible only via non-linear readouts. To verify that the rankings and conclusions reported in the main paper are not artifacts of this choice, we conduct two complementary stability analyses: (i) replacing the linear probe with a non-linear MLP probe (Section E.1), and (ii) varying the regularization strength of the linear probe across few-shot regimes (Section E.2). Both analyses are performed on a representative subset of 11 models and 13 tasks, where each task is drawn from a distinct functional category to ensure coverage of the benchmark's diversity. The selected models span the full range of architectures, tokenization schemes, pretraining corpora, and parameter scales evaluated in GENEB; the model and task subsets are summarized in Tables 6 and 7.

*Table 6.* Representative model subset used for probe stability and protocol sensitivity analyses. The subset spans all major architectural families, tokenization schemes, pretraining corpora, and three orders of magnitude in parameter scale.

| Model | Params | Architecture | Tokenization | Pretraining |
|---|---|---|---|---|
| GENERATOR-EUKARYOTE-3B | 3B | Transformer-decoder | $k$-mer | Eukaryotic genes |
| LUCAONE | 2B | Transformer-encoder | Other (custom) | Multi-species |
| GENERATOR-EUKARYOTE-1.2B | 1.2B | Transformer-decoder | $k$-mer | Eukaryotic genes |
| GENOMEOCEAN-4B | 4B | Transformer-decoder | BPE | Multi-species |
| OMNI-DNA-1B | 1B | Transformer-decoder | BPE | Multi-species |
| GENOMEOCEAN-500M | 500M | Transformer-decoder | BPE | Multi-species |
| GENA-LM-LARGE-T2T | 336M | Transformer-encoder | BPE | Multi-species |
| GROVER | 87M | Transformer-encoder | BPE | Human |
| MUTBERT | 86M | Transformer-encoder | Single-nucleotide | Human |
| HYENADNA-LARGE-1M | 55M | Hyena | Single-nucleotide | Human |
| NT-V2-50M-MS | 50M | Transformer-encoder | $k$-mer | Multi-species |

*Table 7.* Representative task subset used for probe stability and protocol sensitivity analyses. One task is selected from each of the 13 functional categories in GENEB.

| Category | Task |
|---|---|
| Histone Modifications | GUE EMP H3K4me1 |
| Promoters | iPro GM12878 |
| Enhancers | GB Human enh. (Cohn) |
| DNA Methylation | 4mC *E. coli* |
| Splice Sites | NT Splice (all) |
| lncRNA | PGB lncRNA *Z. mays* |
| Mouse Enhancers | GUE Mouse enh.-0 |
| TF Binding | GUE TF-0 |
| Species Classification | GUE Virus-40 |
| Regulatory | GB Ensembl regulatory |
| Virus/Phage | GUE Phage fragments |
| Coding/Non-coding | GB Coding/Non-coding |
| Chromatin Accessibility | iDHS DNase-I |

## E.1. Probe Stability Analysis

**Setup.** For each of the $11 \times 13 = 143$ model–task combinations, we compute two MCC values: one using the linear probe (logistic regression) employed throughout the main paper, and one using a non-linear MLP probe consisting of a single hidden layer of 256 units with ReLU activation, trained with early stopping. All other components of the evaluation pipeline – feature extraction, normalization, train/test splits, and random seeds – are held identical across the two probes. We then compare the resulting rankings and absolute MCC values to assess whether the linear probe provides a faithful proxy for representation quality under non-linear readouts.

**Rankings are highly stable across probes.** Spearman rank correlation between the two probes is $\rho = 0.964$ ($p < 0.001$) across all 143 model–task pairs, and $\rho = 0.973$ ($p < 0.001$) when computed on per-model average MCC. The top-3 and top-5 models, ranked by mean MCC across the 13 tasks, are identical under both probes (Table 8). Per-task Spearman correlations are positive for 12 of 13 tasks (Table 9), with a median of $\rho = 0.855$. The sole exception is GB ENSEMBL

REGULATORY, where MCC values are tightly clustered across models, leaving rank ordering dominated by stochastic noise rather than substantive differences in representation quality.

*Table 8.* Per-model average MCC under linear and MLP probes across the 13 representative tasks. Models are ordered by linear-probe MCC. The top-5 positions are identical under both probes. The largest absolute difference (HYENADNA-LARGE-1M, +0.052) does not alter the model's relative rank.

| Model | Linear MCC | MLP MCC | $\Delta$ |
|---|---|---|---|
| GENERATOR-EUKARYOTE-3B | 0.605 | 0.609 | +0.004 |
| GENERATOR-EUKARYOTE-1.2B | 0.579 | 0.595 | +0.016 |
| LUCAONE | 0.573 | 0.600 | +0.027 |
| GENOMEOCEAN-4B | 0.552 | 0.552 | +0.000 |
| OMNI-DNA-1B | 0.550 | 0.542 | −0.009 |
| GENOMEOCEAN-500M | 0.536 | 0.535 | −0.001 |
| GENA-LM-LARGE-T2T | 0.530 | 0.535 | +0.005 |
| MUTBERT | 0.516 | 0.517 | +0.001 |
| NT-v2-50M-MS | 0.511 | 0.521 | +0.010 |
| GROVER | 0.466 | 0.477 | +0.011 |
| HYENADNA-LARGE-1M | 0.427 | 0.479 | +0.052 |

*Table 9.* Per-task Spearman rank correlation between linear and MLP probe rankings across the 11 representative models. Tasks are ordered by correlation strength. Twelve of thirteen tasks exhibit strongly positive rank correlation. The single negative value (GB ENSEMBL REGULATORY) corresponds to a task where MCC values are tightly clustered across models, leaving rank ordering dominated by noise rather than substantive performance differences.

| Task | $\rho$ |
|---|---|
| GB Coding/Non-coding | +0.982 |
| GUE Phage fragments | +0.982 |
| PGB lncRNA *Z. mays* | +0.973 |
| GUE Virus-40 | +0.945 |
| GUE EMP H3K4me1 | +0.891 |
| GUE Mouse enh.-0 | +0.891 |
| GB Human enh. (Cohn) | +0.855 |
| NT Splice (all) | +0.818 |
| GUE TF-0 | +0.809 |
| iDHS DNase-I | +0.743 |
| 4mC *E. coli* | +0.673 |
| iPro GM12878 | +0.391 |
| GB Ensembl regulatory | −0.082 |

**Absolute MCC shifts are small.** The mean signed difference between linear and MLP probe MCC across the 11 models is +0.011, indicating only a marginal aggregate benefit from non-linear readouts. The largest single shift is observed for HYENADNA-LARGE-1M (+0.052), which is consistent with the hypothesis that Hyena representations benefit modestly from non-linear projection. However, this shift does not change the model's relative rank, and the overall ordering remains intact.

**Conclusion.** These results indicate that the rankings reported in the main paper are robust to probe choice within the representative subset evaluated here: model orderings under linear probing constitute a reliable proxy for those obtained with non-linear MLP probes. The main empirical conclusions of GENEB – including the scale–performance disconnect, architectural dominance under matched conditions, and category-dependent specialization – are therefore unlikely to be artifacts of the linear probing protocol. We note that the present analysis addresses the stability of relative comparisons under frozen representations; the question of whether rankings remain stable under full task-specific fine-tuning is discussed in the main paper as a limitation.

### E.2. Few-Shot Protocol Sensitivity

**Setup.** A second potential source of artifact in our few-shot conclusions is the choice of regularization strength in the linear probe, particularly in the low-data 1-shot and 10-shot regimes where logistic regression behavior can be sensitive

to hyperparameter settings. To assess this, we sweep the inverse regularization strength $C \in \{0.01, 0.1, 1.0, 10.0, 100.0\}$ across all three shot regimes (1-shot, 10-shot, and full-data) for the same $11 \times 13$ model–task subset used in Section E.1. We sweep $C$ as the principal regularization hyperparameter of logistic regression; other settings (maximum iterations, solver, convergence tolerance) are held identical to the main-paper protocol, as are feature normalization, train/test splits, and random seeds. We then quantify both ranking stability and absolute MCC sensitivity as functions of $C$.

**Rankings are essentially invariant in the 1-shot regime.** Pairwise Spearman correlations between rankings at different $C$ values are summarized in Table 10. In the 1-shot regime, mean pairwise $\rho = 0.993$, with a minimum of 0.982 across all $\binom{5}{2} = 10$ pairs of $C$ values (Table 11). This near-invariance indicates that rankings in the most data-constrained regime are dominated by intrinsic representation quality rather than by regularization choice.

*Table 10.* Ranking stability across regularization strengths $C \in \{0.01, 0.1, 1, 10, 100\}$. Each row reports the mean, minimum, and maximum pairwise Spearman correlation between model rankings under different values of $C$ within a given shot regime. Rankings are highly stable in the 1-shot regime and remain stable for adjacent values of $C$ in the 10-shot and full-data regimes.

| Regime | Mean pairwise $\rho$ | Min $\rho$ | Max $\rho$ |
|---|---|---|---|
| 1-shot | 0.993 | 0.982 | 1.000 |
| 10-shot | 0.805 | 0.582 | 0.982 |
| Full-data | 0.766 | 0.436 | 0.955 |

*Table 11.* Pairwise Spearman correlation between model rankings at different regularization strengths in the 1-shot regime. Rankings are nearly invariant ($\rho \geq 0.982$) across all $C$ pairs.

| | $C = 0.01$ | $C = 0.1$ | $C = 1$ | $C = 10$ | $C = 100$ |
|---|---|---|---|---|---|
| $C = 0.01$ | – | 0.982 | 0.982 | 0.982 | 0.982 |
| $C = 0.1$ | | – | 1.000 | 1.000 | 1.000 |
| $C = 1$ | | | – | 1.000 | 1.000 |
| $C = 10$ | | | | – | 1.000 |
| $C = 100$ | | | | | – |

**Rankings remain stable for adjacent $C$ values in the 10-shot regime.** At 10-shot, mean pairwise $\rho = 0.805$, with adjacent $C$ values yielding $\rho \geq 0.9$ (Table 12). Larger divergences appear only between extreme settings (e.g., $C = 0.01$ vs. $C = 100$, $\rho = 0.582$), reflecting the increased influence of regularization when small but non-trivial amounts of supervision are available. The full-data regime exhibits a similar pattern (mean $\rho = 0.766$; min $\rho = 0.436$), indicating that ranking fluctuations are driven primarily by extreme regularization choices rather than by typical hyperparameter selections.

*Table 12.* Pairwise Spearman correlation between model rankings at different regularization strengths in the 10-shot regime. Adjacent $C$ values yield $\rho \geq 0.9$, with substantial divergence appearing only between extreme settings.

| | $C = 0.01$ | $C = 0.1$ | $C = 1$ | $C = 10$ | $C = 100$ |
|---|---|---|---|---|---|
| $C = 0.01$ | – | 0.927 | 0.809 | 0.636 | 0.582 |
| $C = 0.1$ | | – | 0.900 | 0.727 | 0.664 |
| $C = 1$ | | | – | 0.936 | 0.891 |
| $C = 10$ | | | | – | 0.982 |
| $C = 100$ | | | | | – |

**Absolute MCC sensitivity is bounded and concentrated in a small subset of models.** Per-model MCC ranges across $C$ values in the full-data regime are reported in Table 13. For most models, the range is well below 0.10 MCC, with the smallest sensitivity observed for GENOMEOCEAN-500M (range 0.014). The largest sensitivities are observed for LUCAONE (0.199), HYENADNA-LARGE-1M (0.144), and NT-V2-50M-MS (0.108), suggesting that representations from these models are somewhat more dependent on regularization tuning than those of the remaining models. We note that absolute MCC sensitivity does not translate into ranking instability for typical regularization choices: across adjacent $C$ values, the relative ordering of models remains substantively unchanged (Tables 11 and 12).

*Table 13.* Per-model MCC range across regularization strengths $C \in \{0.01, 0.1, 1, 10, 100\}$ in the full-data regime. The largest sensitivity is observed for LucaOne, HyenaDNA-Large-1M, and NT-v2-50M-MS; the smallest for GenomeOcean-500M.

| Model | Min MCC | Max MCC | Range |
|---|---|---|---|
| GenomeOcean-500M | 0.523 | 0.537 | 0.014 |
| GROVER | 0.457 | 0.473 | 0.017 |
| GENERator-Eukaryote-1.2B | 0.563 | 0.583 | 0.021 |
| GENERator-Eukaryote-3B | 0.572 | 0.603 | 0.031 |
| GenomeOcean-4B | 0.517 | 0.557 | 0.040 |
| GENA-LM-Large-T2T | 0.477 | 0.529 | 0.053 |
| Omni-DNA-1B | 0.483 | 0.550 | 0.067 |
| MutBERT | 0.452 | 0.527 | 0.076 |
| NT-v2-50M-MS | 0.404 | 0.512 | 0.108 |
| HyenaDNA-Large-1M | 0.318 | 0.462 | 0.144 |
| LucaOne | 0.415 | 0.614 | 0.199 |

**The principal few-shot finding is protocol-stable.** Crucially, the central few-shot conclusion reported in the main paper – the sharp degradation of mean MCC from full-data to 1-shot – is replicated at every value of $C$ tested. The magnitude of this degradation varies modestly across regularization choices, and its direction and severity are preserved without exception.

**Conclusion.** These results indicate that the few-shot findings reported in the main paper reflect properties of the evaluated representations under data-constrained regimes, rather than artifacts of a particular regularization choice. Rankings are essentially invariant under 1-shot evaluation, remain stable for typical regularization settings under 10-shot and full-data evaluation, and the qualitative pattern of severe low-data degradation persists uniformly across the regularization sweep.

### E.3. Controlled-Pair Comparisons and Residual Confounds

**Methodology.** Throughout the main paper, comparative claims about architecture, tokenization, and pretraining data are based on *matched pairs* of models that differ in exactly one factor of interest while holding others as constant as the available model set permits. This controlled-pair design substantially reduces the risk of attributing observed performance differences to the wrong cause and is preferable to unmatched comparisons across the full benchmark. However, perfect isolation is impossible in practice: genomic foundation models differ along multiple correlated axes (architecture, tokenization, training corpus, scale, training duration, and pretraining objective), so even carefully matched pairs retain residual confounds. To make these confounds explicit and to enable readers to assess the strength of each claim, we enumerate the full set of 30 controlled pairs underlying the comparative analyses.

*Table 14.* Summary of the 30 controlled-pair comparisons used in the analyses of architecture, pretraining data, and tokenization. Each pair varies a single factor of interest while holding the others constant.

| Comparison Type | Pairs |
|---|---|
| **A. Architecture** | 9 |
| Decoder vs. Encoder | 6 |
| Decoder vs. Mamba | 1 |
| Encoder vs. Graph-Transformer | 2 |
| **B. Pretraining Data** | 9 |
| Human vs. Multi-species | 6 |
| Multi-species vs. Microbial | 2 |
| Eukaryotic-genes vs. Multi-species | 1 |
| **C. Tokenization** | 12 |
| BPE vs. $k$-mer | 8 |
| Single-nucleotide vs. BPE | 2 |
| Other | 2 |
| **Total** | **30** |

**Summary of comparisons.** Table 14 provides an overview of the matched pairs by factor type. Architecture comparisons (9 pairs) hold tokenization and pretraining corpus type constant while varying the architectural family. Pretraining-data

comparisons (9 pairs) hold architecture and tokenization constant while varying the corpus type. Tokenization comparisons (12 pairs) hold architecture and pretraining corpus constant while varying the tokenization scheme.

**Architecture comparisons.** Table 15 enumerates the 9 matched pairs used to isolate architectural effects. In each pair, tokenization and pretraining corpus type are held constant; the varied factor is architectural family. For each pair we report the macro-averaged MCC difference $\Delta = \text{MCC}_A - \text{MCC}_B$; positive values indicate that the architectural family listed first in the "Variable" column wins. Residual confounds for this group of comparisons include model size differences (constrained to within a factor of two where possible), training duration, exact composition of the pretraining data, pretraining objective (masked language modeling vs. causal language modeling), and depth-to-width ratio.

*Table 15.* Architecture-controlled pairs. Each pair holds tokenization and pretraining corpus type constant and varies the architectural family. $\Delta$ macro-MCC is the difference between model A and model B; positive values indicate that the architectural family listed first in the "Variable" column wins.

| Model A | Model B | Controlled | Variable | $\Delta$ |
|---|---|---|---|---|
| Omni-DNA-1B | eccDNAMamba | BPE, multi-species | Decoder vs. Mamba | +0.149 |
| GenomeOcean-500M | GENA-LM-Large-T2T | BPE, multi-species | Decoder vs. Encoder | −0.002 |
| Omni-DNA-300M | GENA-LM-Large-T2T | BPE, multi-species | Decoder vs. Encoder | −0.014 |
| DNA-GPT-3B-M | NT-2.5B-MS | $k$-mer, multi-species | Decoder vs. Encoder | −0.031 |
| DNA-GPT-0.1B-H | NT-v2-100M-MS | $k$-mer, multi-species | Decoder vs. Encoder | −0.040 |
| DNA-GPT-0.1B-H | Genomics-FM | $k$-mer, multi-species | Decoder vs. Encoder | −0.047 |
| GPT2-Gene-Multi-v2 | NT-v2-250M-MS | $k$-mer, multi-species | Decoder vs. Encoder | −0.070 |
| GROVER | DeepGene | BPE, human | Encoder vs. Graph-Trans. | +0.061 |
| DeepGene | GENA-LM | BPE, human | Graph-Trans. vs. Encoder | −0.066 |

**Pretraining-data comparisons.** Table 16 enumerates the 9 matched pairs used to isolate the effect of pretraining corpus type. In each pair, architecture and tokenization are held constant; the varied factor is the taxonomic composition of the pretraining data. For each pair we report the macro-averaged MCC difference $\Delta = \text{MCC}_A - \text{MCC}_B$; positive values indicate that the corpus type listed first in the "Variable" column wins. Residual confounds include exact corpus scale and diversity, training duration, and learning-rate schedules. We additionally note that the *Eukaryotic-genes vs. Multi-species* comparison is supported by a single matched pair (GENERATOR-EUKARYOTE-3B vs. DNA-GPT-3B-M); claims derived from this comparison are accordingly flagged in the main text and should be interpreted with additional caution.

*Table 16.* Pretraining-data-controlled pairs. Each pair holds architecture and tokenization constant and varies the pretraining corpus type. $\Delta$ macro-MCC is the difference between model A and model B; positive values indicate the corpus type listed first in "Variable" wins.

| Model A | Model B | Controlled | Variable | $\Delta$ |
|---|---|---|---|---|
| Genomics-FM | DNABERT-S | Encoder, $k$-mer | Multi vs. Microbial | +0.088 |
| NT-v2-100M-MS | DNABERT-S | Encoder, $k$-mer | Multi vs. Microbial | +0.081 |
| GENERATOR-EUKARYOTE-3B | DNA-GPT-3B-M | Decoder, $k$-mer | Eukaryotic vs. Multi | +0.063 |
| DNABERT-2 | GENA-LM | Encoder, BPE | Multi vs. Human | +0.014 |
| GENA-LM-T2T-Multi | GENA-LM | Encoder, BPE | Multi vs. Human | +0.025 |
| DNABERT-2 | GROVER | Encoder, BPE | Multi vs. Human | +0.019 |
| GENA-LM-T2T-Multi | GROVER | Encoder, BPE | Multi vs. Human | +0.030 |
| GPT2-Gene-Multi-v2 | GPT2-Gene-v1 | Decoder, $k$-mer | Multi vs. Human | −0.009 |
| DNA-GPT-0.1B-H | GPT2-Gene-v1 | Decoder, $k$-mer | Multi vs. Human | −0.007 |

**Tokenization comparisons.** Table 17 enumerates the 12 matched pairs used to isolate the effect of tokenization scheme. In each pair, architecture and pretraining corpus type are held constant; the varied factor is the tokenization strategy. For each pair we report the macro-averaged MCC difference $\Delta = \text{MCC}_A - \text{MCC}_B$, where positive values indicate that the tokenization listed first in the "Variable" column wins. Residual confounds include model size differences (within $\pm 2\times$), vocabulary size (tightly coupled with tokenization scheme), and model-specific training regimes.

Aggregating across the 12 pairs reveals three regimes. (i) In matched Transformer-decoder comparisons with multi-species pretraining, BPE exceeds $k$-mer on average (+0.020 across 3 pairs), with a large positive gap for one pair (+0.071) and one reversal (−0.042). (ii) In matched Transformer-encoder comparisons with multi-species pretraining, BPE and $k$-mer perform

*Table 17.* Tokenization-controlled pairs. Each pair holds architecture and pretraining corpus type constant and varies the tokenization scheme. $\Delta$ macro-MCC is the difference between model A and model B; positive values indicate that the tokenization listed first in the "Variable" column wins.

| Model A | Model B | Controlled | Variable | $\Delta$ |
|---|---|---|---|---|
| OMNI-DNA-300M | GPT2-GENE-MULTI-V2 | Decoder, multi-species | BPE vs. $k$-mer | +0.071 |
| GENOMEOCEAN-4B | DNA-GPT-3B-M | Decoder, multi-species | BPE vs. $k$-mer | +0.032 |
| OMNINA-220M | GPT2-GENE-MULTI-V2 | Decoder, multi-species | BPE vs. $k$-mer | −0.042 |
| GENA-LM-LARGE-T2T | NT-V2-250M-MS | Encoder, multi-species | BPE vs. $k$-mer | +0.015 |
| GENA-LM-T2T-MULTI | NT-V2-100M-MS | Encoder, multi-species | BPE vs. $k$-mer | +0.013 |
| GENA-LM-T2T-MULTI | GENOMICS-FM | Encoder, multi-species | BPE vs. $k$-mer | +0.006 |
| DNABERT-2 | NT-V2-100M-MS | Encoder, multi-species | BPE vs. $k$-mer | +0.002 |
| DNABERT-2 | GENOMICS-FM | Encoder, multi-species | BPE vs. $k$-mer | −0.005 |
| MUTBERT | GENA-LM | Encoder, human | Single-nt vs. BPE | +0.033 |
| GROVER | MUTBERT | Encoder, human | BPE vs. Single-nt | −0.038 |
| GPT2-GENE-V1 | BIOFM-265M | Decoder, human | $k$-mer vs. Other | +0.134 |
| LUCAONE | NT-2.5B-MS | Encoder, multi-species | Other vs. $k$-mer | +0.017 |

comparably (+0.006 across 5 pairs, with all gaps within ±0.02 MCC). (iii) In matched Transformer-encoder comparisons with human pretraining, single-nucleotide tokenization (MUTBERT) consistently outperforms BPE baselines (+0.033 and +0.038 over GENA-LM and GROVER, respectively). The two comparisons involving non-standard tokenization schemes (BIOFM-265M's BioToken; LUCAONE's mixed vocabulary) are reported for completeness but should be interpreted with caution, as these schemes introduce confounds beyond a simple choice of tokenization unit.

**Conclusion.** Throughout the main paper, observations derived from these matched pairs are framed as *consistent with* or *associated with* the varied factor, rather than as *caused by* it. Single-pair comparisons – most notably the *Eukaryotic-genes vs. Multi-species* contrast – are flagged as such on first appearance, and conclusions relying on them are stated with correspondingly reduced confidence. We view this controlled-pair methodology as a principled middle ground between unmatched comparisons across the full benchmark, which conflate multiple factors, and fully randomized causal experiments, which are infeasible at the scale of foundation-model pretraining.

### E.4. Micro- vs. Macro-Averaged Aggregate Performance

The main paper reports macro-averaged MCC as the principal aggregation metric, computed by first averaging within each of the 13 functional categories and then averaging across categories with equal weight per category. This choice avoids implicitly overweighting categories with many tasks (e.g., histone modifications with 30 tasks, promoters with 22 tasks) and instead treats each functional category as a unit of biological interest. To verify that the central findings of GENEB are robust to this weighting choice, we additionally compute *micro-averaged* MCC by simple averaging across all 100 tasks and compare the two aggregation schemes.

Figure 8 compares the two aggregation schemes across all 40 models. The Spearman rank correlation between micro- and macro-averaged orderings is $\rho = 0.988$ ($p < 0.001$), indicating that the relative ordering of models is largely preserved. The mean absolute shift in aggregate MCC is $|\Delta| = 0.009$, and the top-5 set is identical under both schemes (within the top-5, LUCAONE and GENERATOR-EUKARYOTE-1.2B swap positions). The largest individual shifts are concentrated in out-of-domain or specialized models: EVO-1-131K ($\Delta = -0.044$), CADUCEUS-PS-131K ($\Delta = -0.028$), and PLANTCADUCEUS ($\Delta = -0.024$), all of which exhibit highly uneven category-level performance that is amplified under category-balanced averaging.

The central empirical findings of GENEB – the substantial overall correlation between scale and performance, the instability of category rankings, and the dominance of architectural and pretraining alignment over parameter count – hold under both averaging schemes. We view the sensitivity of out-of-domain models to weighting choice as itself consistent with the broader argument of this paper: single-score leaderboards are an unreliable basis for genomic model selection.

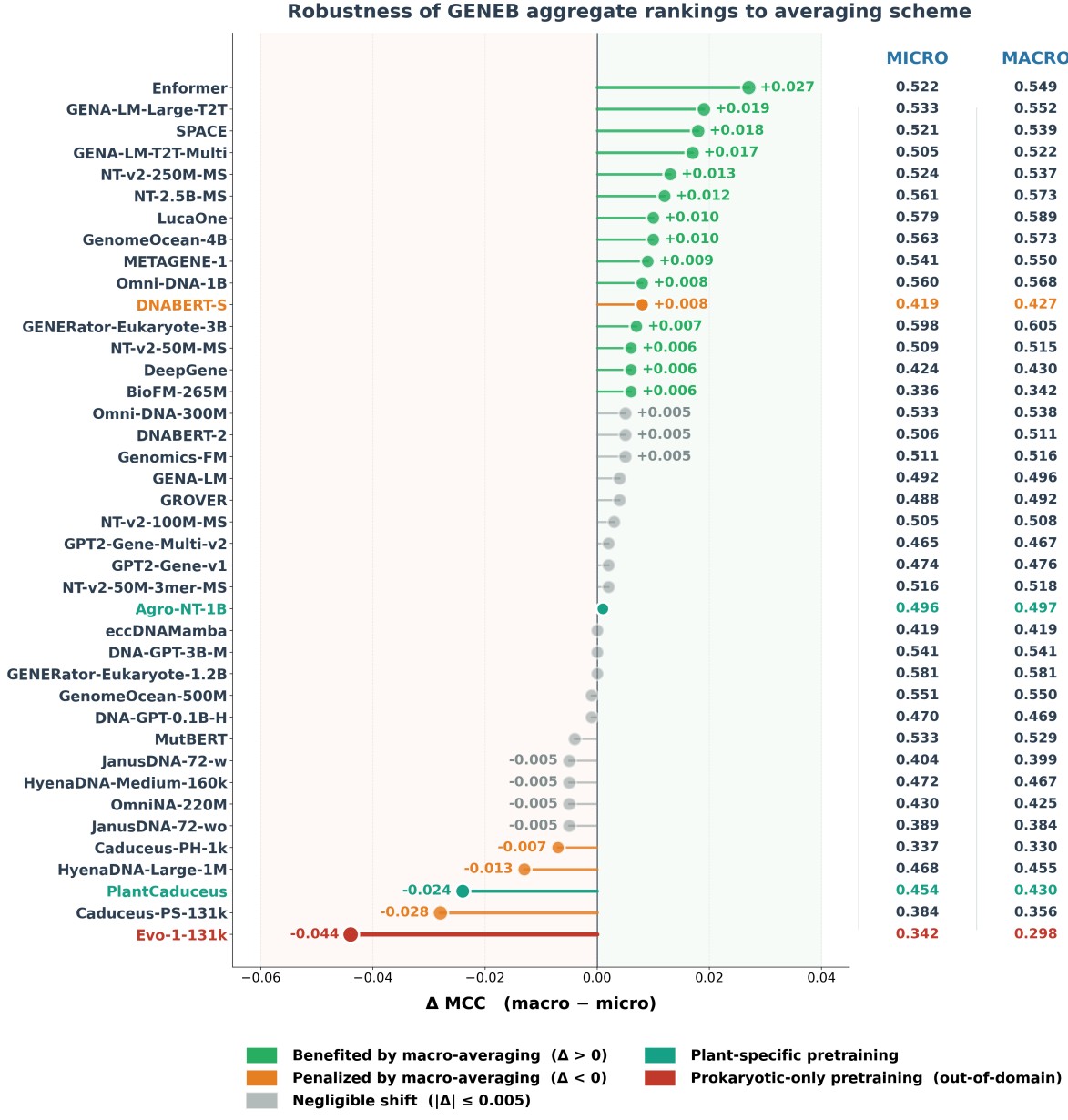

*Figure 8.* **Robustness of GENEB aggregate rankings to averaging scheme.** For each of the 40 models, $\Delta = \text{MCC}_{\text{macro}} - \text{MCC}_{\text{micro}}$ is shown in the left panel; the side panel reports the underlying micro- and macro-averaged MCC values. Models are sorted from largest negative shift to largest positive shift. Out-of-domain models are highlighted: prokaryotic-only EVO-1-131K (red, $\Delta = -0.044$), microbial-only DNABERT-S (orange, $\Delta = +0.008$), and plant-specific PLANTCADUCEUS and AGRO-NT-1B (teal). Across all 40 models the Spearman rank correlation between micro- and macro-averaged orderings is $\rho = 0.988$ ($p < 0.001$); the mean absolute shift is $|\Delta| = 0.009$ MCC.

# F. Results Analysis

This section presents a systematic analysis of the benchmark results across 40 DNA foundation models evaluated on 100 genomic tasks spanning 13 functional categories. We structure our analysis around three fundamental questions: (1) how do architectural choices influence model performance, (2) what role does tokenization strategy play, and (3) how does model scale interact with pretraining design decisions?

## F.1. Experimental Overview

Our benchmark encompasses substantial diversity in both models and tasks. The 40 evaluated models span the following architectural families: Transformer-encoder ($n = 15$), Transformer-decoder ($n = 13$), Mamba-SSM ($n = 4$), Hybrid-Mamba-MoE ($n = 2$), Hyena ($n = 2$), CNN-Transformer hybrids ($n = 2$, including ENFORMER and SPACE), Graph-Transformer ($n = 1$, DEEPGENE), and StripedHyena ($n = 1$, EVO-1-131K). Model sizes range from under 2M to 7B parameters, with tokenization strategies including $k$-mer ($n = 13$), BPE ($n = 13$), single-nucleotide ($n = 11$), and three custom schemes (BIOFM-265M's BioToken, LUCAONE's mixed nucleotide–amino-acid vocabulary, and GENOMICS-FM's ensemble of BPE and $k$-mer). Pretraining corpora vary across multi-species genomes ($n = 20$), human-only ($n = 12$), eukaryotic gene sequences ($n = 2$), plant genomes ($n = 2$), human-mouse epigenomic profiles ($n = 2$), prokaryotic sequences ($n = 1$), and multi-species microbial genomes ($n = 1$).

The 100 downstream tasks are organized into 13 categories reflecting distinct aspects of genomic function: Histone Modifications (30 tasks), Promoters (22), Enhancers (8), DNA Methylation (8), Splice Sites (7), lncRNA (6), Mouse Enhancers (5), TF Binding (5), Species Classification (3), Regulatory (2), Virus/Phage (2), Coding/Non-coding (1), and Chromatin Accessibility (1). The full task taxonomy is given in Appendix B. This distribution enables fine-grained analysis of model capabilities across regulatory, structural, and evolutionary prediction problems.

## F.2. Aggregate Performance Patterns

Before examining specific architectural and design factors, we first characterize overall model performance across the benchmark. Figure 9 provides boxplot distributions of MCC scores within each task category for the top-15 models per category, illustrating that performance variance differs substantially across categories. Figure 10 presents mean MCC scores for the top-10 models across all 13 task categories, revealing substantial variation in category-specific performance even among leading models. Figure 11 displays the distribution of model rankings across all 100 tasks, highlighting that even top-ranked models exhibit considerable variance in per-task rankings. Finally, Figure 12 shows the distribution of task-level wins (achieving the highest MCC on a given task) and indicates that no single model dominates the benchmark: the top model wins only 20 of 100 tasks, with the remaining wins distributed across 15 additional models. This fragmentation underscores the importance of task-aware model selection.

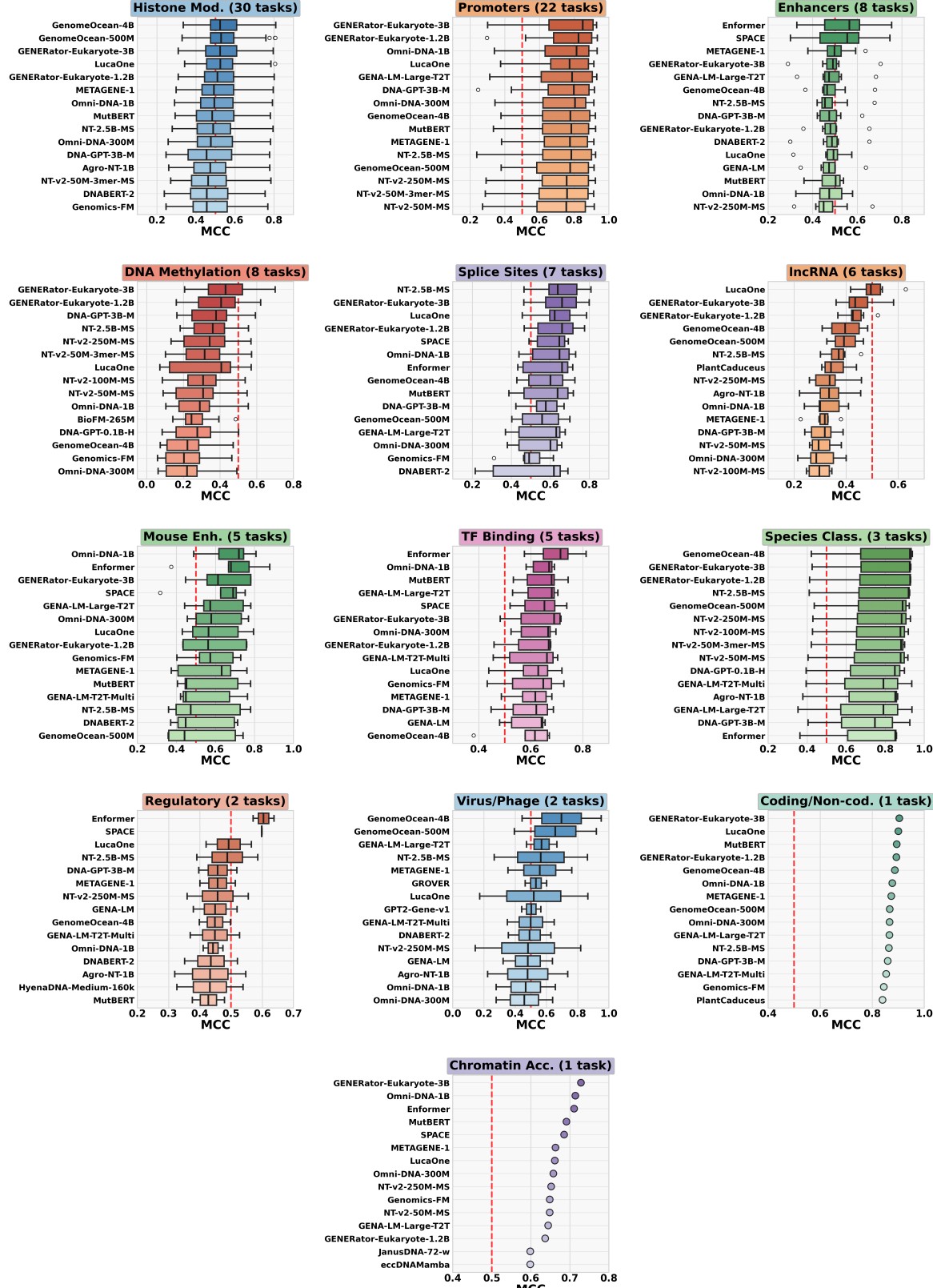

*Figure 9.* **Category-wise MCC distributions across top-performing models.** For each functional category, the figure reports the top-15 models ranked by mean MCC. Boxplots show the distribution of per-task MCC values within the category: boxes denote the interquartile range, central lines indicate medians, whiskers show the non-outlier range, and points mark outlier tasks. For single-task categories, individual MCC values are shown. The dashed red line marks MCC = 0.5.

## F.3. Performance Landscape by Model Capacity

We stratify models into four capacity tiers to examine the relationship between parameter count and downstream performance. Tiny models ($<100$M parameters, $n = 11$) achieve a mean macro-MCC of $0.443$, with MUTBERT (86M) leading the tier at $0.529$. Small models (100M–500M, $n = 16$) show modest improvement to $0.485$ mean macro-MCC, topped by GENA-LM-LARGE-T2T ($0.552$) and ENFORMER ($0.549$). Medium-scale models (500M–2B, $n = 6$) reach $0.526$ mean macro-MCC, with GENERATOR-EUKARYOTE-1.2B achieving the tier-best $0.581$. Large models ($\geq$ 2B, $n = 7$) attain $0.533$ mean macro-MCC, led by GENERATOR-EUKARYOTE-3B at $0.605$.

These tier-level statistics reveal a substantial scaling pattern: the Spearman correlation between $\log_{10}$(parameter count) and macro-MCC is $\rho = 0.565$ ($p < 0.001$; see Section 4 and Figure 2). Models above 1B parameters achieve mean macro-MCC of $0.527$ compared to $0.463$ for models below 200M, a gap of $+0.064$ macro-MCC. While this association is statistically robust, it does not preclude substantial within-tier variation: as shown in Section 4, 31 in-domain models demonstrate cases where a model at least $5\times$ smaller outperforms a larger counterpart, indicating that scale alone is not a sufficient predictor of category-level performance.

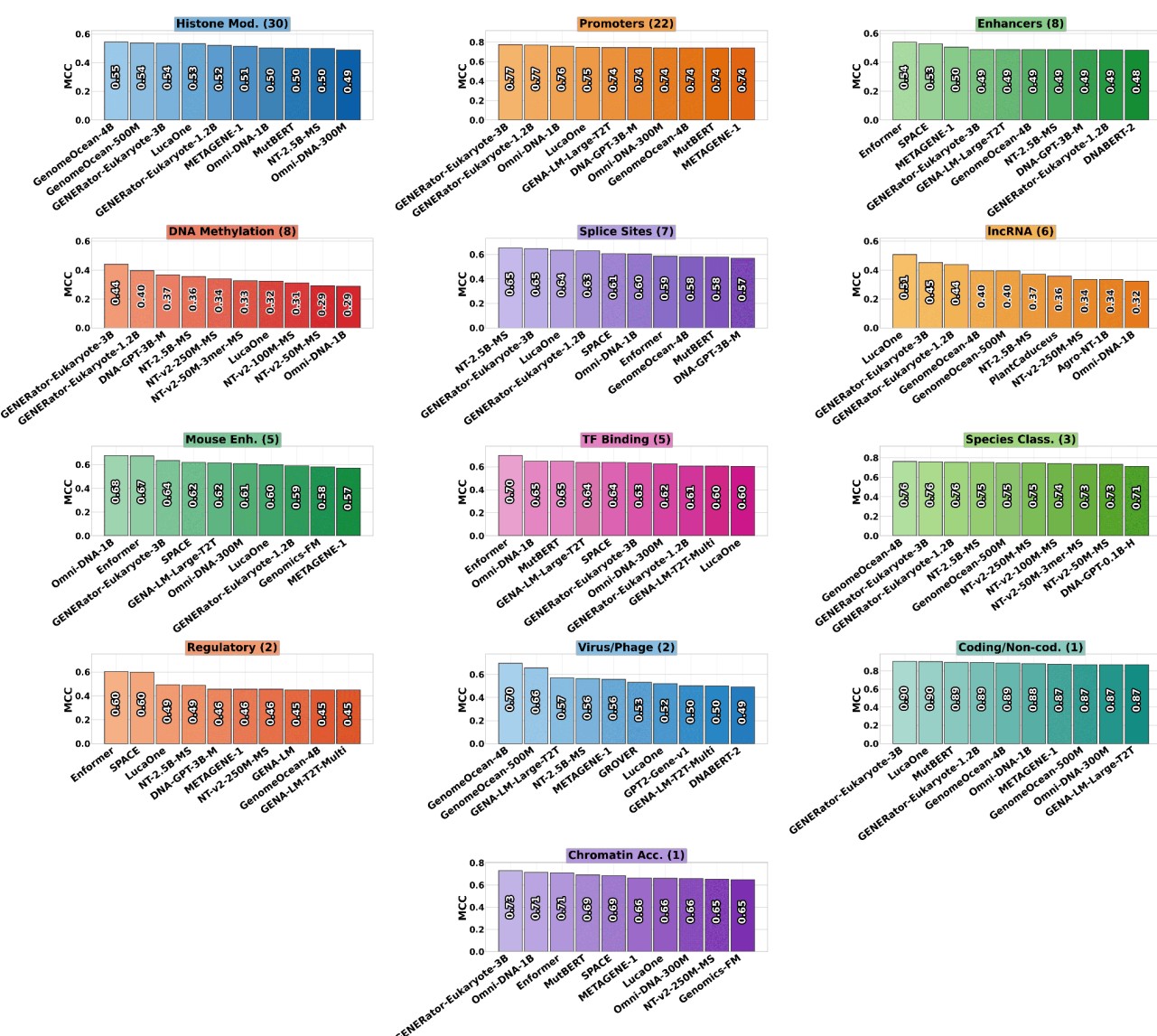

*Figure 10.* **Top-10 model performance across task categories.** Mean MCC is shown for the 10 best-performing models within each of the 13 functional task categories. Models are ranked independently within each category by category-level mean MCC, highlighting task-specific leaders and performance differences across genomic prediction settings.

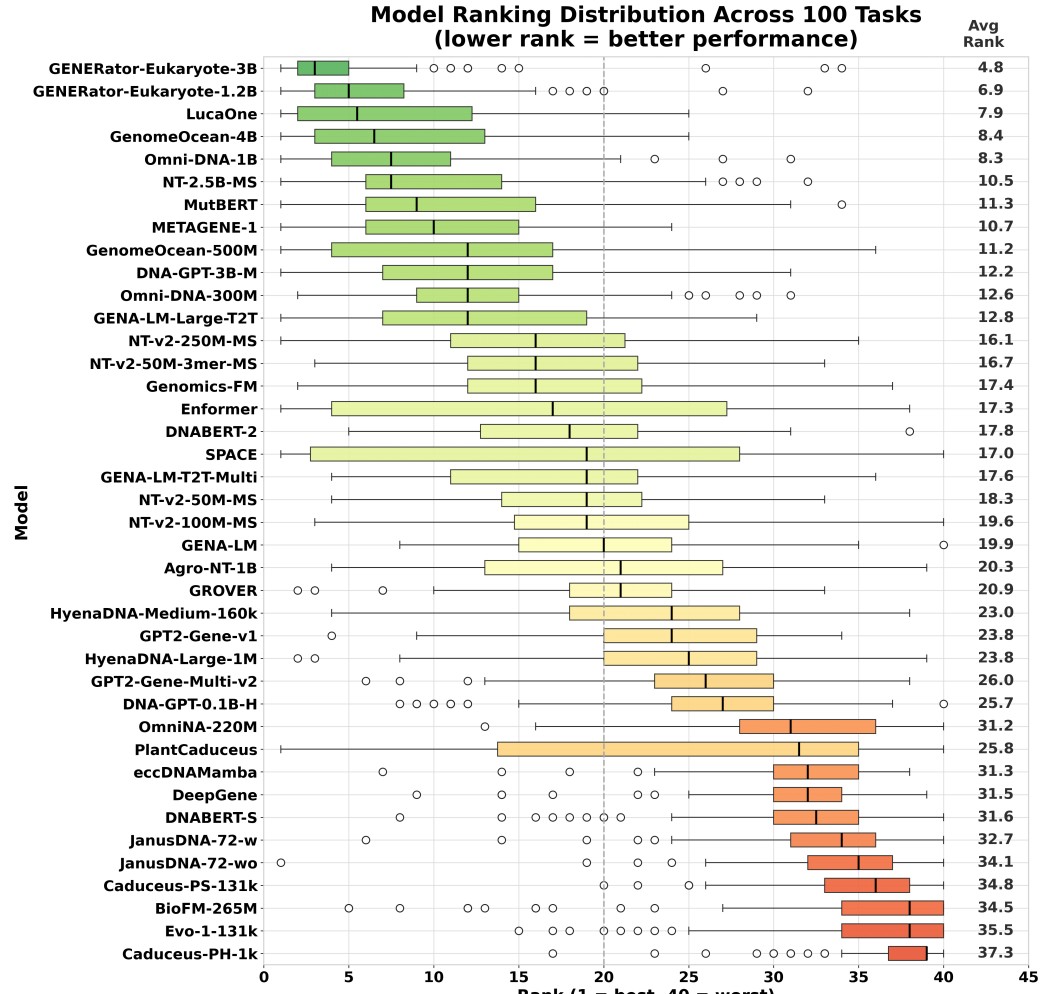

*Figure 11.* **Distribution of model ranks across benchmark tasks.** For each model, the boxplot shows the distribution of its task-level ranks across 100 benchmark tasks, where lower rank indicates better performance. Models are ordered by median rank, and the right column reports the average rank across all tasks. The leading models combine low median rank with relatively compact rank distributions, indicating consistently strong cross-task performance. In contrast, models with wide interquartile ranges or numerous outliers show substantial task-dependent variability, suggesting that aggregate performance can obscure pronounced category- and task-specific strengths.

## F.4. Architecture Comparison via Controlled Experiments

To isolate architectural effects from confounding factors, we identify model pairs sharing pretraining data and tokenization strategy while differing in architecture. The full enumeration of controlled pairs is given in Appendix E.3. This controlled comparison reveals consistent patterns favoring attention-based architectures over the evaluated state-space alternative under matched conditions.

OMNI-DNA-1B (Transformer-decoder) outperforms ECCDNAMAMBA (Mamba-SSM) by +0.149 macro-MCC (0.568 vs. 0.419), with both models trained on multi-species data using BPE tokenization. A second matched pair shows the same direction: GENOMEOCEAN-500M (Transformer-decoder) exceeds ECCDNAMAMBA by +0.131 macro-MCC (0.550 vs. 0.419) under identical pretraining and tokenization conditions. This pattern, observed across two independent matched pairs, is consistent with attention-based context modeling providing benefits over the evaluated state-space architecture in this setting.

Within the Transformer family, the encoder-vs-decoder ranking is less clear-cut. GENA-LM-LARGE-T2T (Transformer-encoder) outperforms OMNINA-220M (Transformer-decoder) by +0.127 macro-MCC (0.552 vs. 0.425) under matched multi-species/BPE conditions, and DNABERT-2 and GENA-LM-T2T-MULTI similarly exceed OMNINA-220M by

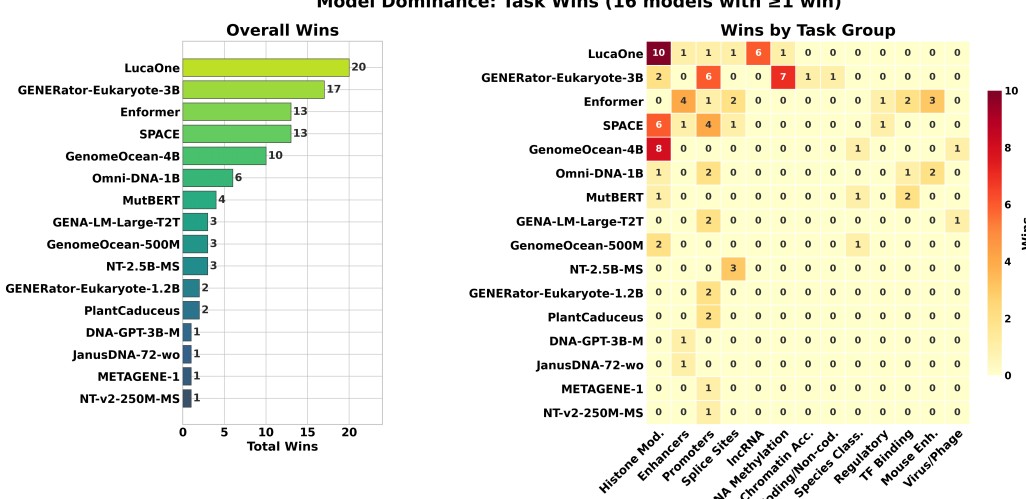

*Figure 12.* **Distribution of task-level wins across models.** The figure reports the number of benchmark tasks, out of 100, on which each model achieves the highest MCC. Only models with at least one task-level win are included. The left panel summarizes total wins per model, while the right panel decomposes these wins by functional task category. The dispersed pattern of wins across models and categories indicates that benchmark performance is strongly task-dependent and that no single model dominates uniformly across genomic prediction settings.

+0.086 and +0.097 macro-MCC respectively. However, as discussed in Section 4, the encoder–decoder comparison is task- and setting-dependent; the matched pairs reported here all share a single Transformer-decoder baseline (OMNINA-220M), and the pattern may not generalize to other Transformer-decoder models.

### F.5. Tokenization Strategy Effects

Tokenization represents a fundamental design choice that determines how nucleotide sequences are discretized for model consumption. Our controlled comparisons reveal architecture- and setting-dependent effects rather than a universally optimal strategy (Appendix E.3).

GPT2-GENE-V1 ($k$-mer) exceeds BIOFM-265M (BioToken custom scheme) by +0.134 macro-MCC (0.476 vs. 0.342), with both sharing a Transformer-decoder architecture and human pretraining. However, $k$-mer is not uniformly preferred: under matched multi-species pretraining, OMNI-DNA-300M (BPE) outperforms GPT2-GENE-MULTI-V2 ($k$-mer) by +0.071 macro-MCC (0.538 vs. 0.467) within the Transformer-decoder family. Under matched human pretraining, single-nucleotide tokenization (MUTBERT) exceeds both BPE baselines, by +0.038 over GROVER and +0.033 over GENA-LM.

These mixed results indicate that tokenization interacts with other design choices – particularly pretraining data composition and model architecture – in ways that preclude a single global ordering. The broader picture, including all 12 tokenization-controlled pairs, is discussed in Section 4 and enumerated in Appendix E.3.

### F.6. The Scale–Performance Paradox

A striking finding emerges from cross-scale comparisons: numerous smaller models substantially outperform larger counterparts. Among the 36 in-domain models (excluding prokaryotic-only, microbial-only, and plant-specific pretraining), we identify 31 instances where a model achieves superior aggregate macro-MCC despite being at least $5\times$ smaller than its comparison target. Within the full 40-model set, the count rises to 74, reflecting the additional contribution of cross-domain cases.

The most dramatic example involves MUTBERT (86M parameters, macro-MCC = 0.529) outperforming EVO-1-131K (7B parameters, macro-MCC = 0.298) by +0.231 macro-MCC despite an $81\times$ size disadvantage. The pattern repeats across cross-domain comparisons with EVO-1-131K as the larger model: OMNI-DNA-300M exceeds EVO-1-131K by +0.240 macro-MCC at a $23\times$ size differential; GENOMEOCEAN-500M achieves a +0.252 macro-MCC advantage at $14\times$ smaller scale; and OMNI-DNA-1B outperforms by +0.270 macro-MCC at $7\times$ fewer parameters. In-domain reversals (architectural

rather than domain-driven) are illustrated in the main paper by MUTBERT outperforming ECCDNAMAMBA by $+0.110$ macro-MCC at an $11.6\times$ size ratio (Section 4).

The cross-domain reversals are best interpreted as evidence of domain mismatch rather than as a universal critique of scaling. EVO-1-131K was pretrained exclusively on prokaryotic sequences, whereas 12 of 13 GENEB categories evaluate eukaryotic genomic functions (Domain Mismatch paragraph, Section 4); the size disadvantages reported above therefore conflate scale with domain alignment. The systematic analysis of pretraining data effects, restricted to controlled pairs matched on architecture and tokenization, is reported in Appendix E.3.

### F.7. Few-Shot Learning Dynamics

We evaluate model robustness under data-limited conditions through systematic few-shot experiments. Across all 40 models, mean macro-MCC degrades from 0.488 (full training data) to 0.253 (10-shot) to 0.106 (1-shot), corresponding to a 78.2% performance reduction in the extreme low-data regime (Figure 5).

Degradation patterns exhibit some heterogeneity across model capacity tiers but no monotonic trend. Tiny models ($<$100M, $n = 11$) decline from 0.443 to 0.109 (75.4% degradation); small models (100M–500M, $n = 16$) show a steeper decline from 0.485 to 0.093 (80.9%); medium models (500M–2B, $n = 6$) decline from 0.526 to 0.121 (77.1%); and large models ($\geq$ 2B, $n = 7$) decline from 0.533 to 0.121 (77.4%). The lack of a clear size-robustness trend at the tier level is consistent with the model-level pattern reported below.

Counterintuitively, the most few-shot-robust models in absolute terms are those with the weakest full-data performance. The five smallest absolute drops are observed for EVO-1-131K ($\Delta = 0.196$), CADUCEUS-PH-1K (0.220), JANUSDNA-72-WO (0.272), CADUCEUS-PS-131K (0.272), and JANUSDNA-72-W (0.275), all of which rank among the weakest models in full-shot evaluation. Conversely, the strongest full-shot performers exhibit the largest drops: GENERATOR-EUKARYOTE-3B ($\Delta = 0.489$), GENERATOR-EUKARYOTE-1.2B (0.463), LUCAONE (0.461), and NT-2.5B-MS (0.456), all exceeding 0.42 in absolute drop.

This inverse relationship should not be read as evidence of greater representational robustness in the lower-performing models: a small absolute drop reflects a low full-shot ceiling that leaves limited room for further degradation, not recovery of useful signal under 1-shot supervision (Section 4, Few-Shot Robustness paragraph). The practical implication is that aggregate few-shot rankings conflate task tractability with model quality and should be interpreted alongside full-shot performance, as discussed in Section 4.

### F.8. Pretraining Corpus Effects on Aggregate Performance

Grouping the 40 evaluated models by pretraining corpus type yields an ordering by mean macro-MCC that is consistent with the controlled pretraining-data comparisons reported in Appendix E.3. Eukaryotic gene-focused pretraining yields the highest aggregate performance (GENERATOR-EUKARYOTE-3B, GENERATOR-EUKARYOTE-1.2B; $n = 2$, mean macro-MCC $= 0.593$), followed by human-mouse epigenomic profiles (ENFORMER, SPACE; $n = 2$, 0.544) and broad multi-species corpora ($n = 20$, 0.522). Plant-specific pretraining (PLANTCADUCEUS, AGRO-NT-1B; $n = 2$, 0.463), human-only ($n = 12$, 0.430), and multi-species microbial (DNABERT-S; $n = 1$, 0.427) corpora form an intermediate band, with prokaryotic pretraining (EVO-1-131K; $n = 1$, 0.298) yielding the lowest aggregate macro-MCC.

We emphasize that this ordering is descriptive and confounded by architecture, tokenization, and parameter scale: only the controlled pairs reported in Appendix E.3 support clean attribution to pretraining corpus. Within those controlled comparisons (matched architecture and tokenization), the largest effect observed in GENEB is the multi-species vs. microbial contrast: GENOMICS-FM (multi-species) exceeds DNABERT-S (multi-species-microbial) by $+0.088$ macro-MCC under matched Transformer-encoder/$k$-mer conditions, and NT-V2-100M-MS similarly exceeds DNABERT-S by $+0.081$ macro-MCC. The eukaryotic-genes vs. broad multi-species contrast, supported by a single matched pair (GENERATOR-EUKARYOTE-3B vs. DNA-GPT-3B-M; both 3B, Transformer-decoder, $k$-mer), shows a $+0.063$ macro-MCC advantage for the eukaryotic-genes corpus, but rests on insufficient data to support strong claims (Transfer Learning Analysis paragraph, Section 4).

The ranking of aggregate means above is consistent with the biological expectation that taxonomic alignment between pretraining and downstream tasks supports transfer, since 12 of 13 GENEB categories evaluate eukaryotic genomic functions. Models pretrained on prokaryotic or microbial corpora are correspondingly disadvantaged under aggregate evaluation, as discussed in the Domain Mismatch paragraph (Section 4).

## F.9. Synthesis: Design Principles for DNA Foundation Models

Our systematic analysis yields several actionable principles for DNA foundation model development. First, architectural choice matters but interacts with other factors: Transformer-based models consistently outperform the evaluated state-space alternative under controlled conditions, while the encoder vs. decoder distinction is task- and setting-dependent (Section 4, Architecture Comparison paragraph). Second, tokenization effects are context-dependent and cannot be optimized in isolation from architecture and pretraining choices. Third, scale shows a substantial but non-deterministic association with performance (Spearman $\rho = 0.565$, $p < 0.001$): in-domain cross-scale reversals are common, and scale cannot compensate for data-domain mismatch – an 86M-parameter model trained on human sequences (MUTBERT) outperforms a 7B-parameter model trained on prokaryotic data (EVO-1-131K) by $+0.231$ macro-MCC.

Most critically, pretraining corpus composition is a substantial contributor to downstream performance. The descriptive ranking of aggregate means – eukaryotic genes > human-mouse profiles > multi-species > plant-specific > human-only $\approx$ microbial > prokaryotic – reflects taxonomic alignment with evaluation tasks, although the ordering is confounded by architecture, tokenization, and parameter scale, and only the controlled pairs reported in Appendix E.3 support clean attribution to corpus type. This pattern suggests that practitioners should prioritize domain-appropriate pretraining alongside architectural and scale considerations when computational resources are constrained.

## F.10. Transfer Learning Analysis

The diversity of pretraining data sources in our benchmark enables systematic investigation of cross-domain transfer dynamics. We analyze how representations learned from different taxonomic domains transfer to downstream tasks, revealing both positive and negative transfer phenomena with substantial practical implications. All values in this section are macro-averaged MCC (per-category averaging across the 13 functional categories of GENEB) unless otherwise stated; the full enumeration of matched pairs underlying the controlled comparisons appears in Appendix E.3.

### F.10.1. METHODOLOGICAL FRAMEWORK

To isolate pretraining-data effects from confounding architectural and tokenization factors, we rely on controlled-pair comparisons. The GENA-LM pair provides an ideal natural experiment: GENA-LM (human-only) and GENA-LM-T2T-MULTI (multi-species) share identical Transformer-encoder architecture and BPE tokenization, differing principally in pretraining corpus. This controlled setup enables more direct attribution of performance differences to data composition. The full 30-pair controlled-comparison set is detailed in Appendix E.3.

### F.10.2. CONTROLLED COMPARISON: HUMAN VERSUS MULTI-SPECIES PRETRAINING

The GENA-LM comparison reveals systematic advantages for multi-species pretraining across the task spectrum. Multi-species training yields superior per-category mean MCC in 11 of 13 task categories. The largest advantages emerge for Chromatin Accessibility ($\Delta = +0.123$ MCC: 0.583 vs. 0.461), Mouse Enhancers ($+0.067$: 0.548 vs. 0.480), and Species Classification ($+0.033$: 0.707 vs. 0.674). Human-only pretraining shows marginal advantages only for Enhancers ($+0.010$ MCC in favor of human-only) and Regulatory tasks ($+0.001$), with differences within noise margins.

This pattern generalizes beyond the GENA-LM pair. Across all six available human-vs. multi-species controlled comparisons (Appendix E.3, Table 16), multi-species pretraining is favored for Chromatin Accessibility (6/6 pairs), lncRNA (5/6 pairs), Splice Sites (4/6 pairs), and Mouse Enhancers (4/6 pairs). Human-only pretraining shows consistent advantage only for Virus/Phage (4/6 pairs, mean $\Delta = -0.034$ MCC), likely reflecting the predominance of human-associated viral sequences in human-genome training data.

### F.10.3. NEGATIVE TRANSFER: HUMAN TO PLANT DOMAINS

The lncRNA task category provides a stringent test of cross-kingdom transfer, comprising six plant-specific classification tasks spanning *Glycine max* (soybean), *Manihot esculenta* (cassava), *Sorghum bicolor* (sorghum), *Solanum lycopersicum* (tomato), *Triticum aestivum* (wheat), and *Zea mays* (maize). Human-trained models exhibit substantial negative transfer on these tasks, achieving mean lncRNA MCC of only 0.157 compared to 0.347 for plant-trained models – a deficit of 0.190 MCC, or roughly a 121% relative gain from domain-appropriate pretraining.

Task-level analysis reveals consistent patterns across all six plant species. On *G. max* lncRNA classification, human-trained models average 0.120 MCC versus 0.309 for plant-trained models. Similar gaps emerge for *S. bicolor* (0.172 vs. 0.408), *M.*

*esculenta* (0.137 vs. 0.408), and the remaining three species. Even the best-performing human-trained model (MUTBERT, lncRNA MCC = 0.260) substantially underperforms plant-specialized models (PLANTCADUCEUS, 0.357; AGRO-NT-1B, 0.336).

Notably, multi-species models achieve intermediate performance (mean lncRNA MCC = 0.304), with LUCAONE reaching 0.508 – the best overall result on plant lncRNA tasks. This suggests that broad taxonomic coverage partially compensates for the lack of plant-specific pretraining, though dedicated plant models retain advantages on most individual tasks.

### F.10.4. POSITIVE TRANSFER: MULTI-SPECIES TO HUMAN-SPECIFIC TASKS

A counterintuitive finding emerges from analysis of predominantly human-derived tasks: multi-species pretraining consistently outperforms human-only pretraining even on task categories whose underlying datasets come primarily from human genomic data. This positive transfer phenomenon manifests across all five human-centric task categories examined here.

For Histone Modifications, multi-species models achieve mean MCC of 0.473 compared to 0.397 for human-trained models ($\Delta = +0.076$). Chromatin Accessibility shows the largest gap at +0.097 (0.602 vs. 0.505). Regulatory element prediction exhibits +0.059 advantage (0.424 vs. 0.365), Enhancer detection +0.050 (0.462 vs. 0.412), and TF Binding +0.027 (0.569 vs. 0.542). The consistency of multi-species advantages across functionally diverse task categories is consistent with exposure to evolutionarily conserved sequence patterns during pretraining supporting representation quality even for species-specific downstream applications.

### F.10.5. CATASTROPHIC TRANSFER FAILURE: PROKARYOTIC TO EUKARYOTIC DOMAINS

The most extreme transfer failure involves EVO-1-131K, a 7B-parameter model pretrained exclusively on prokaryotic sequences. Despite its substantial scale, EVO-1-131K achieves only 0.298 overall macro-MCC, ranking last (40 of 40 models) and underperforming MUTBERT (86M parameters, human pretraining) by 0.231 macro-MCC – an 81-fold parameter disadvantage yielding inferior results.

Category-level analysis reveals the biological basis for this failure. EVO-1-131K ranks last (40/40) on Splice Sites (MCC = 0.160), TF Binding (0.173), Species Classification (0.285), and DNA Methylation (0.073). These failures reflect fundamental differences between prokaryotic and eukaryotic genomic organization: prokaryotes lack spliceosomal introns, employ distinct transcription factor families, and utilize different DNA methylation machinery. The only category where EVO-1-131K achieves competitive performance is Coding/Non-coding Classification (MCC = 0.719, rank 36/40), reflecting the more universal nature of coding sequence signatures across domains of life.

We emphasize three scope qualifiers for this observation. *First*, the result reflects performance under frozen linear-probing of pretrained representations – the evaluation protocol applied uniformly across all 40 benchmark models. It does not preclude the possibility that task-specific full fine-tuning of EVO-1-131K on eukaryotic tasks would close part of the gap, and is not a statement about EVO-1-131K's capabilities within its intended prokaryotic application domain, on which it was not evaluated here. *Second*, EVO-1-131K is the only prokaryotic-only model in GENEB; broader claims about prokaryotic-to-eukaryotic transfer would require additional prokaryotic-pretrained models, which we flag as a coverage limitation. *Third*, the magnitude of the failure on biologically structured tasks (splicing, DNA methylation) is consistent with the prior expectation that representations of prokaryotic sequence statistics carry limited information about eukaryotic-specific molecular machinery, and we frame this observation as evidence of domain mismatch rather than a universal critique of the underlying model architecture or training methodology.

### F.10.6. PARTIAL TRANSFER: MICROBIAL TO EUKARYOTIC DOMAINS

DNABERT-S, pretrained on multi-species microbial genomes, provides an intermediate case between prokaryotic-only and eukaryotic multi-species pretraining. With overall macro-MCC of 0.427, DNABERT-S substantially outperforms prokaryotic-only EVO-1-131K (0.298) but underperforms eukaryotic multi-species models (mean 0.522).

Controlled comparison with NT-V2-100M-MS (eukaryotic multi-species, matched architecture and tokenization) reveals systematic deficits for the microbial corpus: NT-V2-100M-MS exceeds DNABERT-S by +0.081 overall macro-MCC, with the largest per-category gaps on Splice Sites (+0.183) and Species Classification (+0.187). The splice site deficit again reflects the absence of spliceosomal machinery in microbial training sequences. Interestingly, DNABERT-S shows unusual strength on Regulatory tasks (+0.069 vs. GENOMICS-FM), potentially reflecting transferable sequence features in regulatory regions across bacterial and eukaryotic genomes. This intermediate-corpus observation rests on a single

microbial-pretrained model in GENEB (Appendix D); broader claims about microbial-to-eukaryotic transfer await additional matched models.

### F.10.7. STRONG POSITIVE TRANSFER: EUKARYOTIC GENE-FOCUSED PRETRAINING

The GENERATOR models, pretrained on curated eukaryotic gene sequences, achieve the strongest overall transfer performance in our benchmark (mean macro-MCC = 0.593). Remarkably, these models exceed benchmark-wide averages across all 13 task categories, with largest advantages on DNA Methylation (+0.200 vs. benchmark mean), Splice Sites (+0.198), lncRNA (+0.187), Species Classification (+0.137), Mouse Enhancers (+0.136), and Chromatin Accessibility (+0.117).

Controlled comparison between GENERATOR-EUKARYOTE-3B and DNA-GPT-3B-M (both 3B parameters, Transformer-decoder, $k$-mer tokenization) isolates the effect of gene-focused versus general multi-species pretraining. The gene-focused approach yields +0.063 overall macro-MCC advantage, with particularly strong gains on Chromatin Accessibility (+0.191), lncRNA (+0.142), and Mouse Enhancers (+0.124). This pattern is consistent with curation of pretraining data toward functionally annotated genomic regions supporting downstream task performance beyond what raw sequence diversity provides. We emphasize that this contrast rests on a single matched pair and should be interpreted with corresponding caution; see the Transfer Learning Analysis paragraph (Section 4) and the single-pair caveat in Appendix E.3.

### F.10.8. SPECIALIZED TRANSFER: HUMAN-MOUSE EPIGENOMIC PROFILES

ENFORMER and SPACE, pretrained on human and mouse epigenomic profiles, exhibit a distinctive transfer pattern characterized by strong performance on regulatory tasks but deficits on sequence-intrinsic features. These models achieve mean macro-MCC of 0.544 overall, with substantial advantages on Regulatory tasks (+0.194 vs. benchmark mean), Mouse Enhancers (+0.170), Chromatin Accessibility (+0.133), and TF Binding (+0.116).

However, this specialization comes at a cost: human-mouse-profile models underperform on DNA Methylation (−0.104), Coding/Non-coding Classification (−0.061), and lncRNA (−0.021). This trade-off reflects the nature of epigenomic profile pretraining, which emphasizes chromatin state and regulatory-element patterns while potentially underweighting primary sequence features. Practitioners should consider this specialization when selecting models for specific application domains (see also Practitioner Recommendations, Section 4).

### F.10.9. TRANSFER DYNAMICS ON MOUSE ENHANCER TASKS

The five Mouse Enhancer tasks provide a detailed view of how pretraining data influences performance on a homogeneous task category. Human-mouse-profile models (ENFORMER, SPACE) achieve top-1 performance on tasks 0 (ENFORMER MCC = 0.667), 2 (0.878), and 3 (0.772), with substantial margins over human-only models on these tasks. The remaining two tasks (1 and 4) are won by OMNI-DNA-1B (Transformer-decoder, multi-species, BPE; MCC = 0.807 and 0.488 respectively). Eukaryotic-gene GENERATOR models show consistent strong performance across all five tasks (mean 0.612) without claiming any individual top-1 position. Aggregating by pretraining-corpus group, mean macro-MCC on the Mouse Enhancer category is 0.646 for human-mouse-profile models, 0.612 for GENERATOR, 0.514 for broad multi-species, 0.399 for plant-trained, and 0.395 for human-only models.

The superior performance of human-mouse-profile models on mouse enhancer tasks – despite these models not being trained on general genomic sequences – demonstrates that task-relevant pretraining signals can outweigh broader sequence coverage when the downstream task aligns tightly with the pretraining target. This pattern is consistent with the practitioner guidance that model selection should be informed by task–domain alignment rather than aggregate benchmark performance alone (see Practitioner Recommendations, Section 4).

### F.10.10. SYNTHESIS: TRANSFER LEARNING PRINCIPLES

Our systematic analysis of cross-domain transfer reveals five recurring patterns governing DNA foundation model generalization.

First, taxonomic alignment between pretraining and downstream domains is critical for successful transfer. Prokaryotic pretraining fails on eukaryotic tasks, while human pretraining shows negative transfer to plant-specific tasks. The magnitude of these failures – 7B parameters underperforming 86M parameters by 0.231 macro-MCC – underscores that domain mismatch cannot be overcome through scale alone.

Second, taxonomic diversity in pretraining provides positive transfer even to ostensibly species-specific tasks. Multi-species models outperform human-only models on all five human-specific task categories examined here, consistent with broad sequence exposure during pretraining supporting transferable representations. This pattern argues against narrow species-specific pretraining for general-purpose foundation models.

Third, specialized pretraining creates performance trade-offs rather than uniform improvements. Human-mouse epigenomic profile models excel on regulatory tasks ($+0.194$ vs. benchmark mean) but underperform on DNA methylation ($-0.104$). Gene-focused eukaryotic pretraining achieves the highest overall performance but relies on curated training data. These trade-offs suggest that no single pretraining strategy optimally serves all downstream applications.

Fourth, model scale cannot compensate for fundamental data-domain mismatch. The $81$-fold parameter advantage of EVO-1-131K over MUTBERT is entirely negated by the prokaryotic–eukaryotic domain gap. This finding has practical resource-allocation implications: investment in domain-appropriate training data may yield higher returns than equivalent investment in model scale when domain mismatch exists.

Fifth, the observed transfer patterns suggest that model selection should be informed by task–domain alignment rather than aggregate benchmark performance alone. Different pretraining strategies excel on different task categories, motivating per-category selection guidance: regulatory tasks favor epigenomic-profile models (ENFORMER, SPACE); general eukaryotic tasks favor multi-species or gene-focused models; and for plant-specific tasks, dedicated plant-trained models substantially exceed human-trained baselines (mean lncRNA MCC $0.347$ vs. $0.157$), while the best individual results come from select multi-species models such as LUCAONE (lncRNA MCC $= 0.508$). The per-category Practitioner Recommendations in Section 4 provide operational guidance consistent with these alignment principles.

### F.11. Specialization Score: Formal Definition and Computation

For each model $m$ and each task $t$ we compute the per-task rank $r_{m,t} \in \{1, \ldots, 40\}$ from full-shot MCC, with rank $1$ denoting the best-performing model on task $t$. For a category $C$ containing tasks $\mathcal{T}_C$, the within-category mean rank of model $m$ is

$$\bar{r}_{m,C} \;=\; \frac{1}{|\mathcal{T}_C|} \sum_{t \in \mathcal{T}_C} r_{m,t},$$

and the corresponding outside-category mean rank is

$$\bar{r}_{m,\neg C} \;=\; \frac{1}{|\mathcal{T} \setminus \mathcal{T}_C|} \sum_{t \notin \mathcal{T}_C} r_{m,t},$$

where $\mathcal{T}$ is the full set of $100$ GENEB tasks. The specialization score of model $m$ on category $C$ is then

$$\Delta_{m,C} \;=\; \bar{r}_{m,\neg C} \;-\; \bar{r}_{m,C}.$$

A positive $\Delta_{m,C}$ indicates that model $m$ ranks better on category $C$ than on the rest of the benchmark; values above $\sim 5$ reflect substantial relative strength. The largest specialization scores observed in GENEB are BIOFM-265M on DNA methylation ($\Delta = 26.3$), PLANTCADUCEUS on lncRNA ($\Delta = 19.0$), JANUSDNA-72-W on chromatin accessibility ($\Delta = 17.9$), and ECCDNAMAMBA on chromatin accessibility ($\Delta = 16.5$). We use the score as a diagnostic measure throughout Appendix F, particularly when discussing model families whose category-level strengths differ from their aggregate ranking.

## F.12. Task-Category-Specific Analysis

Having established general principles governing DNA foundation model performance, we now examine category-specific patterns across six major task groups. This fine-grained analysis reveals substantial heterogeneity in optimal model configurations across functional genomic prediction problems.

### F.12.1. HISTONE MODIFICATIONS

The Histone Modifications category ($n = 30$ tasks) spans diverse chromatin marks including H3K4me1, H3K4me2, H3K4me3, H3K27ac, H3K36me3, and H4 modifications. Task difficulty varies substantially: the easiest individual task is GUE EMP H4 prediction (mean MCC = 0.720 across the 40 models), while the most challenging is GUE EMP H3K4me3 prediction (mean MCC = 0.243). Results for the 30 histone modification prediction tasks are presented in Figures 13–15.

Architecture comparisons under controlled conditions reveal substantial advantages for Transformer-based models. GENOMEOCEAN-500M (Transformer-decoder) exceeds ECCDNAMAMBA (Mamba-SSM-bidirectional) by 0.153 MCC (0.537 vs. 0.384), with both models trained on multi-species data using BPE tokenization. A within-Transformer contrast (decoder vs. encoder) between GENOMEOCEAN-500M (500M, decoder, BPE, multi-species) and GENA-LM-LARGE-T2T (336M, encoder, BPE, multi-species) shows a 0.069 MCC advantage for the decoder model, although the two models differ in parameter count and the contrast is therefore not strictly matched on scale.

Pretraining data effects follow the general hierarchy established in Section F.8. Eukaryotic gene-focused GENERATOR-EUKARYOTE-3B exceeds multi-species DNA-GPT-3B-M by 0.060 MCC (0.537 vs. 0.477) in controlled comparison (both 3B, Transformer-decoder, $k$-mer tokenization). Multi-species models consistently outperform microbial-trained alternatives, with GENOMICS-FM exceeding DNABERT-S by 0.096 MCC (0.469 vs. 0.373) under matched Transformer-encoder/$k$-mer conditions. Among small-scale (<100M) models, MUTBERT (86M) achieves the highest Histone Modifications MCC (0.501), ranking 8th of 40 overall on this category.

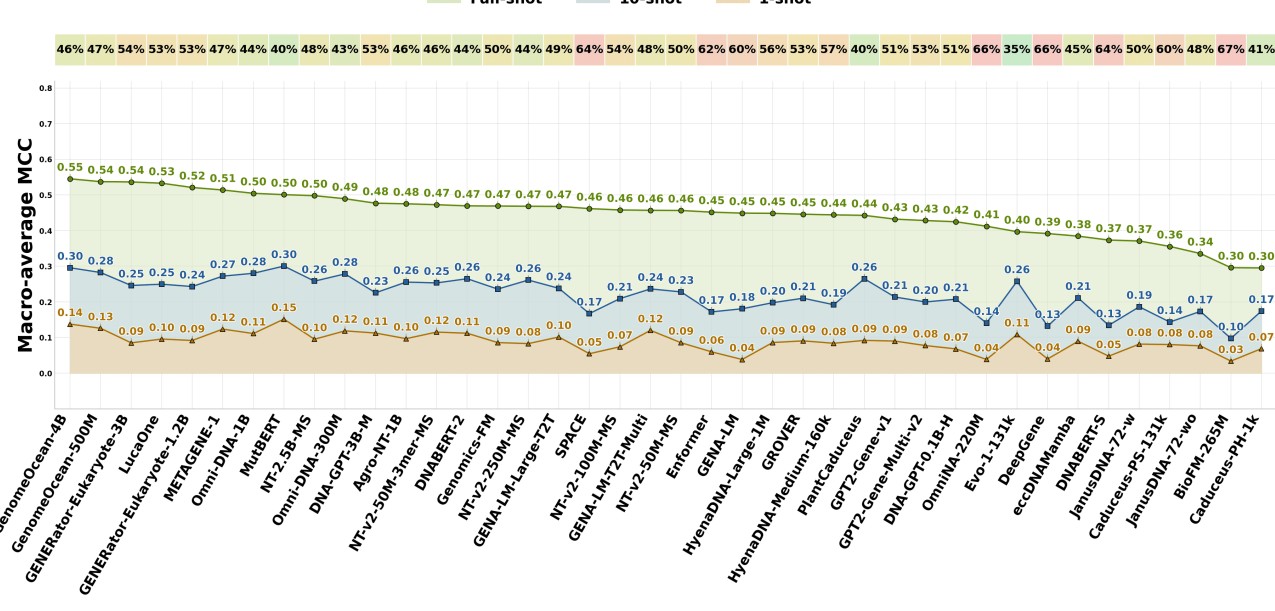

*Figure 13.* **Few-shot performance degradation on Histone Modifications.** For each of the 40 models, macro-average MCC across 30 histone modification tasks under full-shot, 10-shot, and 1-shot regimes; models ordered by full-shot performance. The top band shows the relative drop from full-shot to 10-shot per model. Benchmark-wide mean degradation: 51.0% for 10-shot, 80.3% for 1-shot. The 1-shot regime collapses to near-random performance, while 10-shot retains discriminative signal; full-shot and 10-shot rankings differ substantially (Spearman $\rho = 0.77$).

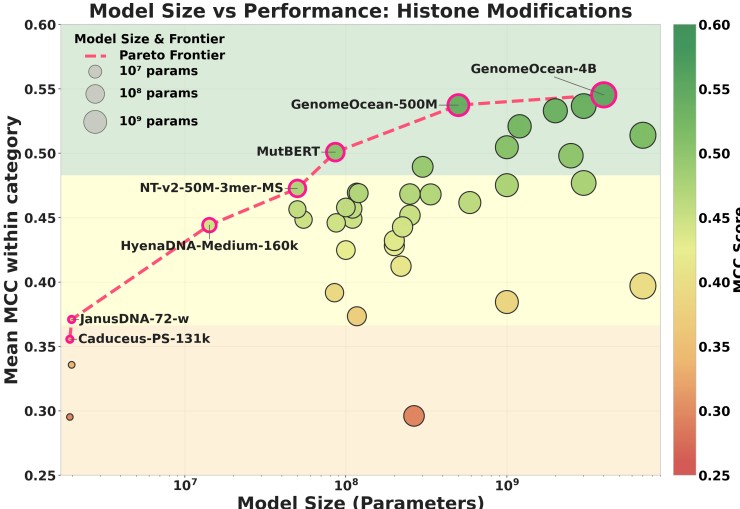

*Figure 14.* **Pareto frontier for Histone Modifications: mean MCC vs. parameter count.** Each point represents one of the 40 genomic foundation models, with parameter count on a logarithmic x-axis and mean full-shot Histone Modifications MCC on the y-axis. Marker size and color both encode MCC. The dashed line marks the Pareto frontier of best performance–size trade-offs. Scale shows a positive but non-deterministic association with Histone Modifications performance: several smaller models sit on or near the frontier, including MUTBERT (86M, mean MCC = 0.501), which is the strongest sub-100M model on this category. Other small models such as JANUSDNA-72-W and JANUSDNA-72-WO (both 1.98M parameters) sit well below the frontier, illustrating that small parameter count is neither sufficient nor consistent for competitive performance on this category.

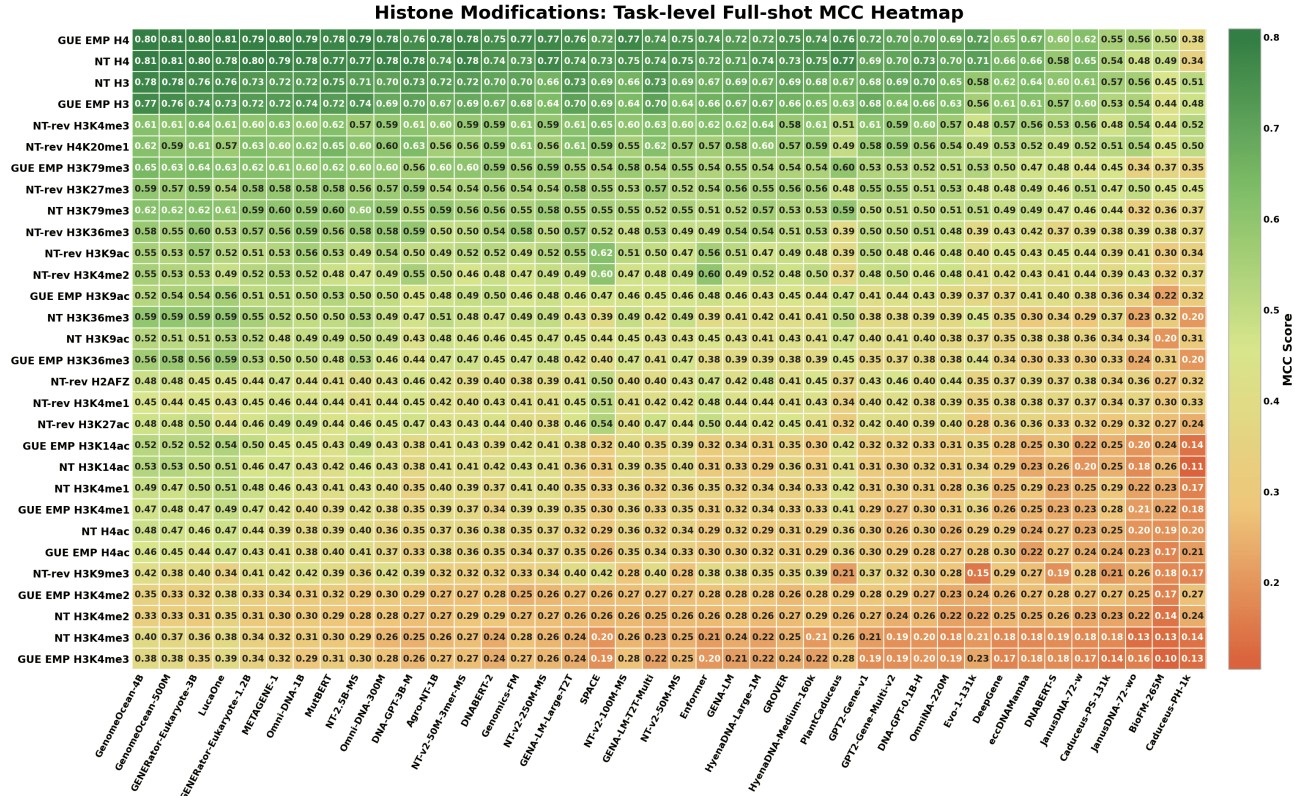

*Figure 15.* **Per-task MCC for Histone Modifications.** Heatmap shows full-shot MCC for each of the 40 genomic foundation models on the 30 histone modification tasks, with models sorted by mean Histone Modifications MCC. Cell values report per-task MCC, with colors ranging from red/orange for lower scores to green for higher scores. Task difficulty varies substantially across marks: H4-family tasks (H4, H4ac, H4K20me1) are consistently among the easiest, while H3K4me2/me3 tasks remain among the hardest across most models. Tasks from the NT, NT-revised, and GUE EMP sources show broadly similar within-mark patterns, supporting the consolidation of these sources into a single category.

Few-shot performance on histone modification tasks degrades severely. The mean macro-MCC across models drops from 0.447 (full data) to 0.219 (10-shot) to 0.088 (1-shot), corresponding to 51.0% and 80.3% relative reductions respectively. We focus on the 10-shot regime here as the 1-shot regime collapses to near-random performance for all models (no model exceeds 0.16 macro-MCC at 1-shot), limiting its discriminative value.

At 10-shot, the ranking of models differs substantially from full-shot performance (Spearman $\rho = 0.77$ across all 40 models). The top 10-shot performer is MUTBERT (86M parameters; 10-shot MCC = 0.300, full-shot = 0.501), which ranks 11th at full-shot, illustrating that compact, efficiency-oriented models can retain operationally useful transferability under data-limited conditions where larger gene-focused models do not (e.g., GENERATOR-EUKARYOTE-3B drops to 10-shot MCC = 0.246 from a full-shot MCC of 0.537). Of the top-5 models by full-shot MCC, only 2 remain in the top-5 at 10-shot (GENOMEOCEAN-4B, GENOMEOCEAN-500M). As discussed in Section F.7, this reranking reflects task- and setting-dependent few-shot dynamics rather than a single property of model robustness.

### F.12.2. PROMOTER RECOGNITION

Promoter recognition tasks ($n = 22$ tasks) span bacterial, plant, and mammalian systems with substantial difficulty variation. Cell-type-specific human promoter tasks dominate the easy end (IPRO-WAEL HUVEC: mean MCC = 0.890; HeLa-S3: 0.875; GM12878: 0.856), while bacterial promoter prediction for *R. capsulatus* is the most challenging task (0.274). Results for the 22 promoter prediction tasks are presented in Figures 16–18.

The scaling relationship is weaker for promoter tasks than for the benchmark overall. The Spearman correlation between $\log_{10}$(parameter count) and mean Promoter MCC is $\rho = 0.487$ ($p = 0.001$), compared to $\rho = 0.565$ across all categories (Section 4). Models with at least 1B parameters achieve mean MCC of 0.717 versus 0.682 for models below 200M – a gap of 0.035 MCC, narrower than the +0.064 tier gap observed at the benchmark-aggregate level. This compressed performance range suggests that promoter sequence features are relatively accessible to smaller models.

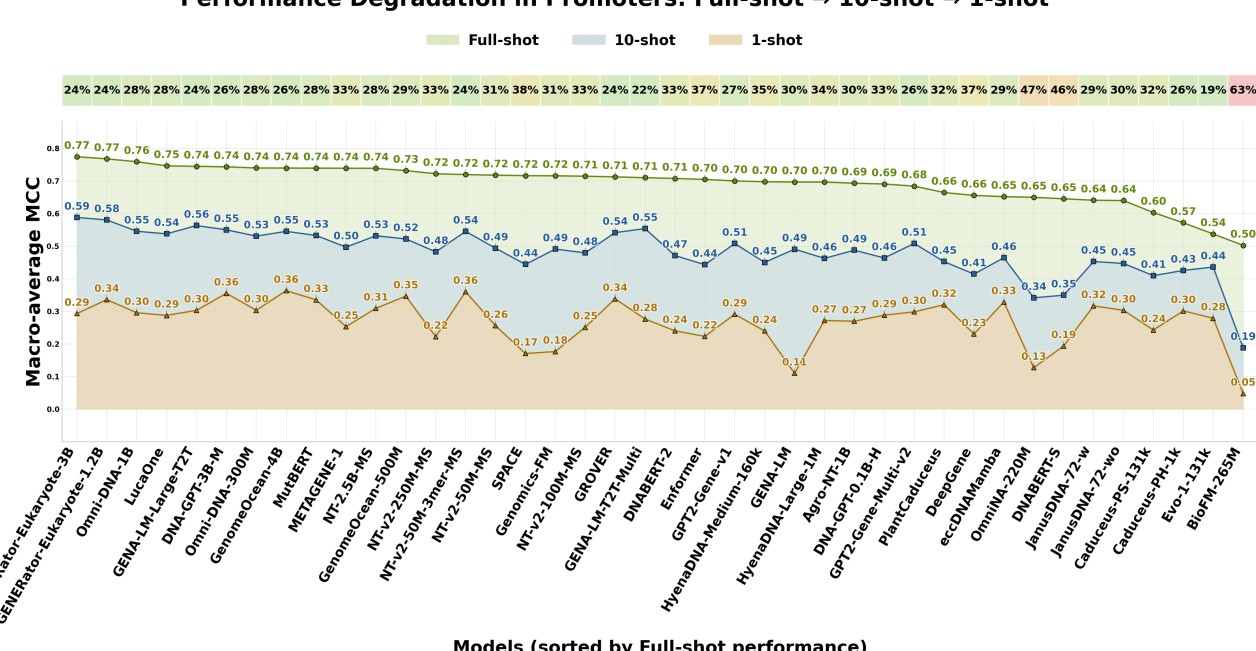

*Figure 16.* **Few-shot performance degradation on Promoter Recognition.** For each of the 40 models, macro-average MCC across 22 promoter prediction tasks under full-shot, 10-shot, and 1-shot regimes; models ordered by full-shot performance. The top band shows the relative drop from full-shot to 10-shot per model. Benchmark-wide mean degradation: 30.7% for 10-shot, 61.2% for 1-shot. Unlike histone modifications, the 1-shot regime retains operationally meaningful signal on this category (maximum 1-shot MCC = 0.363); full-shot and 10-shot rankings remain strongly correlated (Spearman $\rho = 0.85$).

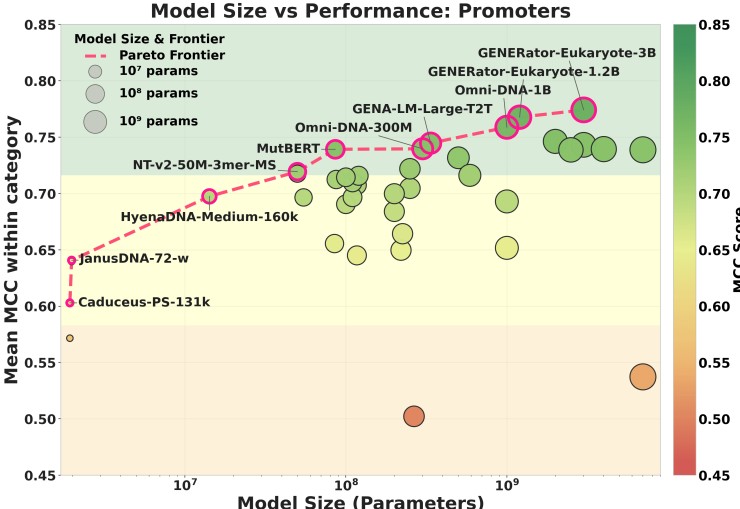

*Figure 17.* **Pareto frontier for Promoter Recognition: mean MCC vs. parameter count.** Each point represents one of the 40 genomic foundation models, with parameter count on a logarithmic x-axis and mean full-shot Promoter MCC on the y-axis. Marker size and color both encode MCC. The dashed line marks the Pareto frontier of best performance–size trade-offs. Scaling is comparatively weak on this category (Spearman $\rho = 0.487$; tier gap $\geq$ 1B vs. <200M of only 0.035 MCC), consistent with promoter sequence features being accessible to smaller models. MUTBERT (86M, mean MCC = 0.739, ranked 9th overall on this category) is the strongest sub-100M model, sitting near the Pareto frontier alongside several multi-billion-parameter models.

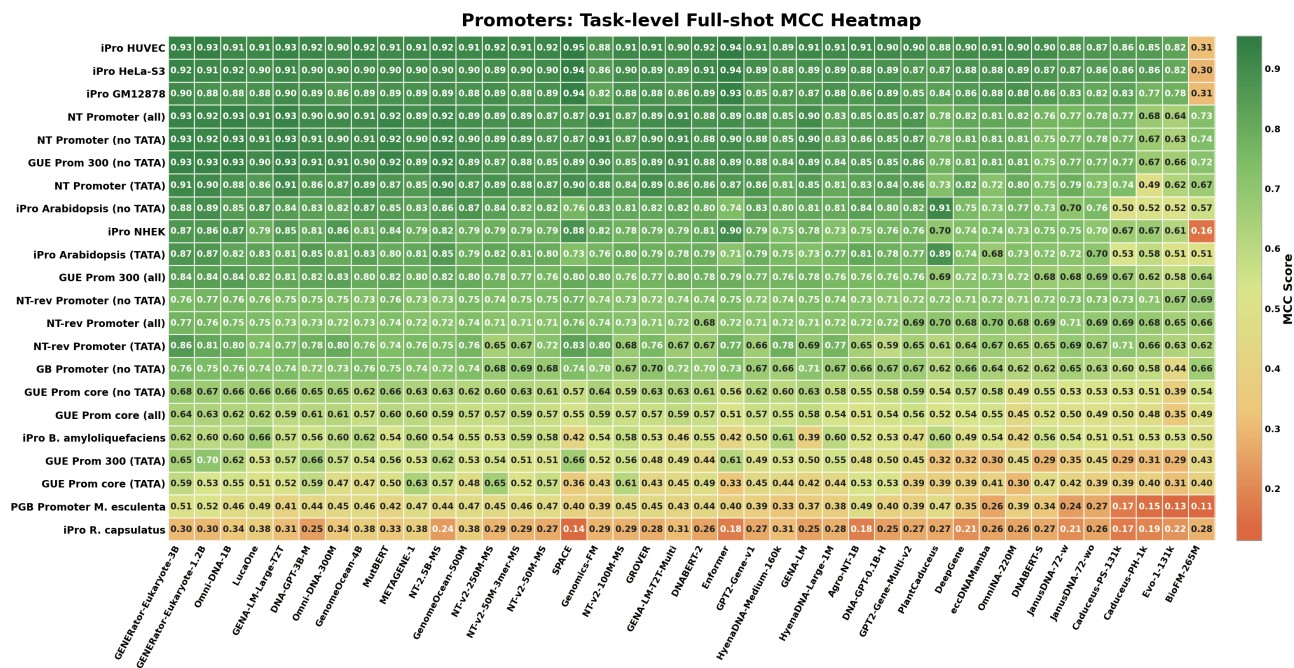

*Figure 18.* **Per-task MCC for Promoter Recognition.** Heatmap shows full-shot MCC for each of the 40 genomic foundation models on the 22 promoter prediction tasks, with models sorted by mean Promoter MCC. Cell values report per-task MCC, with colors ranging from red/orange for lower scores to green for higher scores. Task difficulty varies substantially with sequence source and host species: cell-type-specific human promoter prediction (HUVEC, HeLa-S3, GM12878) is consistently the easiest, while bacterial promoter prediction (*R. capsulatus*, *B. amyloliquefaciens*) and plant promoter prediction (*M. esculenta*) are the most challenging.

Architecture comparisons maintain Transformer advantages over the evaluated state-space alternative: OMNI-DNA-1B (Transformer-decoder, 1B) outperforms ECCDNAMAMBA (Mamba-SSM, 1B) by 0.107 MCC (0.759 vs. 0.652), with both models trained on multi-species data using BPE tokenization. Within Transformer-family comparisons, GENA-LM-

LARGE-T2T (encoder, BPE, multi-species, 336M) exceeds OMNINA-220M (decoder, BPE, multi-species, 220M) by 0.095 MCC (0.745 vs. 0.650), although the two models differ in parameter count and the contrast is therefore not strictly matched on scale. We note that this single-pair encoder advantage on Promoters contrasts with the within-Transformer decoder advantage observed on Histone Modifications, supporting the broader observation (Section 4, Architecture Comparison paragraph) that the encoder–decoder distinction is task- and setting-dependent.

Pretraining data comparisons reveal nuanced patterns for promoter tasks. GENA-LM-T2T-MULTI (multi-species) exceeds GENA-LM (human-only) by only 0.013 MCC (0.710 vs. 0.697) under matched Transformer-encoder/BPE conditions, a narrower advantage than the average multi-vs.-human pretraining contrast reported in Appendix E.3. Notably, in the Transformer-decoder/$k$-mer matched pair, GPT2-GENE-V1 (human-only) slightly exceeds GPT2-GENE-MULTI-V2 (multi-species) by 0.016 MCC, consistent with the observation that the multi-species advantage is not invariant to architecture and tokenization choice (Section F.10).

Few-shot performance on promoter tasks degrades more gracefully than on histone modifications. Across the 40 models, mean MCC drops from 0.693 (full data) to 0.480 (10-shot) to 0.269 (1-shot), corresponding to 30.7% and 61.2% relative reductions respectively. Unlike histone modification, several models retain operationally meaningful 1-shot performance (maximum 1-shot MCC = 0.363 for GENOMEOCEAN-4B). At 10-shot, the top performers remain GENERATOR-EUKARYOTE-3B (0.588), GENERATOR-EUKARYOTE-1.2B (0.580), and GENA-LM-LARGE-T2T (0.563); the full-shot top-5 and 10-shot top-5 overlap by 3 models, and full-shot vs. 10-shot rankings correlate at Spearman $\rho = 0.85$.

### F.12.3. ENHANCER PREDICTION

Enhancer prediction tasks ($n = 8$ tasks) present moderate difficulty with mean MCC of 0.446 across the 40 models. The category spans human enhancer prediction (NT, NT-rev, and GB Cohn and Ensembl subsets), enhancer-type classification (NT and NT-rev multi-class variants), *Drosophila* enhancers (GB Stark), and a mouse enhancer task (GB Ensembl). Within-category difficulty varies from a low of 0.321 (NT Enhancers (types), multi-class) to a high of 0.567 (GB Mouse enh. (Ensembl)). Results for the 8 enhancer prediction tasks are presented in Figures 19–21.

Human-mouse epigenomic profile models demonstrate the strongest full-shot performance on enhancer tasks, consistent with their specialized pretraining on regulatory elements. ENFORMER achieves the top mean MCC on the category (0.539; rank 1 of 40 by mean Enhancer MCC) while ranking 10th of 40 overall on the benchmark, a category-specific advantage. SPACE follows closely (mean MCC = 0.526; rank 2 on Enhancers). The advantage reflects the close relationship between enhancer sequences and the chromatin-state signals present in human-mouse epigenomic profile pretraining (Section F.10).

Category-favoring specialization (defined as the gap between a model's average per-task rank on the 92 non-enhancer tasks and its average per-task rank on the 8 enhancer tasks) is observed across diverse architectures rather than being confined to a single architectural family. Beyond ENFORMER ($\Delta = +11.3$) and SPACE ($\Delta = +7.2$), notable specializers include JANUSDNA-72-WO ($\Delta = +7.6$; Hybrid-Mamba-MoE, 1.98M parameters), GENA-LM ($\Delta = +7.2$; Transformer-encoder, human-only, 110M), DNABERT-2 ($\Delta = +6.4$; Transformer-encoder, multi-species, 117M), and GROVER ($\Delta = +5.2$; Transformer-encoder, human-only, 87M). That JANUSDNA-72-WO appears among the top specializers at 1.98M parameters is noteworthy given its very small scale; among sub-100M models on this category, however, MUTBERT (86M, mean MCC = 0.475) achieves the highest full-shot Enhancer MCC. The architectural diversity of strong specializers argues against attributing enhancer specialization to any single design choice.

Few-shot performance degrades substantially on enhancer tasks. Across the 40 models, mean MCC drops from 0.446 (full data) to 0.278 (10-shot) to 0.134 (1-shot), corresponding to 37.6% and 70.0% relative reductions respectively. The 1-shot regime collapses to near-random performance (maximum 1-shot MCC = 0.195 for ECCDNAMAMBA). The 10-shot ranking differs sharply from the full-shot ranking (Spearman $\rho = 0.64$; top-5 overlap of only 1 model out of 5), with both epigenomic-profile specialists losing their full-shot advantage: ENFORMER drops from full-shot mean MCC 0.539 (rank 1) to 10-shot MCC 0.252 (rank 34), and SPACE drops from 0.526 (rank 2) to 0.307 (rank 10), illustrating that strong full-data category specialization need not translate to few-shot transferability. At 10-shot, the top performers are GENA-LM-LARGE-T2T (0.372), OMNI-DNA-300M (0.342), and OMNI-DNA-1B (0.342).

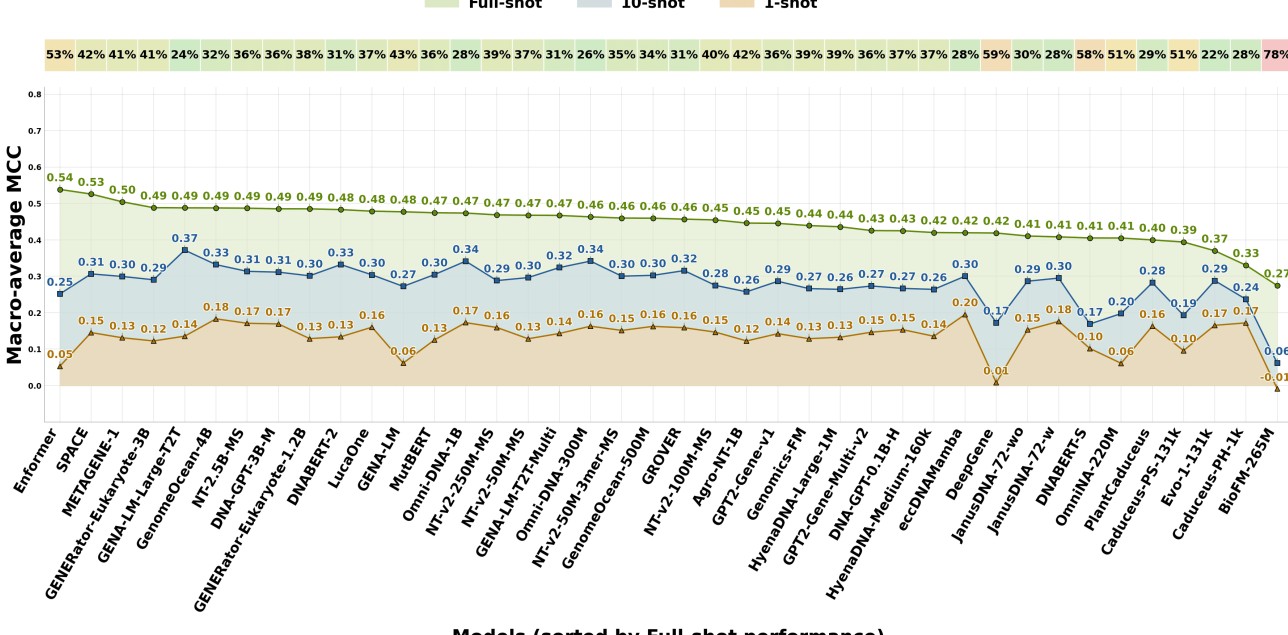

*Figure 19.* **Few-shot performance degradation on Enhancer Prediction.** For each of the 40 models, macro-average MCC across 8 enhancer prediction tasks under full-shot, 10-shot, and 1-shot regimes; models ordered by full-shot performance. The top band shows the relative drop from full-shot to 10-shot per model. Benchmark-wide mean degradation: 37.6% for 10-shot, 70.0% for 1-shot. The 1-shot regime collapses to near-random performance (maximum 1-shot MCC = 0.195); the 10-shot ranking diverges sharply from the full-shot ranking (Spearman $\rho = 0.64$), with epigenomic-profile specialists losing their full-shot advantage (ENFORMER: full = 0.539, 10-shot = 0.252; SPACE: full = 0.526, 10-shot = 0.307).

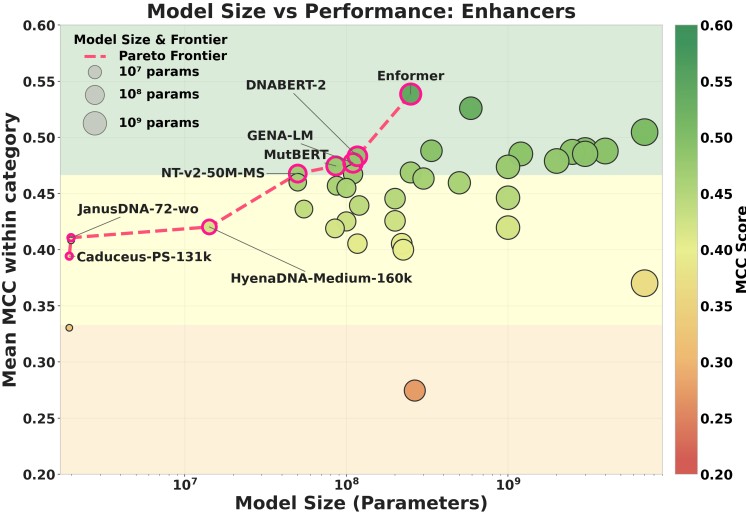

*Figure 20.* **Pareto frontier for Enhancer Prediction: mean MCC vs. parameter count.** Each point represents one of the 40 genomic foundation models, with parameter count on a logarithmic $x$-axis and mean full-shot Enhancer MCC on the $y$-axis. Marker size and color both encode MCC. The dashed line marks the Pareto frontier of best performance–size trade-offs. Scaling on this category is moderate (Spearman $\rho = 0.491$; tier gap $\geq$ 1B vs. <200M of 0.031 MCC). Specialized human-mouse epigenomic-profile models occupy the frontier at moderate sizes (ENFORMER, 250M, MCC = 0.539; SPACE, 589M, MCC = 0.526), outperforming multi-billion-parameter generalist models. MUTBERT (86M, mean MCC = 0.475) is the strongest sub-100M model on this category.

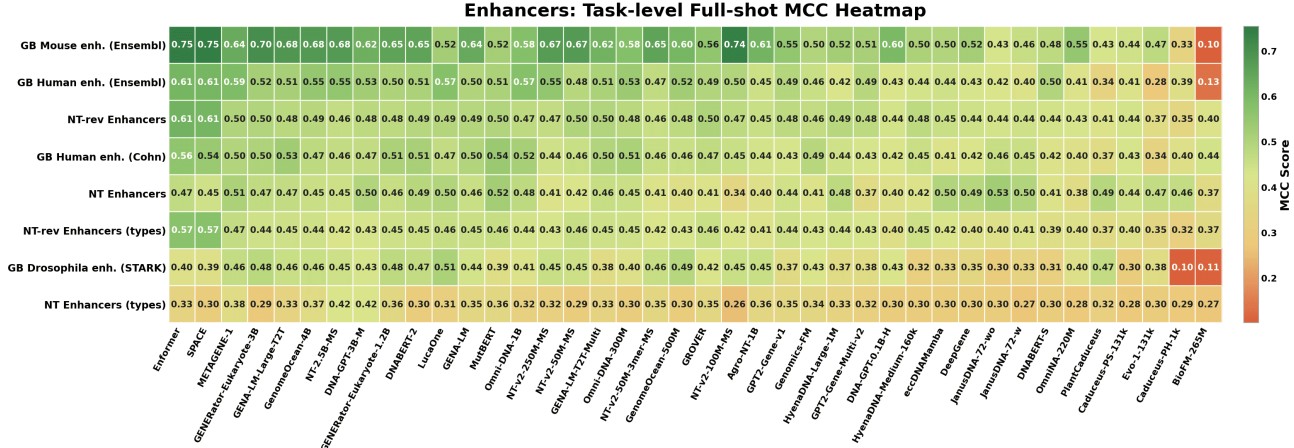

*Figure 21.* **Per-task MCC for Enhancer Prediction.** Heatmap shows full-shot MCC for each of the 40 genomic foundation models on the 8 enhancer prediction tasks, with models sorted by mean Enhancer MCC. Cell values report per-task MCC, with colors ranging from red/orange for lower scores to green for higher scores. Task difficulty varies substantially: the multi-class NT Enhancers (types) task is the most challenging (mean MCC = 0.321), while GB Mouse enh. (Ensembl) is the easiest (mean MCC = 0.567). Human-mouse epigenomic-profile models (ENFORMER, SPACE) and Transformer-encoder models (DNABERT-2, GENA-LM, GROVER) cluster at the top of the model ordering.

### F.12.4. DNA METHYLATION

DNA methylation prediction ($n = 8$ tasks) reveals the most pronounced specialization effects in our benchmark. Task difficulty spans an extreme range: the iDNA-ABF 6mA task achieves $0.450$ mean MCC while the hardest 4mC task (*G. subterraneus*) approaches random performance at $0.061$ MCC. Results for the 8 DNA methylation prediction tasks are presented in Figures 22–24.

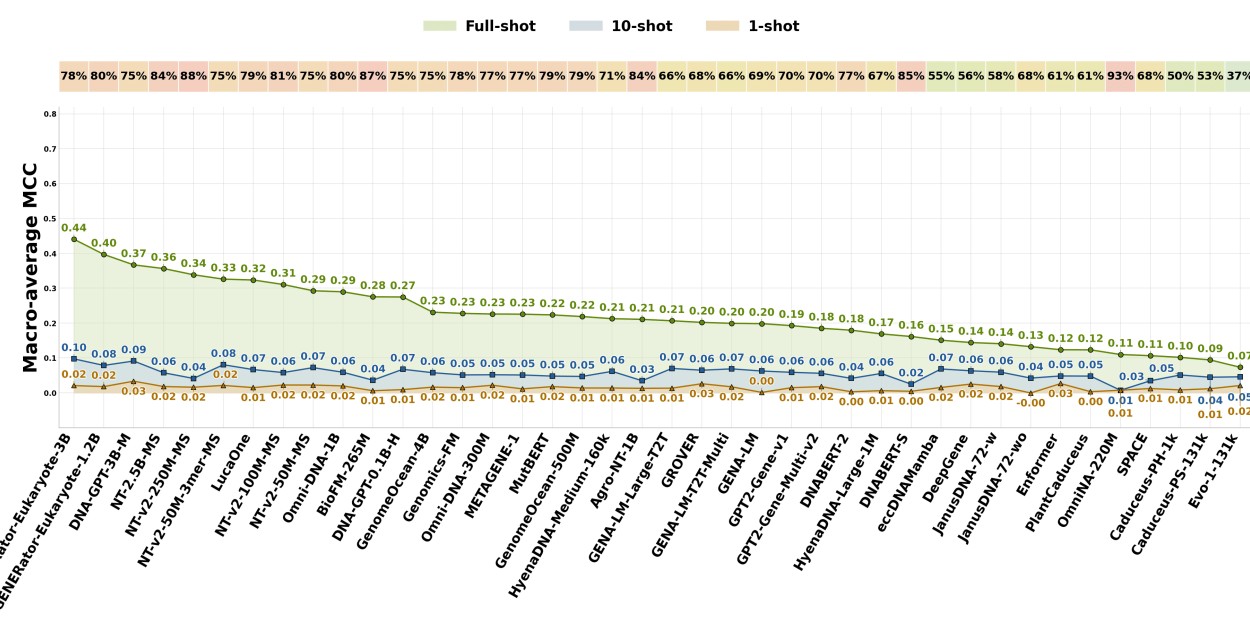

*Figure 22.* **Few-shot performance degradation on DNA Methylation.** For each of the 40 models, macro-average MCC across 8 DNA methylation tasks under full-shot, 10-shot, and 1-shot regimes; models ordered by full-shot performance. The top band shows the relative drop from full-shot to 10-shot per model. Benchmark-wide mean degradation: 74.7% for 10-shot, 93.2% for 1-shot – among the most severe of any task category. Both 10-shot (maximum MCC = 0.097) and 1-shot (maximum = 0.033) regimes collapse to near-random performance for all models; full-shot and 10-shot rankings correlate at Spearman $\rho = 0.50$.

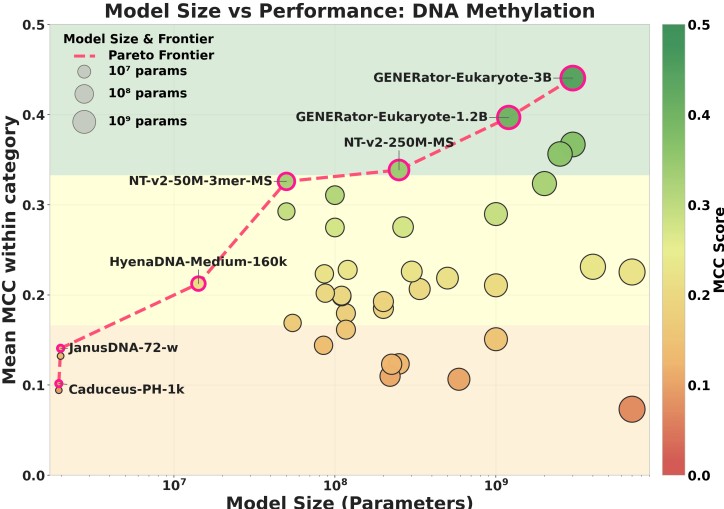

*Figure 23.* **Pareto frontier for DNA Methylation: mean MCC vs. parameter count.** Each point represents one of the 40 genomic foundation models, with parameter count on a logarithmic x-axis and mean full-shot DNA Methylation MCC on the y-axis. Marker size and color both encode MCC. The dashed line marks the Pareto frontier of best performance–size trade-offs. NT-v2-50M-3MER-MS (50M, mean MCC = 0.326) is the strongest sub-100M model on this category, ranking 6th of 40 overall and sitting near the frontier alongside multi-billion-parameter GENERATOR models.

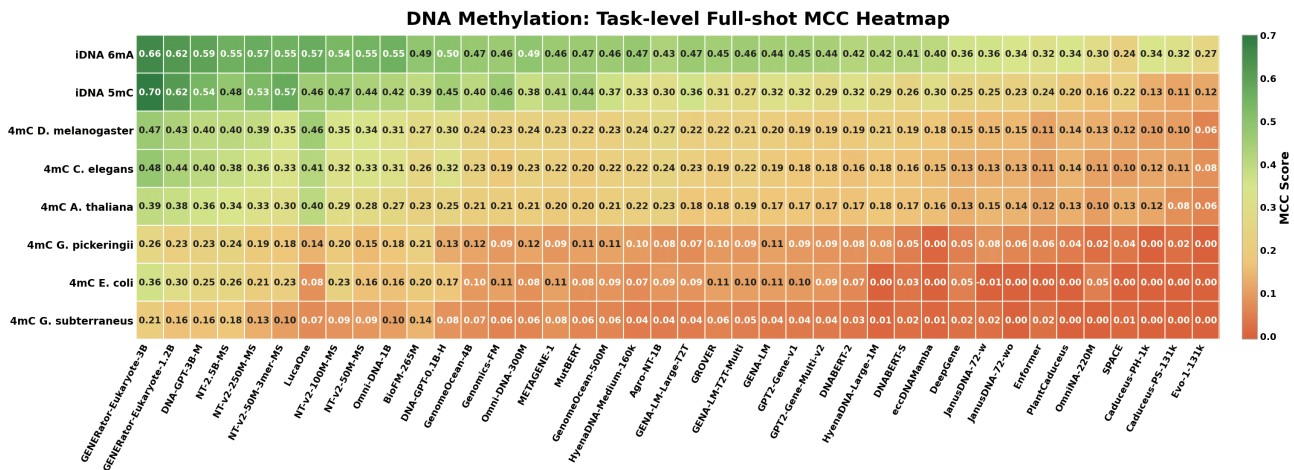

*Figure 24.* **Per-task MCC for DNA Methylation.** Heatmap shows full-shot MCC for each of the 40 genomic foundation models on the 8 DNA methylation tasks (six 4mC, one 5mC, one 6mA), with models sorted by mean DNA Methylation MCC. Cell values report per-task MCC, with colors ranging from red/orange for lower scores to green for higher scores.

Model specialization is particularly striking. BIOFM-265M achieves rank 10.4 on methylation tasks versus rank 36.7 across other categories – a specialization score of 26.3, the highest in our benchmark. Similarly, DNA-GPT-0.1B-H shows rank 10.6 on methylation versus 27.0 elsewhere (score 16.4). All four NT-v2 multi-species variants demonstrate methylation specialization (specialization scores ranging from $\Delta = 9.7$ for NT-v2-50M-MS to $\Delta = 12.9$ for NT-v2-100M-MS).

Scaling on methylation tasks is modest. Spearman correlation between model size and MCC is $\rho = 0.347$ ($p = 0.028$), weaker than the benchmark-wide $\rho = 0.565$. This suggests that methylation prediction depends more on specific sequence features captured during pretraining than on model capacity.

Tokenization effects diverge from overall patterns. $k$-mer tokenization shows clear advantages: DNA-GPT-3B-M ($k$-mer) exceeds GENOMEOCEAN-4B (BPE) by 0.136 MCC (0.367 vs. 0.231) despite smaller size. NT-v2-250M-MS ($k$-mer) outperforms GENA-LM-LARGE-T2T (BPE) by 0.132 MCC (0.339 vs. 0.207). This advantage likely reflects the importance of specific $k$-mer motifs in methylation site recognition.

Few-shot degradation is among the most severe in our benchmark: 74.7% at 10-shot (0.219 to 0.056) and 93.2% at 1-shot (0.219 to 0.015). Even top-performing models collapse: GENERATOR-EUKARYOTE-3B degrades from 0.440 to 0.097 at 10-shot and to 0.021 at 1-shot. The 10-shot ranking remains moderately correlated with full-shot (Spearman $\rho = 0.50$), with GENERATOR-EUKARYOTE-3B, DNA-GPT-3B-M, and NT-V2-50M-3MER-MS occupying the top three 10-shot positions.

### F.12.5. SPLICE SITE DETECTION

Splice site detection ($n = 7$ tasks) represents a critical test of genomic sequence understanding, as accurate splicing prediction requires recognition of complex sequence patterns spanning donor and acceptor sites. Task difficulty varies from canonical donor/acceptor classification (mean MCC $\approx 0.52$) to the reconstructed splice site challenge (mean MCC $= 0.306$). Results for the 7 splice site prediction tasks are presented in Figures 25–27.

*Figure 25.* **Few-shot performance degradation on Splice Site Detection.** For each of the 40 models, macro-average MCC across 7 splice site tasks under full-shot, 10-shot, and 1-shot regimes; models ordered by full-shot performance. The top band shows the relative drop from full-shot to 10-shot per model. Benchmark-wide mean degradation: 67.6% for 10-shot, 86.4% for 1-shot. The 10-shot regime retains discriminative signal (maximum MCC $= 0.298$ for LUCAONE), while the 1-shot regime collapses to near-random performance (maximum $= 0.107$). The 10-shot ranking diverges noticeably from full-shot (Spearman $\rho = 0.77$, top-5 overlap 2 of 5), with compact models such as MUTBERT entering the 10-shot top-5.

The scale-performance paradox manifests dramatically for splice sites. GENERATOR-EUKARYOTE-1.2B (1.2B parameters) exceeds EVO-1-131K (7B parameters) by 0.469 MCC (0.629 vs. 0.160) – among the largest pair-wise gaps observed across any task category. This result directly reflects the fundamental incompatibility between prokaryotic pretraining and eukaryotic splicing prediction: prokaryotes lack the spliceosomal machinery present in eukaryotes, rendering EVO-1's 7B parameters essentially uninformative for this task class.

Architecture comparisons reveal the largest Transformer-over-SSM advantages observed in our benchmark. OMNI-DNA-1B (Transformer-decoder) outperforms ECCDNAMAMBA (Mamba-SSM) by 0.352 MCC (0.604 vs. 0.252) under matched conditions (both 1B, multi-species, BPE), while GENOMEOCEAN-500M exceeds ECCDNAMAMBA by 0.299 MCC. These substantial gaps suggest that attention mechanisms are particularly well-suited for capturing the long-range dependencies inherent in splice site recognition.

Tokenization effects favor single-nucleotide approaches. MUTBERT (single-nucleotide) exceeds GROVER (BPE) by 0.190 MCC (0.579 vs. 0.390) under matched Transformer-encoder/human-pretraining conditions (both $\approx$ 87M), and exceeds

GENA-LM (BPE, Transformer-encoder, human) by 0.159 MCC, although the MUTBERT–GENA-LM contrast is not strictly matched on scale (86M vs. 110M). These advantages likely reflect the importance of precise positional information at splice junctions, which coarser subword tokenization may obscure.

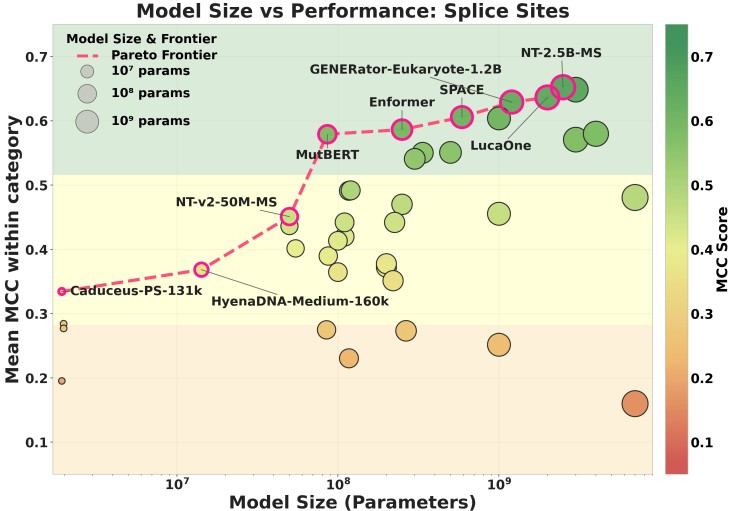

*Figure 26.* **Pareto frontier for Splice Site Detection: mean MCC vs. parameter count.** Each point represents one of the 40 genomic foundation models, with parameter count on a logarithmic x-axis and mean full-shot Splice Sites MCC on the y-axis. Marker size and color both encode MCC. The dashed line marks the Pareto frontier of best performance–size trade-offs. Scaling on this category is strong (Spearman $\rho = 0.535$, $p < 0.001$; $\rho = 0.654$ excluding prokaryotic EVO-1-131K). MUTBERT (86M, mean MCC = 0.579) is the strongest sub-100M model on this category, ranking 9th of 40 overall and sitting near the Pareto frontier alongside multi-billion-parameter models.

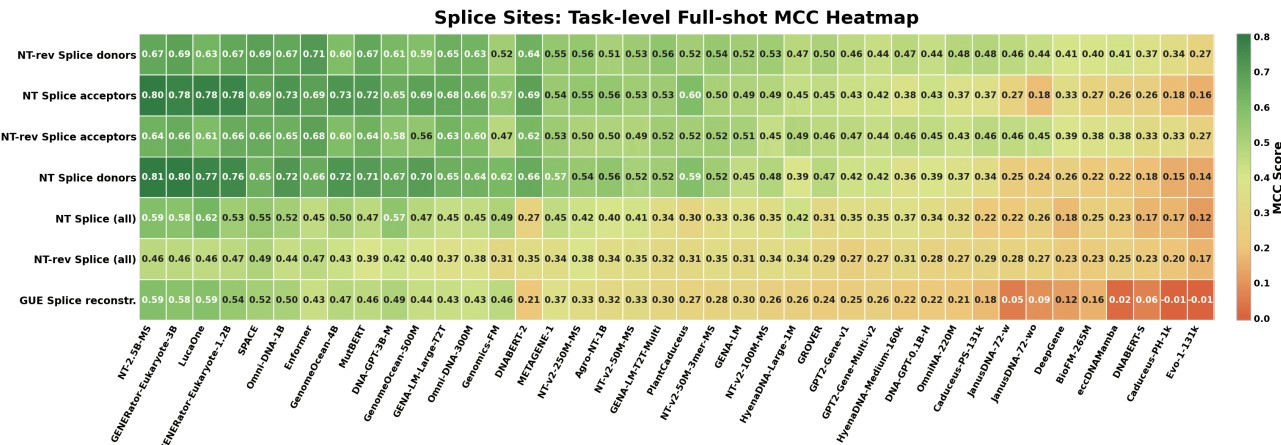

*Figure 27.* **Per-task MCC for Splice Site Detection.** Heatmap shows full-shot MCC for each of the 40 genomic foundation models on the 7 splice site tasks, with models sorted by mean Splice Sites MCC. Cell values report per-task MCC, with colors ranging from red/orange for lower scores to green for higher scores. Donor and acceptor classification tasks (NT and NT-revised sources) are consistently easier than the joint splice-sites-all task or the GUE reconstructed splice site challenge. Eukaryotic gene-focused and multi-species models dominate the top of the model ordering, with the human-mouse epigenomic-profile model SPACE in the top-5 and MUTBERT (86M) entering the top-10 despite its small scale.

Pretraining data effects are pronounced. Multi-species GENOMICS-FM exceeds microbial DNABERT-S by 0.261 MCC (0.492 vs. 0.230) – the largest data-source gap observed across task categories. Eukaryotic gene-focused GENERATOR-EUKARYOTE-3B outperforms multi-species DNA-GPT-3B-M by 0.078 MCC (0.648 vs. 0.571) under matched Transformer-decoder/$k$-mer conditions (both 3B), confirming the advantage of gene-centric pretraining for splicing-related tasks.

F.12.6. LONG NON-CODING RNA CLASSIFICATION

The lncRNA category ($n = 6$ tasks) exclusively comprises plant species classification problems spanning *Glycine max* (soybean), *Manihot esculenta* (cassava), *Sorghum bicolor* (sorghum), *Solanum lycopersicum* (tomato), *Triticum aestivum* (wheat), and *Zea mays* (maize). This composition makes lncRNA tasks the primary testbed for plant-specific transfer learning in our benchmark. Results for the 6 lncRNA prediction tasks are presented in Figures 28–30.

Plant-specialized models demonstrate clear advantages. PLANTCADUCEUS achieves average per-task rank 8.0 on lncRNA tasks versus 27.0 across the remaining 94 tasks, yielding a specialization gap of $\Delta = 19.0$ – second only to BIOFM-265M's methylation specialization ($\Delta = 26.3$). AGRO-NT-1B shows a similar but smaller pattern ($\Delta = 10.1$). These results validate the importance of taxonomically-aligned pretraining for plant genomics applications.

Human-trained models exhibit substantial negative transfer. The best human-trained model (MUTBERT, MCC $= 0.260$) substantially underperforms plant-trained alternatives (PLANTCADUCEUS, MCC $= 0.357$). Across all human-trained models, mean lncRNA MCC is only $0.157$ compared to $0.347$ for plant-trained models – a gap of $0.190$ MCC representing $121\%$ relative improvement. The negative transfer is particularly striking for BIOFM-265M, the only model with a negative mean lncRNA MCC at full-shot ($-0.018$), indicating worse-than-random performance on plant lncRNA classification despite the model's strong methylation specialization.

Notably, the overall lncRNA leader LUCAONE (multi-species, mean MCC $= 0.508$) achieves the best per-category performance, suggesting that sufficiently diverse pretraining can compensate for the lack of plant-specific data. LUCAONE's advantage over PLANTCADUCEUS is consistent across all six individual plant species (per-task $\Delta$ ranging from $+0.07$ on *M. esculenta* to $+0.27$ on *S. bicolor*), indicating that broad taxonomic exposure can outweigh plant-specific pretraining on this category even at the per-task level.

*Figure 28.* **Few-shot performance degradation on lncRNA Classification.** For each of the 40 models, macro-average MCC across 6 plant lncRNA tasks under full-shot, 10-shot, and 1-shot regimes; models ordered by full-shot performance. The top band shows the relative drop from full-shot to 10-shot per model. Benchmark-wide mean degradation: 79.8% for 10-shot, 91.3% for 1-shot. Both regimes collapse to near-random performance (10-shot maximum MCC $= 0.207$ for PLANTCADUCEUS; 1-shot maximum $= 0.055$). Despite the collapse, the 10-shot ranking remains strongly correlated with full-shot (Spearman $\rho = 0.90$, top-5 overlap 4 of 5); notably, the plant-specialized PLANTCADUCEUS rises from rank 7 at full-shot to rank 1 at 10-shot.

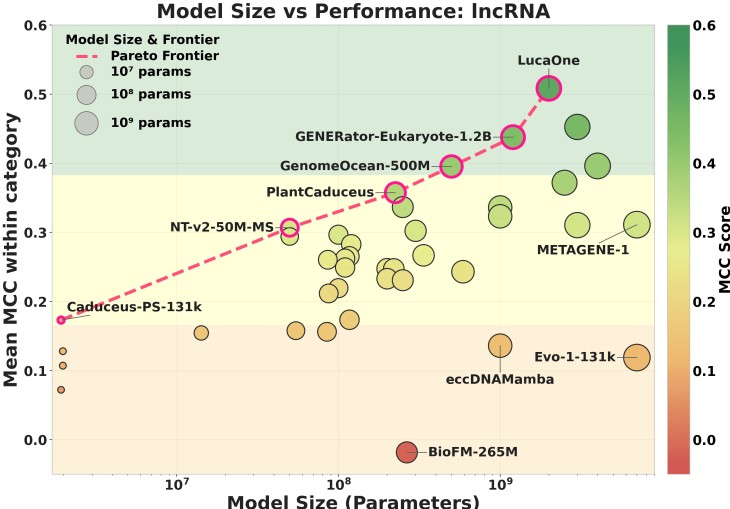

*Figure 29.* **Pareto frontier for lncRNA Classification: mean MCC vs. parameter count.** Each point represents one of the 40 genomic foundation models, with parameter count on a logarithmic x-axis and mean full-shot lncRNA MCC on the y-axis. Marker size and color both encode MCC. The dashed line marks the Pareto frontier of best performance–size trade-offs. Scaling on this category is moderate (Spearman $\rho = 0.565$, $p < 0.001$). The frontier reflects the plant-specific nature of the tasks: PLANTCADUCEUS (225M, MCC $= 0.357$) and AGRO-NT-1B (1B, MCC $= 0.336$) sit near the frontier despite modest scale, alongside multi-species models with much larger parameter counts. Among sub-100M models, NT-V2-50M-MS (50M, MCC $= 0.307$) achieves the highest lncRNA MCC.

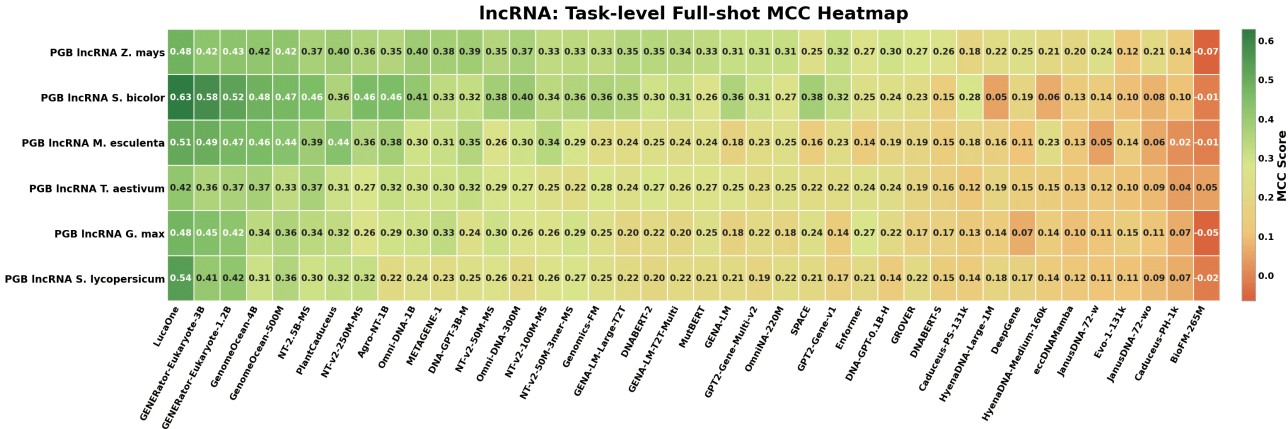

*Figure 30.* **Per-task MCC for lncRNA Classification.** Heatmap shows full-shot MCC for each of the 40 genomic foundation models on the 6 plant lncRNA classification tasks, with models sorted by mean lncRNA MCC. Cell values report per-task MCC, with colors ranging from red/orange for lower scores to green for higher scores. Multi-species (LUCAONE) and eukaryotic gene-focused (GENERATOR) models dominate the top of the ordering, with plant-specialized models (PLANTCADUCEUS, AGRO-NT-1B) close behind. Human-pretrained models cluster near the bottom.

Few-shot degradation is severe (79.8% relative drop at 10-shot, 91.3% at 1-shot), with all models collapsing to near-random performance (10-shot maximum MCC $= 0.207$). The 10-shot ranking remains strongly correlated with full-shot (Spearman $\rho = 0.90$), but the plant-specialized PLANTCADUCEUS rises from full-shot rank 7 to 10-shot rank 1, while LUCAONE drops to rank 2 – indicating that plant-aligned pretraining provides somewhat greater robustness under data scarcity even when not yielding the highest full-shot score.

F.12.7. MOUSE ENHANCER PREDICTION

Mouse Enhancer prediction ($n = 5$ tasks) evaluates cross-species regulatory element recognition, providing insight into transfer learning from human-centric or multi-species pretraining to a related but distinct mammalian system. Task difficulty varies substantially across the 5 tasks: task 1 is the easiest (mean MCC = 0.680 across the 40 models) and task 4 is the hardest (mean MCC = 0.308). Results for the 5 mouse enhancer prediction tasks are presented in Figures 31–33.

Scaling on Mouse Enhancers is moderate and statistically significant (Spearman $\rho = 0.474$, $p = 0.002$; $\rho = 0.580$ excluding the prokaryotic EVO-1-131K). Models with at least 1B parameters average 0.520 MCC versus 0.438 for models below 200M – a gap of 0.082 MCC, slightly larger than the benchmark-wide +0.064 tier gap.

Architecture comparisons reveal substantial Transformer-over-SSM advantages. OMNI-DNA-1B (Transformer-decoder) exceeds ECCDNAMAMBA (Mamba-SSM) by 0.305 MCC (0.675 vs. 0.370) under matched conditions (both 1B, multi-species, BPE) – the second-largest Transformer-over-SSM gap observed across task categories (after Splice Sites at 0.352). A within-Transformer encoder-vs-decoder contrast on the same category shows GENA-LM-LARGE-T2T (336M, encoder) outperforming OMNINA-220M (220M, decoder) by 0.284 MCC (0.615 vs. 0.332), although the contrast is not strictly matched on scale. This direction is opposite to the encoder-vs-decoder result we observe on Histone Modifications and is consistent with the broader observation (Section 4) that the encoder–decoder distinction is task- and setting-dependent.

Pretraining data effects follow expected patterns with notable magnitude. The controlled GENA-LM comparison shows multi-species outperforming human-only by 0.067 MCC (GENA-LM-T2T-MULTI = 0.548 vs. GENA-LM = 0.480). Eukaryotic gene-focused GENERATOR-EUKARYOTE-3B exceeds multi-species DNA-GPT-3B-M by 0.124 MCC (0.636 vs. 0.512) under matched Transformer-decoder/$k$-mer/3B conditions, while multi-species GENOMICS-FM outperforms microbial DNABERT-S by 0.144 MCC under matched Transformer-encoder conditions ($\approx$120M each).

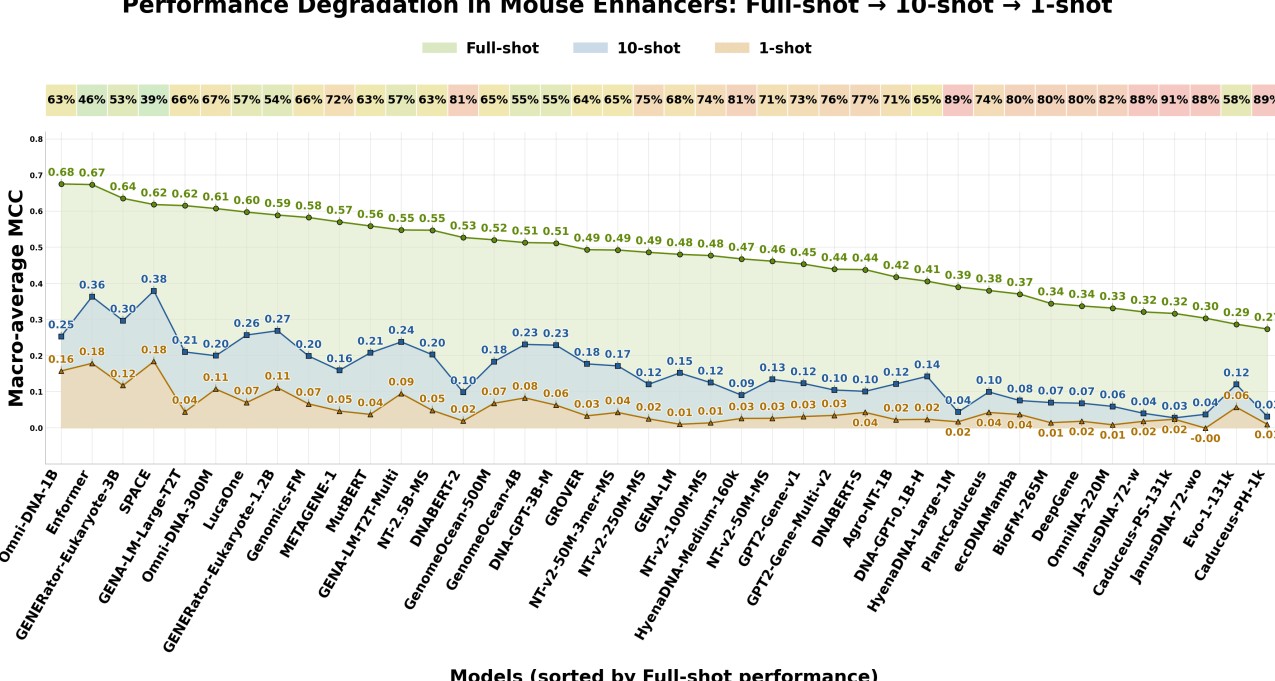

*Figure 31.* **Few-shot performance degradation on Mouse Enhancer Prediction.** For each of the 40 models, macro-average MCC across 5 mouse enhancer tasks under full-shot, 10-shot, and 1-shot regimes; models ordered by full-shot performance. The top band shows the relative drop from full-shot to 10-shot per model. Benchmark-wide mean degradation: 67.4% for 10-shot, 89.2% for 1-shot. Unlike on the (human-centric) Enhancers category, human-mouse epigenomic-profile models retain their advantage under few-shot conditions: SPACE rises from full-shot rank 4 to 10-shot rank 1 (10-shot MCC = 0.379), and ENFORMER holds rank 2 at both regimes (10-shot MCC = 0.363). Full-shot and 10-shot rankings remain strongly correlated (Spearman $\rho = 0.90$, top-5 overlap 3 of 5).

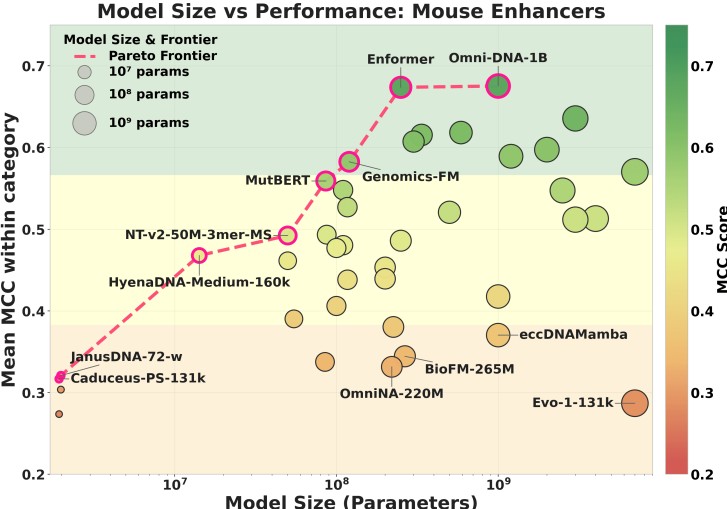

*Figure 32.* **Pareto frontier for Mouse Enhancer Prediction: mean MCC vs. parameter count.** Each point represents one of the 40 genomic foundation models, with parameter count on a logarithmic x-axis and mean full-shot Mouse Enhancer MCC on the y-axis. Marker size and color both encode MCC. The dashed line marks the Pareto frontier of best performance–size trade-offs. Scaling on this category is moderate (Spearman $\rho = 0.474$, $p = 0.002$; $\rho = 0.580$ excluding prokaryotic Evo-1-131K). Human-mouse epigenomic-profile models sit prominently on the frontier (ENFORMER, 250M, MCC = 0.674; SPACE, 589M, MCC = 0.618), and MUTBERT (86M, mean MCC = 0.559) is the strongest sub-100M model on this category.

Human-mouse epigenomic profile models show expected advantages given their training data includes mouse regulatory elements. ENFORMER (mean MCC = 0.674, rank 2 of 40) and SPACE (mean MCC = 0.618, rank 4 of 40) rank among the category leaders, with ENFORMER achieving the notable result of outperforming EVO-1-131K (7B parameters) by 0.387 MCC despite being $28\times$ smaller.

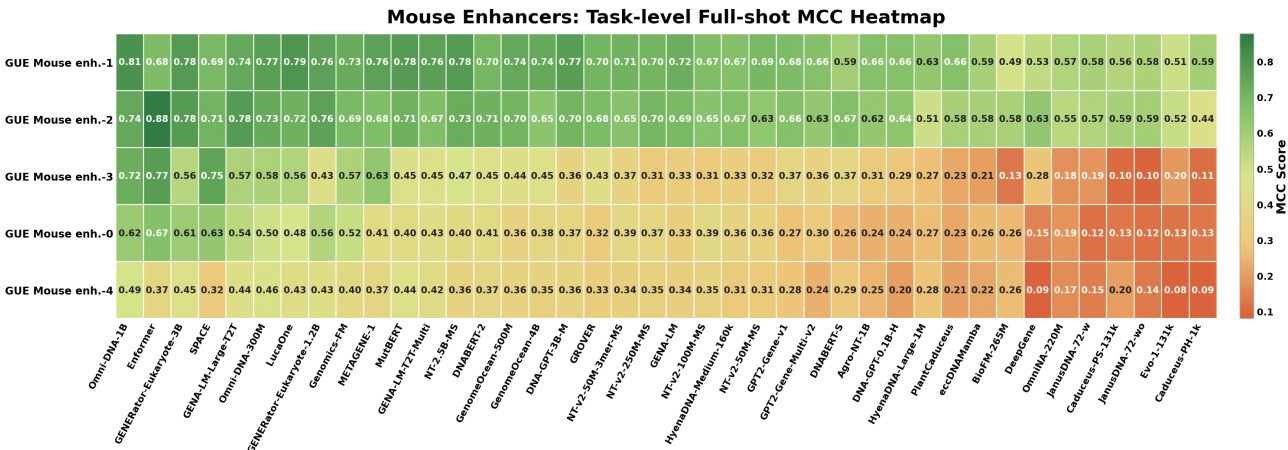

*Figure 33.* **Per-task MCC for Mouse Enhancer Prediction.** Heatmap shows full-shot MCC for each of the 40 genomic foundation models on the 5 mouse enhancer tasks (GUE mouse 0 through 4), with models sorted by mean Mouse Enhancer MCC. Cell values report per-task MCC, with colors ranging from red/orange for lower scores to green for higher scores. Tasks 1 and 2 are uniformly easier across models (mean MCC > 0.65) than tasks 0, 3, and 4 (mean MCC < 0.39). Human-mouse epigenomic-profile models (ENFORMER, SPACE), multi-species models (OMNI-DNA-1B), and eukaryotic gene-focused models (GENERATOR) dominate the top of the model ordering.

F.12.8. TRANSCRIPTION FACTOR BINDING

TF Binding prediction ($n = 5$ tasks) represents a critical challenge in regulatory genomics, requiring recognition of sequence motifs that mediate protein-DNA interactions. This category exhibits one of the most striking scale-performance paradoxes in our benchmark. Results for the 5 transcription factor binding prediction tasks are presented in Figures 34–36.

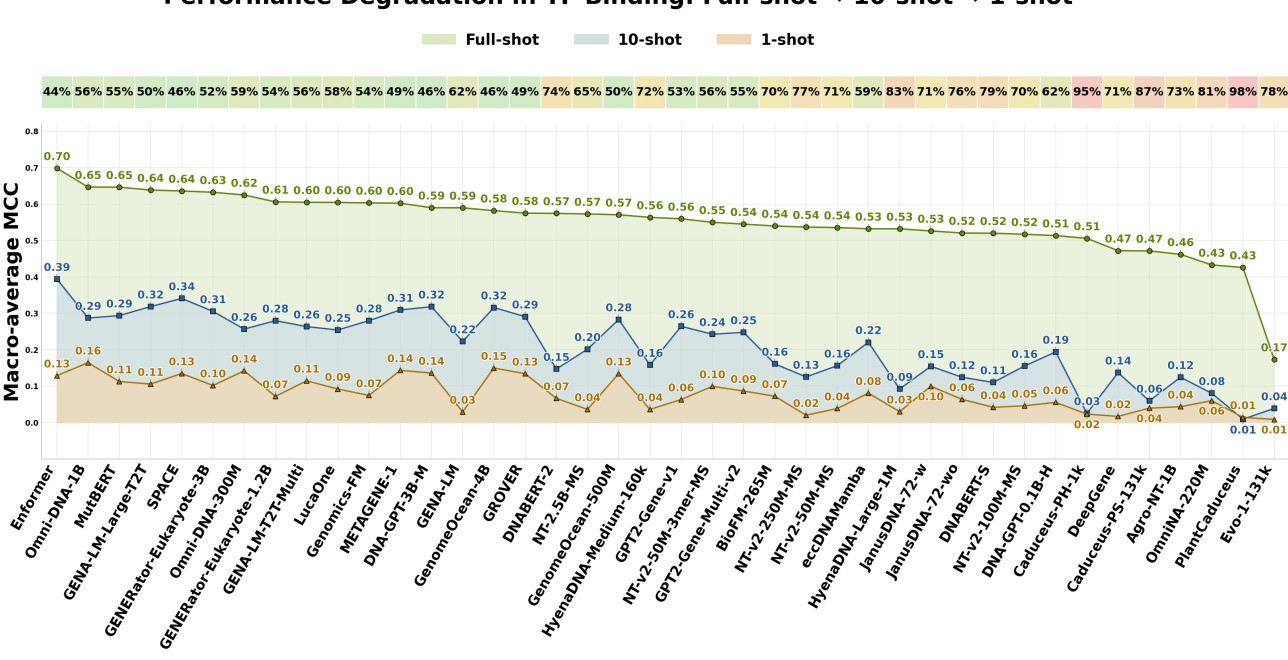

*Figure 34.* **Few-shot performance degradation on TF Binding.** For each of the 40 models, macro-average MCC across 5 TF binding tasks under full-shot, 10-shot, and 1-shot regimes; models ordered by full-shot performance. The top band shows the relative drop from full-shot to 10-shot per model. Benchmark-wide mean degradation: 62.6% for 10-shot, 85.9% for 1-shot. The 10-shot regime retains discriminative signal (maximum MCC = 0.394 for ENFORMER), while 1-shot collapses to near-random performance (maximum = 0.164). Full-shot and 10-shot rankings remain strongly correlated (Spearman $\rho = 0.88$, top-5 overlap 3 of 5); notably, ENFORMER retains rank 1 at both regimes.

Scaling on TF Binding is modest but statistically detectable (Spearman $\rho = 0.358$, $p = 0.023$; $\rho = 0.463$ excluding the prokaryotic EVO-1-131K). Models with at least 1B parameters average 0.546 MCC versus 0.546 for models below 200M – effectively no tier gap at the extremes, in contrast to the benchmark-wide +0.064 gap. This flat-at-the-extremes relationship reflects in large part the catastrophic failure of EVO-1-131K (7B parameters, mean MCC = 0.173, rank 40 of 40), which pulls the $\geq$ 1B group mean downward.

The scale paradox reaches its apex for TF binding: ENFORMER (250M parameters, mean MCC = 0.698, rank 1 of 40) exceeds EVO-1-131K by 0.525 MCC – the largest performance gap between any small-vs-large model pair across our entire benchmark. This $28\times$ size disadvantage yielding $4\times$ better performance starkly illustrates the dominance of pretraining data alignment over model scale.

Architecture comparisons favor Transformer-encoder variants on this category. GENA-LM-LARGE-T2T (encoder, 336M) outperforms OMNINA-220M (decoder, 220M) by 0.206 MCC (0.638 vs. 0.433), and the smaller GENA-LM-T2T-MULTI (encoder, 110M) also outperforms OMNINA-220M (decoder, 220M) by 0.172 MCC. The Transformer-over-SSM gap remains substantial under matched conditions: OMNI-DNA-1B exceeds ECCDNAMAMBA by 0.114 MCC (0.647 vs. 0.532) at matched 1B/multi-species/BPE. These pairwise contrasts are consistent with the broader observation (Section 4) that the encoder–decoder ranking on this benchmark is task- and setting-dependent; for TF Binding the encoder direction holds across both pairs we examined.

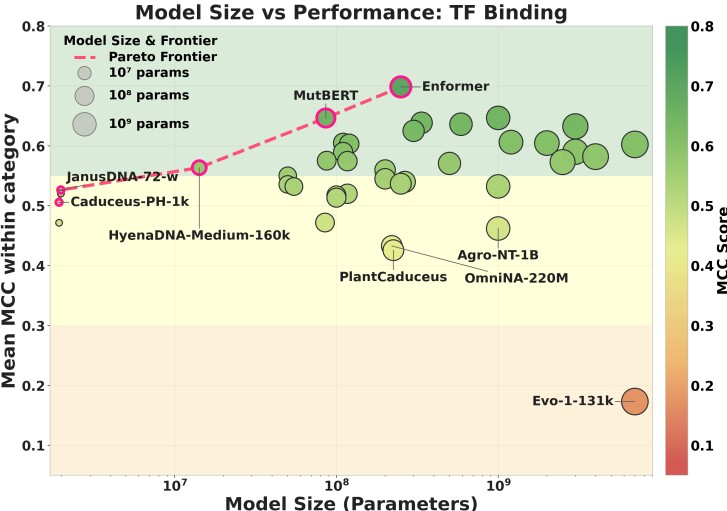

*Figure 35.* **Pareto frontier for TF Binding: mean MCC vs. parameter count.** Each point represents one of the 40 genomic foundation models, with parameter count on a logarithmic x-axis and mean full-shot TF Binding MCC on the y-axis. Marker size and color both encode MCC. The dashed line marks the Pareto frontier of best performance–size trade-offs. Scaling on this category is modest (Spearman $\rho = 0.358$, $p = 0.023$; $\rho = 0.463$ excluding prokaryotic EVO-1-131K), with models exceeding 1B parameters averaging essentially the same MCC as models below 200M (0.546 vs. 0.546). MUTBERT (86M, mean MCC = 0.646) is the strongest sub-100M model on this category and ranks 3rd of 40 overall, between ENFORMER (250M) and OMNI-DNA-1B.

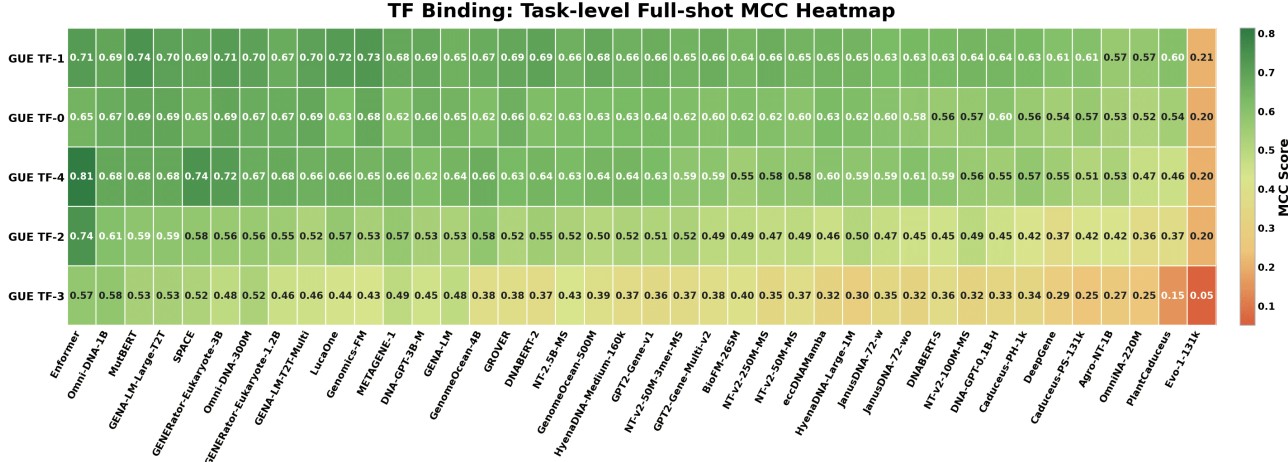

*Figure 36.* **Per-task MCC for TF Binding.** Heatmap shows full-shot MCC for each of the 40 genomic foundation models on the 5 TF binding tasks (GUE human TF 0 through 4), with models sorted by mean TF Binding MCC. Cell values report per-task MCC, with colors ranging from red/orange for lower scores to green for higher scores. Task difficulty varies from GUE TF-3 (mean MCC = 0.385, hardest) to GUE TF-1 (mean MCC = 0.652, easiest). Human-mouse epigenomic-profile models (ENFORMER, SPACE), Transformer-encoder models (GENA-LM-LARGE-T2T, MUTBERT), and multi-species Transformer-decoder models (OMNI-DNA-1B) dominate the top of the model ordering.

Human-mouse epigenomic profile models again demonstrate specialized strength. ENFORMER achieves the highest mean MCC on the category (0.698, rank 1 of 40) and SPACE follows closely (0.636, rank 5 of 40); on per-task average ranks these correspond to 4.0 and 7.0 respectively, reflecting consistent placement among the strongest models across the 5 TF binding tasks (Enformer in top-5 on 4 of 5 tasks; SPACE in top-11 on all 5). This performance reflects the direct relevance of TF binding events to the chromatin-state signals present in human-mouse epigenomic profile pretraining (Section F.10).

F.12.9. SPECIES CLASSIFICATION

Species Classification tasks ($n = 3$ tasks) evaluate a model's ability to distinguish genomic sequences from different organisms, testing whether pretraining captures species-specific sequence signatures. The category consists of one cross-kingdom human/worm classification (GB Human-or-worm) and two fine-grained taxonomic tasks (GUE Fungi-20 and GUE Virus-40). Task difficulty varies substantially: the cross-kingdom task achieves mean MCC $= 0.857$ across the 40 models, while the GUE Virus-40 task is the hardest at $0.323$. Results for the 3 species classification tasks are presented in Figures 37–39.

Scaling on Species Classification is among the weakest in our benchmark (Spearman $\rho = 0.304$, $p = 0.056$, marginally non-significant; $\rho = 0.406$, $p = 0.010$ when excluding EVO-1-131K). Models exceeding 1B parameters average $0.657$ MCC versus $0.606$ for models below 200M ($+0.050$ gap, slightly below the benchmark-wide $+0.064$). This compressed scaling reflects in part the strong performance of small multi-species Nucleotide Transformer variants on this category, with NT-V2-50M-3MER-MS (50M) achieving $0.734$ MCC – within $0.030$ of the top full-shot model GENOMEOCEAN-4B (4B, MCC $= 0.762$) at $1/80$ the parameter count.

The scale paradox is particularly visible for species classification, with 126 pairwise cases where smaller models outperform larger counterparts by at least $5\times$ size ratio. NT-V2-50M-3MER-MS (50M) exceeds EVO-1-131K (7B) by $0.449$ MCC despite a $140\times$ size disadvantage – one of the largest such gaps observed in our benchmark, comparable in magnitude to the ENFORMER–EVO-1-131K contrast on TF Binding ($0.525$; Section F.10).

Architecture comparisons favor Transformer-encoder variants. NT-V2-250M-MS (encoder) outperforms GPT2-GENE-MULTI-V2 (decoder) by $0.189$ MCC ($0.747$ vs. $0.559$) under matched multi-species pretraining and $k$-mer tokenization (250M vs. 200M; the contrast is not strictly matched on scale). The same direction holds for the smaller NT-V2-100M-MS, which outperforms GPT2-GENE-MULTI-V2 by $0.183$ MCC at half the parameter count. As elsewhere in the benchmark (Section 4), the encoder–decoder ranking is task- and setting-dependent; for Species Classification the encoder direction holds across both pairs we examined.

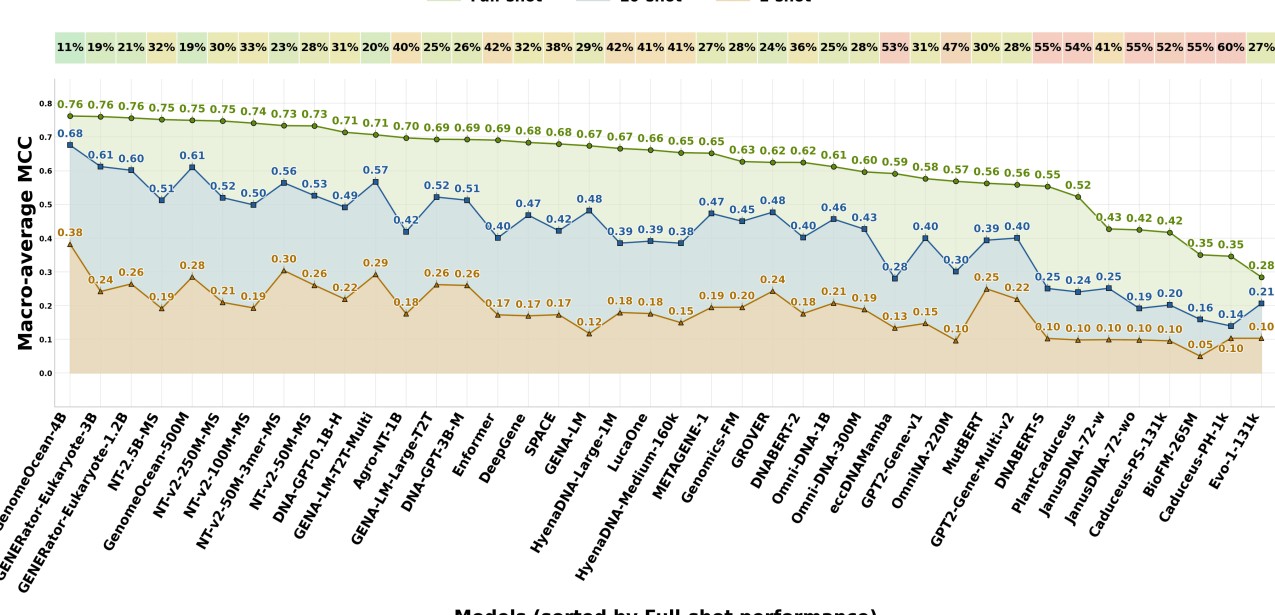

*Figure 37.* **Few-shot performance degradation on Species Classification.** For each of the 40 models, macro-average MCC across 3 species classification tasks under full-shot, 10-shot, and 1-shot regimes; models ordered by full-shot performance. The top band shows the relative drop from full-shot to 10-shot per model. Benchmark-wide mean degradation: $33.0\%$ for 10-shot, $69.9\%$ for 1-shot – among the mildest few-shot degradation of any task category in our benchmark. Both 10-shot (maximum MCC $= 0.676$ for GENOMEOCEAN-4B) and 1-shot (maximum $= 0.382$, also GENOMEOCEAN-4B) regimes retain substantial discriminative signal, indicating that species classification is relatively robust to data scarcity. Full-shot and 10-shot rankings are very strongly correlated (Spearman $\rho = 0.91$, top-5 overlap 4 of 5).

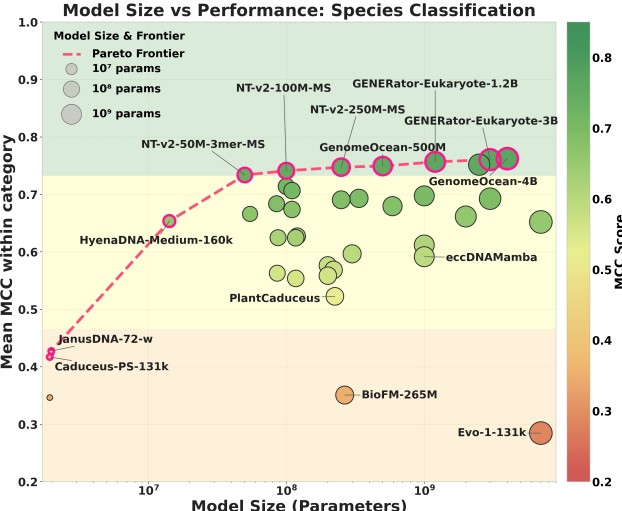

*Figure 38.* **Pareto frontier for Species Classification: mean MCC vs. parameter count.** Each point represents one of the 40 genomic foundation models, with parameter count on a logarithmic x-axis and mean full-shot Species Classification MCC on the y-axis. Marker size and color both encode MCC. The dashed line marks the Pareto frontier of best performance–size trade-offs. Scaling on this category is among the weakest in our benchmark (Spearman $\rho = 0.304$, $p = 0.056$; $\rho = 0.406$, $p = 0.010$ excluding the prokaryotic EVO-1-131K). NT-V2-50M-3MER-MS (50M, mean MCC = 0.734) is the strongest sub-100M model on this category, sitting near the Pareto frontier alongside multi-billion-parameter generalists.

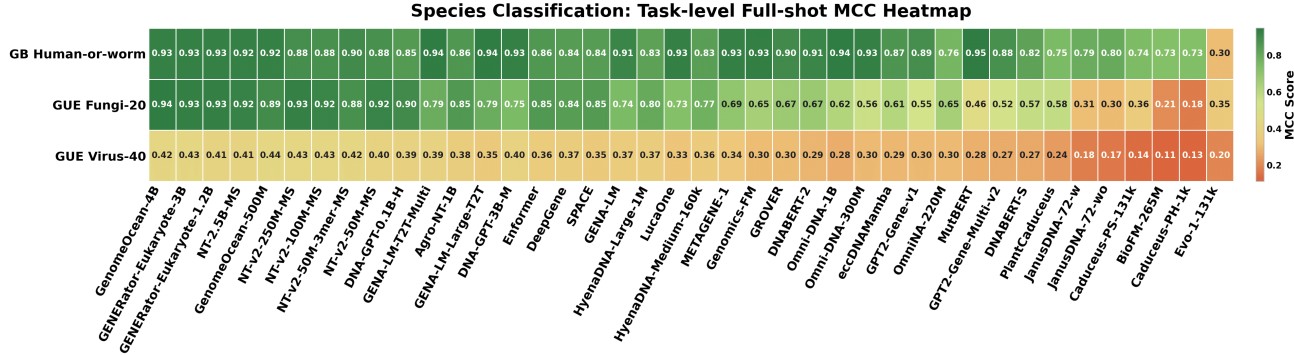

*Figure 39.* **Per-task MCC for Species Classification.** Heatmap shows full-shot MCC for each of the 40 genomic foundation models on the 3 species classification tasks, with models sorted by mean Species Classification MCC. Cell values report per-task MCC, with colors ranging from red/orange for lower scores to green for higher scores. The cross-kingdom GB Human-or-worm task is uniformly easy across models (mean MCC = 0.857), while the fine-grained GUE Virus-40 task is uniformly hard (0.323). Multi-species decoder models (GENOMEOCEAN-4B, GENERATOR-EUKARYOTE-3B) and Nucleotide Transformer encoder models cluster at the top of the model ordering.

Pretraining data effects are pronounced. Multi-species training strongly outperforms human-only: DNA-GPT-0.1B-H (multi-species, 100M, Transformer-decoder, $k$-mer) exceeds GPT2-GENE-V1 (human, 200M, same architecture and tokenization) by 0.137 MCC (0.714 vs. 0.576), despite the multi-species model being half the size. Multi-species also outperforms microbial: NT-V2-100M-MS outperforms DNABERT-S (multi-species-microbial) by 0.187 MCC (0.741 vs. 0.554) under matched Transformer-encoder/$k$-mer conditions (100M vs. 117M). These patterns logically reflect that species classification requires exposure to diverse taxonomic sequences during pretraining.

The Nucleotide Transformer v2 multi-species family shows clear specialization for this category, with all four NT-v2 multi-species variants exhibiting positive specialization (NT-V2-100M-MS: per-task rank 10.7 on Species Classification vs. 19.9 elsewhere, $\Delta = 9.2$; NT-V2-250M-MS: $\Delta = 7.7$; NT-V2-50M-3MER-MS: $\Delta = 5.9$; NT-V2-50M-MS: $\Delta = 5.8$).

F.12.10. REGULATORY ELEMENT PREDICTION

Regulatory element prediction ($n = 2$ tasks: GB Ensembl regulatory and GB OCR Ensembl) focuses on identifying genomic regions involved in transcriptional control. This category reveals one of the largest domain-specific advantages in our benchmark, with two human-mouse epigenomic-profile models substantially outperforming all generalists. Results for the 2 regulatory element prediction tasks are presented in Figures 40–42.

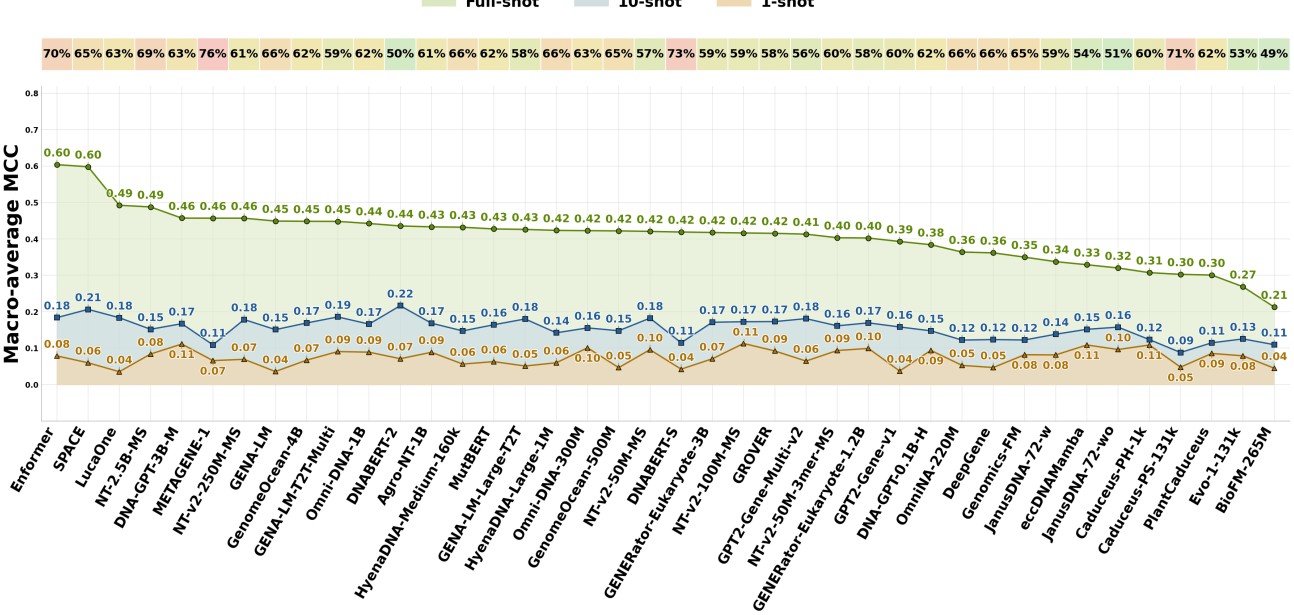

*Figure 40.* **Few-shot performance degradation on Regulatory Element Prediction.** For each of the 40 models, macro-average MCC across 2 regulatory element tasks under full-shot, 10-shot, and 1-shot regimes; models ordered by full-shot performance. The top band shows the relative drop from full-shot to 10-shot per model. Benchmark-wide mean degradation: 62.1% for 10-shot, 81.9% for 1-shot. The 10-shot ranking differs moderately from full-shot (Spearman $\rho = 0.58$, top-5 overlap 3 of 5): ENFORMER drops from full-shot rank 1 (MCC = 0.604) to 10-shot rank 4 (MCC = 0.184), while SPACE retains its top-2 position (full rank 2 → 10-shot rank 2, MCC = 0.206). Both 10-shot (maximum MCC = 0.216 for DNABERT-2) and 1-shot (maximum = 0.113) regimes collapse to near-random performance across most models.

ENFORMER and SPACE, pretrained on human-mouse epigenomic profiles, dominate this category with mean MCC of 0.604 and 0.598 respectively, substantially exceeding all other models. The next-best performer (LUCAONE, multi-species, 2B, MCC = 0.492) trails by over 0.11 MCC. This exceptional performance directly reflects the alignment between pretraining data (regulatory chromatin-state signals across many cell types) and task requirements (regulatory element identification).

Scaling on Regulatory is modest but statistically significant (Spearman $\rho = 0.378$, $p = 0.016$; $\rho = 0.481$, $p = 0.002$ excluding the prokaryotic EVO-1-131K). Models exceeding 1B parameters average 0.421 MCC versus 0.392 for models below 200M (+0.030 gap, well below the benchmark-wide +0.064). The compressed tier gap reflects the dominance of medium-scale specialized models (ENFORMER at 250M, SPACE at 589M) over larger general-purpose alternatives, underscoring that domain-appropriate pretraining trumps scale on this category.

An unusual pattern emerges in pretraining-data comparisons: DNABERT-S (multi-species microbial) outperforms GENOMICS-FM (multi-species, $\approx$120M each, matched Transformer-encoder) by 0.069 MCC (0.419 vs. 0.350) – the only category in our benchmark where this reversal occurs (in all other 12 categories GENOMICS-FM leads this pair). This reversal may indicate that certain regulatory patterns are shared across prokaryotic and eukaryotic systems, or that DNABERT-S's training procedure captures generalizable regulatory features despite its taxonomic focus; given $n = 2$ tasks and a single pairwise contrast, however, this remains an observation rather than a strong claim.

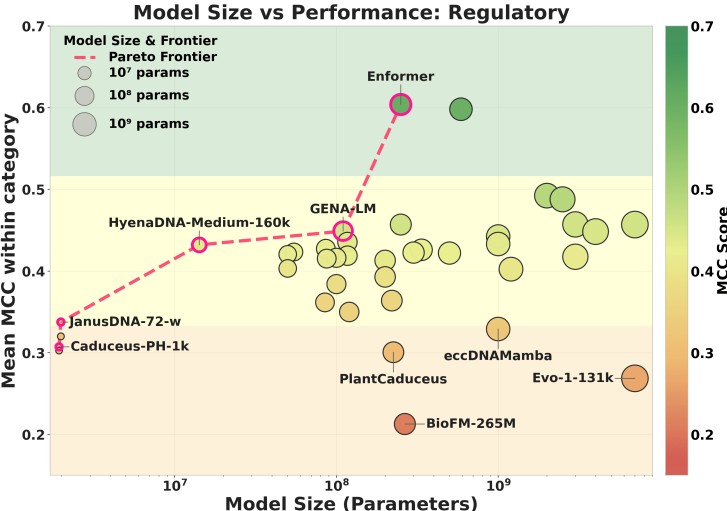

*Figure 41.* **Pareto frontier for Regulatory Element Prediction: mean MCC vs. parameter count.** Each point represents one of the 40 genomic foundation models, with parameter count on a logarithmic x-axis and mean full-shot Regulatory MCC on the y-axis. Marker size and color both encode MCC. The dashed line marks the Pareto frontier of best performance–size trade-offs. Scaling on this category is modest (Spearman $\rho = 0.378$, $p = 0.016$; $\rho = 0.481$, $p = 0.002$ excluding prokaryotic EVO-1-131K). Human-mouse epigenomic-profile models occupy the frontier at moderate sizes (ENFORMER, 250M, MCC = 0.604; SPACE, 589M, MCC = 0.598), outperforming all multi-billion-parameter generalists. HYENADNA-MEDIUM (14M, mean MCC = 0.432) is the strongest sub-100M model on this category.

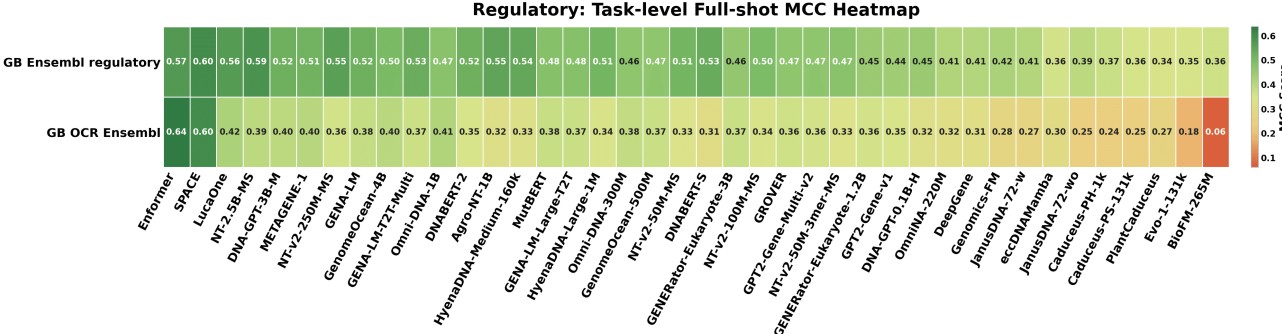

*Figure 42.* **Per-task MCC for Regulatory Element Prediction.** Heatmap shows full-shot MCC for each of the 40 genomic foundation models on the 2 regulatory element tasks, with models sorted by mean Regulatory MCC. Cell values report per-task MCC, with colors ranging from red/orange for lower scores to green for higher scores. The GB Ensembl regulatory task is easier (mean MCC = 0.471) than GB OCR Ensembl (mean MCC = 0.344). Human-mouse epigenomic-profile models (ENFORMER, SPACE) dominate the top of the model ordering, with multi-species models (LUCAONE, NT-2.5B-MS, DNA-GPT-3B-M) following.

GENA-LM shows notable specialization for regulatory tasks, achieving per-task rank 10.5 on Regulatory versus 20.0 across the remaining 98 tasks (specialization $\Delta = 9.5$). The specialization is specific to the human-pretrained GENA-LM base variant: the multi-species GENA-LM-T2T-MULTI shows weaker specialization ($\Delta = 5.8$) and the larger multi-species GENA-LM-LARGE-T2T shows negative specialization ($\Delta = -3.2$). The pattern is consistent with the concentration of well-annotated regulatory elements in human genomic resources, although with $n = 2$ tasks the effect should be interpreted cautiously.

## F.12.11. VIRUS AND PHAGE DETECTION

Virus/Phage detection ($n = 2$ tasks: GUE Phage fragments and GUE COVID variants) presents a distinctive prediction problem involving recognition of viral genomic signatures. Task difficulty varies dramatically: phage fragment classification achieves 0.633 mean MCC while COVID variant prediction proves challenging at 0.200 MCC. Results for the 2 virus and phage prediction tasks are presented in Figures 43–45.

*Figure 43.* **Few-shot performance degradation on Virus/Phage Detection.** For each of the 40 models, macro-average MCC across 2 virus/phage tasks under full-shot, 10-shot, and 1-shot regimes; models ordered by full-shot performance. The top band shows the relative drop from full-shot to 10-shot per model. Benchmark-wide mean degradation: 71.3% for 10-shot, 93.5% for 1-shot – the largest 1-shot degradation of any task category in our benchmark. The 10-shot maximum is 0.377 for GENOMEOCEAN-4B; 1-shot maximum is 0.081. Full-shot and 10-shot rankings remain strongly correlated (Spearman $\rho = 0.83$, top-5 overlap 4 of 5).

Scaling is statistically significant for this category (Spearman $\rho = 0.435$, $p = 0.005$), with the GENOMEOCEAN models achieving strong performance: GENOMEOCEAN-4B leads the category (MCC $= 0.697$, rank 1 of 40) and GENOMEOCEAN-500M follows at rank 2 (MCC $= 0.657$).

A distinctive pattern emerges for this category: human-trained models show competitive or superior performance compared to multi-species alternatives. GPT2-GENE-V1 (human) outperforms both DNA-GPT-0.1B-H (multi-species) by 0.114 MCC and GPT2-GENE-MULTI-V2 (multi-species) by 0.049 MCC. Similarly, GROVER (human) exceeds DNABERT-2 (multi-species) by 0.040 MCC. This pattern likely reflects the predominance of human-associated viral sequences in human genomic training data, providing relevant exposure to viral integration sites and endogenous retroviral elements.

Architecture comparisons reveal large Transformer advantages. GENOMEOCEAN-500M (Transformer-decoder) exceeds ECCDNAMAMBA (Mamba-SSM, 1B) by 0.355 MCC (0.657 vs. 0.302) despite being half the size, among the largest architectural gaps observed. GENA-LM-LARGE-T2T (Transformer-encoder) outperforms OMNINA-220M (Transformer-decoder) by 0.180 MCC (0.569 vs. 0.389).

Tokenization effects show BPE advantages over single-nucleotide approaches, reversing patterns observed for splice sites. GROVER (BPE) exceeds MUTBERT (single-nucleotide) by 0.209 MCC (0.532 vs. 0.323), and GENA-LM (BPE) shows similar advantage ($+0.157$ MCC). This pattern suggests that viral sequence recognition benefits from the longer-range patterns captured by subword tokenization.

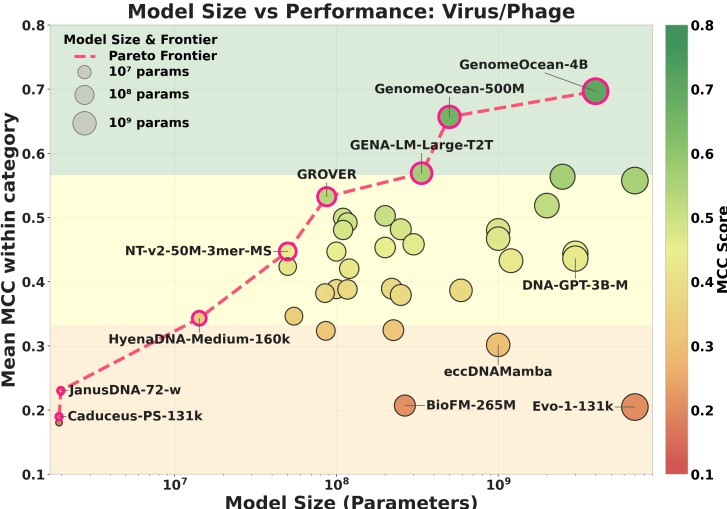

*Figure 44.* **Pareto frontier for Virus/Phage Detection: mean MCC vs. parameter count.** Each point represents one of the 40 genomic foundation models, with parameter count on a logarithmic x-axis and mean full-shot Virus/Phage MCC on the y-axis. Marker size and color both encode MCC. The dashed line marks the Pareto frontier of best performance–size trade-offs. The GENOMEOCEAN multi-species decoders dominate the frontier (GENOMEOCEAN-4B, 4B, MCC = 0.697; GENOMEOCEAN-500M, 500M, MCC = 0.657). GROVER (87M, mean MCC = 0.532) is the strongest sub-100M model on this category.

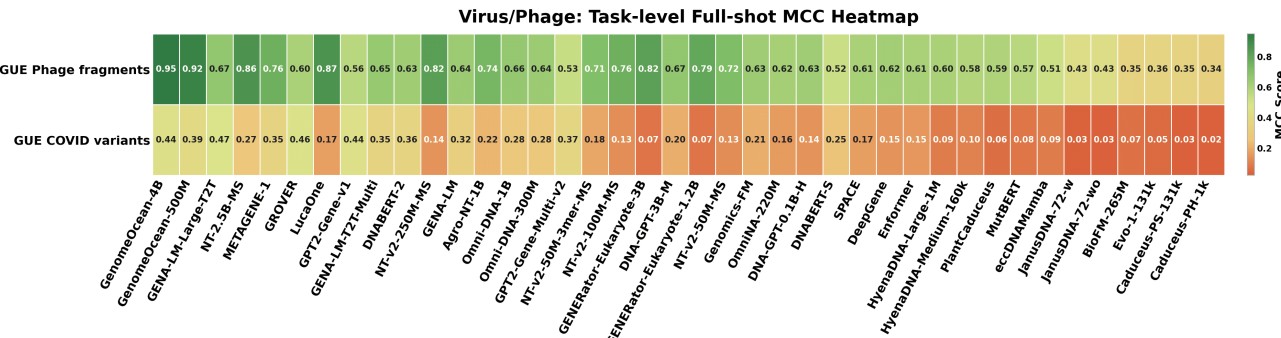

*Figure 45.* **Per-task MCC for Virus/Phage Detection.** Heatmap shows full-shot MCC for each of the 40 genomic foundation models on the 2 virus/phage tasks, with models sorted by mean Virus/Phage MCC. Cell values report per-task MCC, with colors ranging from red/orange for lower scores to green for higher scores. GUE Phage fragments (mean MCC = 0.633) is substantially easier than GUE COVID variants (0.200). Multi-species decoder models (GENOMEOCEAN-4B, GENOMEOCEAN-500M) and Transformer-encoder models dominate the top of the ordering.

Several models show specialization toward virus/phage detection. AGRO-NT-1B achieves per-task rank 12.5 on Virus/Phage versus 20.5 across the remaining 98 tasks (specialization $\Delta = 8.0$), potentially reflecting viral sequences embedded in plant genomic training data. GROVER ($\Delta = 6.5$) and GENA-LM ($\Delta = 6.0$) similarly specialize, both being human-pretrained Transformer-encoders. The multi-species GENA-LM-T2T-MULTI and GENA-LM-LARGE-T2T show smaller positive specialization ($\Delta = 5.3$ and $5.4$ respectively); the human-only direction remains modest within the GENA-LM family.

Few-shot degradation is severe for this category (93.5% at 1-shot, 71.3% at 10-shot), with top performers showing the steepest absolute declines: GENOMEOCEAN-4B degrades by 0.633 and GENOMEOCEAN-500M by 0.595. Despite these large absolute drops, both GENOMEOCEAN models retain their full-shot rank (top-2 at both regimes) with relative drops of 46% and 48% – well below the category mean of 71.3%. Some rank shifts emerge among the runners-up: LUCAONE rises from full-shot rank 7 to 10-shot rank 3, while GROVER drops from 6 to 21.

F.12.12. CODING VERSUS NON-CODING CLASSIFICATION

Coding/Non-coding classification ($n = 1$ task: GB Coding/Non-coding) represents a fundamental sequence annotation problem with relatively high baseline performance (mean MCC $= 0.803$ across the 40 models). This task evaluates whether models capture the statistical signatures distinguishing protein-coding from non-coding sequences. Results for this task are presented in Figures 46–48.

**Performance Degradation in Coding/Non-coding: Full-shot → 10-shot → 1-shot**

*Figure 46.* **Few-shot performance degradation on Coding/Non-coding Classification.** For each of the 40 models, MCC on the coding vs. non-coding task under full-shot, 10-shot, and 1-shot regimes; models ordered by full-shot performance. The top band shows the relative drop from full-shot to 10-shot per model. Benchmark-wide mean degradation: 26.6% for 10-shot, 71.7% for 1-shot – among the mildest 10-shot degradation observed across task categories. MUTBERT (86M, human, encoder) achieves the highest 10-shot MCC (0.748) and the highest 1-shot MCC (0.417), rising from full-shot rank 3 to 10-shot rank 1. Full-shot and 10-shot rankings remain strongly correlated (Spearman $\rho = 0.86$, top-5 overlap 3 of 5).

Coding/Non-coding classification shows significant positive scaling (Spearman $\rho = 0.480$, $p = 0.002$; $\rho = 0.571$, $p < 0.001$ excluding the prokaryotic EVO-1-131K). Models exceeding 1B parameters achieve 0.846 mean MCC compared to 0.781 for models below 200M, a $+0.065$ gap that matches the benchmark-wide tier difference.

GENERATOR-EUKARYOTE-3B leads the category (MCC $= 0.904$), followed closely by LUCAONE (MCC $= 0.901$) and MUTBERT (MCC $= 0.894$). The strong performance of MUTBERT (86M parameters, rank 3 of 40) demonstrates that even on a task that responds to scale, well-designed smaller models remain competitive with multi-billion-parameter counterparts.

Architecture comparisons favor Transformers on this category. MUTBERT (Transformer-encoder) exceeds HYENADNA-LARGE-1M (Hyena) by 0.182 MCC (0.894 vs. 0.712) under matched human-pretraining and single-nucleotide tokenization, although the contrast is not strictly matched on scale (86M vs. 55M). OMNI-DNA-1B (Transformer-decoder) outperforms ECCDNAMAMBA (Mamba-SSM) by 0.144 MCC (0.877 vs. 0.734) under matched 1B/multi-species/BPE conditions.

Pretraining data comparisons show advantages for both eukaryotic gene-focused and multi-species approaches over microbial training. GENOMICS-FM (multi-species) exceeds DNABERT-S (multi-species-microbial) by 0.127 MCC (0.845 vs. 0.718) under matched Transformer-encoder conditions ($\approx$120M each), while GENERATOR-EUKARYOTE-3B (eukaryotic-gene) outperforms DNA-GPT-3B-M (multi-species) by 0.045 MCC under matched Transformer-decoder/$k$-mer/3B conditions.

Notably, EVO-1-131K (prokaryotic pretraining) achieves acceptable performance on this task (MCC $= 0.719$, rank 36 of 40), substantially better than its catastrophic results on Splice Sites (MCC $= 0.160$) or TF Binding (MCC $= 0.173$). This relative success may reflect the more universal nature of coding-sequence signatures across prokaryotic and eukaryotic genomes, where codon usage patterns and open-reading-frame statistics share fundamental properties.

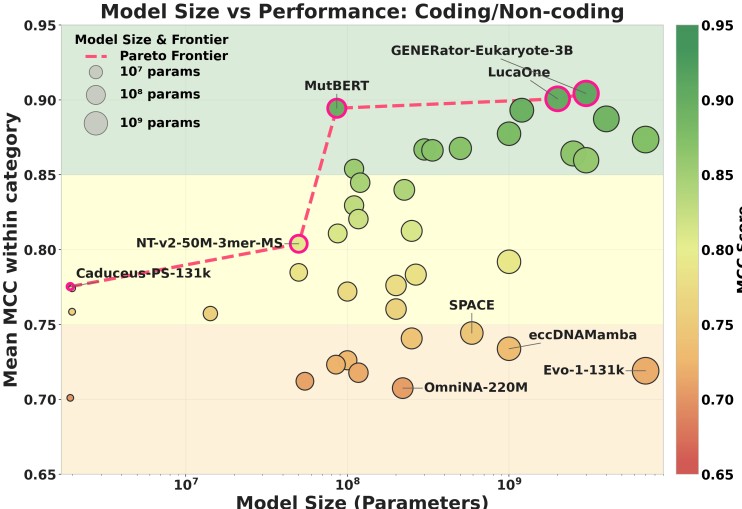

*Figure 47.* **Pareto frontier for Coding/Non-coding Classification: MCC vs. parameter count.** Each point represents one of the 40 genomic foundation models, with parameter count on a logarithmic x-axis and full-shot MCC on the y-axis. Marker size and color both encode MCC. The dashed line marks the Pareto frontier of best performance–size trade-offs. Eukaryotic gene-focused GENERATOR-EUKARYOTE-3B (3B, MCC = 0.904) leads, with multi-species LUCAONE (2B, MCC = 0.901) close behind. MUTBERT (86M, MCC = 0.894) is the strongest sub-100M model on this category and ranks 3rd of 40 overall, demonstrating that for this task small well-designed models remain highly competitive with much larger generalists.

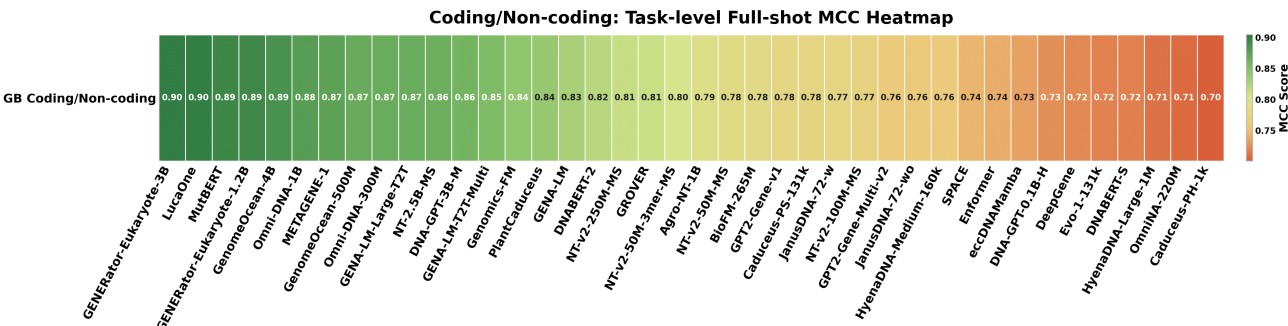

*Figure 48.* **Per-model MCC for Coding/Non-coding Classification.** Heatmap shows full-shot MCC for each of the 40 genomic foundation models on the single coding vs. non-coding task, with models sorted by MCC. Cell values report per-model MCC, with colors ranging from red/orange for lower scores to green for higher scores. Multi-species decoder models (GENERATOR-EUKARYOTE-3B, LUCAONE), Transformer-encoder models (MUTBERT), and large multi-species generalists cluster at the top of the model ordering.

### F.12.13. CHROMATIN ACCESSIBILITY

Chromatin Accessibility prediction ($n = 1$ task: iDHS DNase-I) evaluates recognition of open chromatin regions, a key determinant of transcriptional potential. This category shows distinctive patterns in both architecture and tokenization comparisons, and exhibits the weakest scaling effect of any category in our benchmark. Results for this task are presented in Figures 49–51.

Scaling effects are non-significant on this category (Spearman $\rho = 0.244$, $p = 0.128$; $\rho = 0.332$, $p = 0.039$ excluding the prokaryotic EVO-1-131K) – the weakest scaling relationship of any category in our benchmark. Models exceeding 1B parameters achieve 0.593 mean MCC versus 0.547 for models below 200M (a modest +0.046 gap, well below the benchmark-wide +0.064). GENERATOR-EUKARYOTE-3B leads the category (MCC = 0.728), followed by OMNI-DNA-1B (MCC = 0.714) and ENFORMER (MCC = 0.711).

Pretraining-data effects are pronounced. The controlled GENA-LM comparison shows multi-species exceeding human-only by 0.123 MCC (GENA-LM-T2T-MULTI = 0.583 vs. GENA-LM = 0.461) – the largest gap observed for this model pair across all 13 task categories (next-largest is Mouse Enhancers at 0.068). Eukaryotic gene-focused

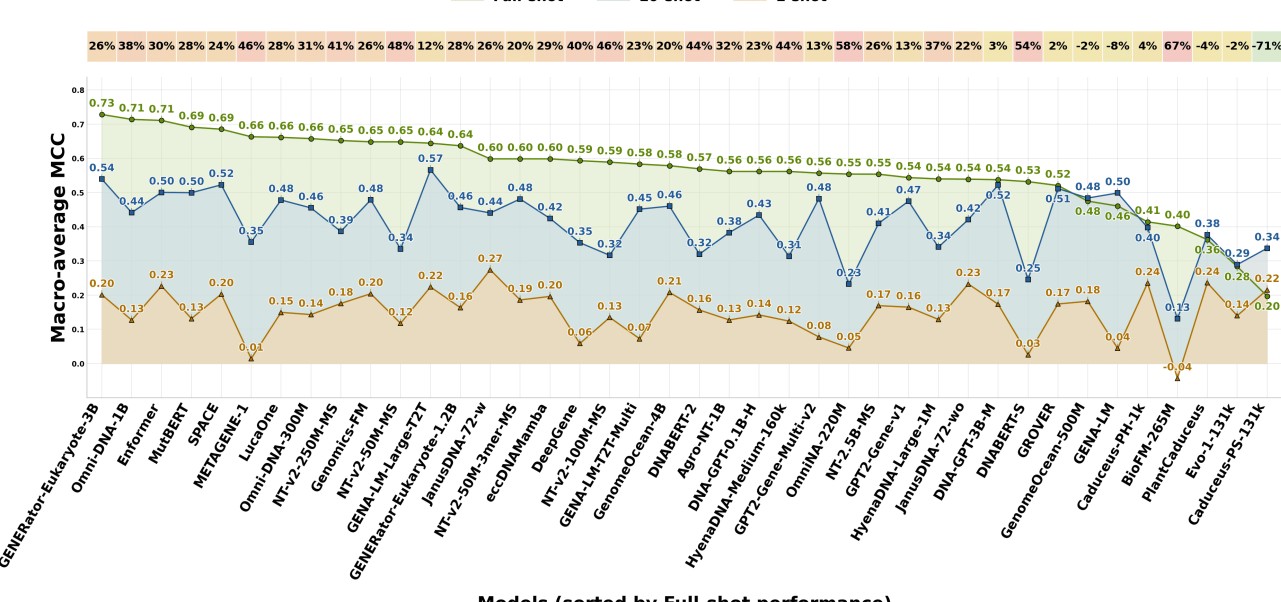

*Figure 49.* **Few-shot performance degradation on Chromatin Accessibility.** For each of the 40 models, MCC on the iDHS DNase-I task under full-shot, 10-shot, and 1-shot regimes; models ordered by full-shot performance. The top band shows the relative drop from full-shot to 10-shot per model. Benchmark-wide mean degradation: 26.8% for 10-shot, 73.6% for 1-shot – among the mildest 10-shot degradation observed across task categories. The 10-shot and 1-shot rankings reshuffle substantially relative to full-shot (Spearman $\rho = 0.38$, top-5 overlap 2 of 5): GENA-LM-LARGE-T2T rises from full-shot rank 12 to 10-shot rank 1 (MCC $= 0.567$), and JANUSDNA-72-W (1.98M parameters) achieves the highest 1-shot MCC (0.274).

GENERATOR-EUKARYOTE-3B outperforms multi-species DNA-GPT-3B-M by 0.191 MCC (0.728 vs. 0.538) under matched 3B/decoder/$k$-mer conditions – also the largest such gap for this controlled pair across the benchmark.

Tokenization comparisons reveal dramatic single-nucleotide advantages. MUTBERT (single-nucleotide) exceeds GENA-LM (BPE) by 0.231 MCC (0.691 vs. 0.461) under matched Transformer-encoder/human-pretraining conditions (86M vs. 110M) – the largest tokenization gap observed for this controlled pair across the benchmark. MUTBERT similarly outperforms GROVER (BPE) by 0.171 MCC (0.691 vs. 0.521) under matched conditions ($\approx$86–87M each). This substantial advantage suggests that chromatin accessibility prediction benefits from fine-grained positional information that coarser tokenization schemes may obscure.

Architecture comparisons show mixed patterns for this category. Under matched 1B/multi-species/BPE conditions, OMNI-DNA-1B (Transformer-decoder) exceeds ECCDNAMAMBA (Mamba-SSM) by 0.116 MCC. However, ECCDNAMAMBA (1B, MCC $= 0.599$) outperforms the smaller GENOMEOCEAN-500M (Transformer-decoder, 500M, MCC $= 0.475$) by 0.124 MCC – a size-favored outcome rather than a clean architectural reversal. Additionally, DEEPGENE (Graph-Transformer, 85M) exceeds GENA-LM (Transformer-encoder, 110M) by 0.133 MCC. These atypical patterns suggest that chromatin accessibility prediction may benefit from architectural features beyond standard attention mechanisms.

Model specialization is striking for this category. The smallest model in our benchmark, JANUSDNA-72-W (Hybrid-Mamba-MoE, 1.98M parameters), achieves per-task rank 15.0 on Chromatin Accessibility versus 32.9 across the remaining 99 tasks (specialization $\Delta = 17.9$) – this Mamba/MoE hybrid at near-toy scale outperforms most multi-billion-parameter generalists on this task. ECCDNAMAMBA (1B Mamba-SSM) shows a similar pattern ($\Delta = 16.5$). The relative strength of these two architectures on chromatin accessibility, despite their generally weaker overall performance, suggests that state-space components may capture specific sequence features relevant to chromatin state prediction; however, other Mamba-based models in our benchmark (CADUCEUS-PH-1K, PLANTCADUCEUS, CADUCEUS-PS-131K) do not show this specialization, so the effect is not a uniform property of the Mamba family.

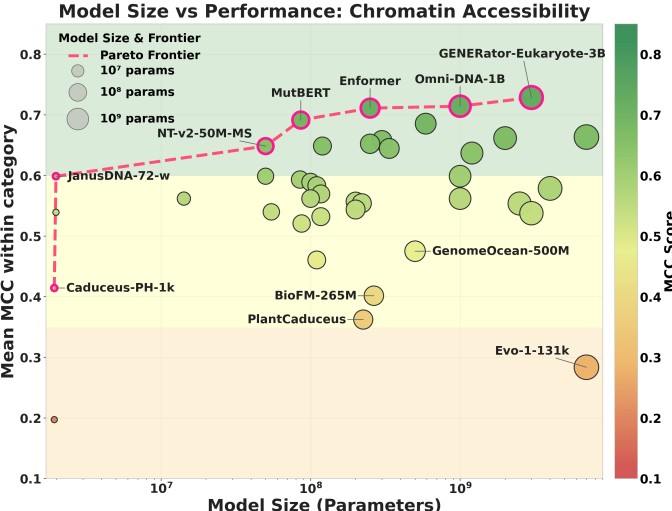

*Figure 50.* **Pareto frontier for Chromatin Accessibility: MCC vs. parameter count.** Each point represents one of the 40 genomic foundation models, with parameter count on a logarithmic x-axis and full-shot MCC on the y-axis. Marker size and color both encode MCC. The dashed line marks the Pareto frontier of best performance–size trade-offs. GENERATOR-EUKARYOTE-3B (3B, MCC = 0.728) leads, with MUTBERT (86M, MCC = 0.691, rank 4 of 40) the strongest sub-100M Transformer-encoder. Strikingly, JANUSDNA-72-W (1.98M, MCC = 0.599) sits prominently on the frontier at rank 14 of 40 – the smallest model in the benchmark outperforming many counterparts that are over 500× larger.

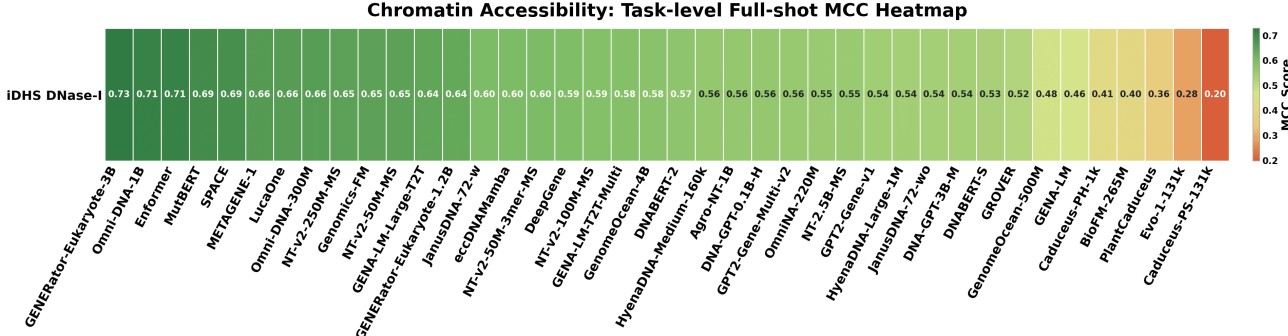

*Figure 51.* **Per-model MCC for Chromatin Accessibility.** Heatmap shows full-shot MCC for each of the 40 genomic foundation models on the single iDHS DNase-I task, with models sorted by MCC. Cell values report per-model MCC, with colors ranging from red/orange for lower scores to green for higher scores. Eukaryotic gene-focused (GENERATOR-EUKARYOTE-3B), multi-species decoders (OMNI-DNA-1B), and human-mouse epigenomic-profile models (ENFORMER, SPACE) cluster at the top of the model ordering.

## F.13. Task Difficulty and Model Differentiation

The 100 tasks in our benchmark span a wide difficulty spectrum, from near-solved problems to challenges that remain largely intractable for current foundation models. Understanding this difficulty landscape provides essential context for interpreting model comparisons and identifying promising directions for future research.

### F.13.1. EASY TASKS: APPROACHING CEILING PERFORMANCE

We identify 18 tasks achieving mean MCC exceeding 0.70 across all 40 models, indicating problems where current approaches converge to robust solutions. Human cell-type-specific promoter recognition tasks dominate this category: iPro HUVEC achieves 0.890 mean MCC, iPro HeLa-S3 reaches 0.875, and iPro GM12878 attains 0.855. The SPACE model achieves best performance on all three tasks, while BIOFM-265M consistently ranks lowest, highlighting the importance of appropriate pretraining even for ostensibly solved problems.

General promoter classification tasks (NT Promoter (all), NT Promoter (no TATA), GUE Prom 300 (no TATA)) cluster at 0.853–0.855 mean MCC, with GENA-LM-LARGE-T2T and OMNI-DNA-1B among the top performers. The GB Human-or-worm species classification task reaches 0.857 mean MCC, with MUTBERT leading (MCC = 0.948) and EVO-1-131K performing worst – a pattern reflecting the prokaryotic model's incompatibility with eukaryotic discrimination tasks.

The GB Coding/Non-coding task achieves 0.803 mean MCC, representing a relatively accessible problem where sequence composition differences between coding and non-coding regions provide strong discriminative signal. GENERATOR-EUKARYOTE-3B leads this task, while CADUCEUS-PH-1K shows weakest performance.

### F.13.2. HARD TASKS: PERSISTENT CHALLENGES

We identify 28 tasks with mean MCC below 0.35, representing problems where even state-of-the-art models achieve limited success. DNA methylation prediction dominates this category, with 4mC *G. subterraneus* achieving only 0.061 mean MCC (maximum 0.206 by GENERATOR-EUKARYOTE-3B). 4mC *E. coli* reaches 0.103 mean MCC and 4mC *G. pickeringii* achieves 0.107. These results suggest that current sequence-based approaches struggle to capture the contextual and enzymatic factors governing methylation site selection.

Plant lncRNA classification tasks prove similarly challenging, with PGB lncRNA *S. lycopersicum* (tomato) at 0.221 mean MCC, PGB lncRNA *G. max* (soybean) at 0.228, and PGB lncRNA *T. aestivum* (wheat) at 0.238. LUCAONE achieves best performance across all 6 plant lncRNA tasks, reaching 0.539 on *S. lycopersicum*, demonstrating that sufficiently diverse multi-species pretraining can partially address plant-specific challenges even without dedicated plant data.

Viral sequence classification presents unexpected difficulty, with the GUE COVID variants task achieving only 0.200 mean MCC. GENA-LM-LARGE-T2T leads at 0.472, suggesting that multi-species genomic pretraining provides some transferable signal for viral sequence analysis despite the substantial evolutionary distance.

### F.13.3. HIGH-VARIANCE TASKS: DISCRIMINATING MODEL CAPABILITIES

Tasks exhibiting high inter-model variance (standard deviation > 0.12) reveal where architectural and pretraining choices most strongly differentiate performance. We identify 13 such "controversial" tasks that serve as natural stress tests for model capabilities.

The GUE Fungi-20 task shows the highest variance (std = 0.216, range = 0.764 MCC), with GENOMEOCEAN-4B achieving 0.939 MCC while CADUCEUS-PH-1K reaches only 0.175. The top performers uniformly employ multi-species or eukaryotic-gene pretraining (GENOMEOCEAN-4B, GENERATOR-EUKARYOTE-3B, GENERATOR-EUKARYOTE-1.2B), while bottom performers use narrower data (CADUCEUS-PH-1K, BIOFM-265M, JANUSDNA-72-WO).

Splice site detection tasks exhibit consistently high variance: NT Splice donors (std = 0.195, range = 0.666), NT Splice acceptors (std = 0.186, range = 0.643), and GUE Splice reconstr. (std = 0.170, range = 0.597). NT-2.5B-MS, LUCAONE, and GENERATOR-EUKARYOTE-3B consistently rank among the top performers across splice tasks, while EVO-1-131K consistently ranks last – achieving negative MCC on GUE Splice reconstr., indicating worse-than-random performance attributable to prokaryotic pretraining's incompatibility with spliceosomal machinery.

Mouse enhancer tasks show substantial variance (std = 0.151–0.175), with ENFORMER and SPACE (human-mouse

epigenomic profiles) dominating while human-only trained JANUSDNA variants and CADUCEUS variants perform poorly. This pattern supports the importance of taxonomically aligned pretraining for cross-species regulatory element prediction.

### F.13.4. PATTERNS IN HIGH-VARIANCE TASK PERFORMANCE

Systematic analysis of top-3 and bottom-3 performers across all 13 high-variance tasks reveals striking patterns in architecture and pretraining data. Among top performers, Transformer-decoder appears 18 times and Transformer-encoder 15 times, with CNN-Transformer architectures (ENFORMER, SPACE) appearing 6 times. Among bottom performers, Mamba-SSM architectures appear 17 times, Hybrid-Mamba-MoE 7 times, and StripedHyena 6 times. This distribution suggests that attention-based architectures substantially outperform state-space alternatives on the most discriminating benchmark tasks.

Pretraining data patterns are even more pronounced. Multi-species training appears in top-3 positions 20 times and eukaryotic-gene training 12 times, while human-mouse-profiles appears 6 times. In contrast, human-only training appears in bottom-3 positions 29 times – more than all other categories combined. Prokaryotic training appears 6 times in bottom positions despite comprising only one model (EVO-1-131K), indicating consistent failure across high-variance tasks. These patterns provide strong empirical support for prioritizing taxonomically diverse pretraining over species-specific approaches.

