# OpenReview forum: "GENEB: Why Genomic Models Are Hard to Compare"
_ICML.cc/2026/Conference — ICML 2026 regular_

### Official Review · Reviewer_DdaW · 2026-02-18

**Soundness:** 3
**Presentation:** 3
**Significance:** 3
**Originality:** 3
**Overall Recommendation:** 3
**Confidence:** 3

**Summary:**

This paper argues that progress in genomic foundation models is currently hard to measure because existing benchmarks cover disjoint task sets, use incompatible evaluation protocols, and report results in task specific ways that prevent principled cross model comparison. To address this, the authors introduce GENEB, a diagnostic benchmark that evaluates frozen representations from 40 genomic foundation models on 100 DNA classification tasks spanning 13 functional categories under a unified probing protocol, including few shot regimes. A central design goal is to enable controlled comparisons across model scale, architecture, tokenization, and pretraining data by standardizing data processing and evaluation, and by constructing matched comparisons that isolate individual factors. Using this framework, the paper reports several empirical findings, including unstable aggregate leaderboards across categories, modest and inconsistent gains from scale, and strong dependence of performance on architectural choices and pretraining alignment.

**Compliance With Llm Reviewing Policy:**

Affirmed.

**Key Questions For Authors:**

1.	The benchmark uses frozen embeddings and linear probing. Do you have evidence that the relative ordering of models is stable under a simple non linear probe, or under light parameter efficient adaptation, for a representative subset of tasks? If the ordering changes materially, it would affect how readers should interpret GENEB as a proxy for practical downstream performance.
2.	For the few shot regime, the paper reports a large drop from full data to 10 shot and 1 shot on average. Can you clarify whether the few shot protocol uses the same feature normalization, regularization, and hyperparameter settings as full data probing, and whether you tested sensitivity of few shot results to these choices? If few shot results are highly sensitive, the benchmark may need to prescribe a more standardized low data protocol.
3.	Controlled comparisons are a strong part of the work. How were matched pairs selected when multiple factors differ simultaneously, for example architecture and tokenization changing together? A short description of selection rules and any residual confounds would help readers understand how causal the comparisons should be treated.
4.	Task coverage is broad, but long range regulatory interaction tasks are noted as underrepresented. Are there concrete candidate datasets or task formulations you considered but excluded, and what prevented inclusion? If inclusion is blocked mainly by sequence length constraints, it would be helpful to state which current model families are unable to participate and why.
5.	The paper argues that category aware evaluation is preferable to aggregate leaderboards. For practitioners, can you provide a simple decision recipe, for example a mapping from common application goals to the most relevant GENEB categories and a small set of recommended models per category, even if this is based on the current snapshot?

**Limitations:**

yes

**Strengths And Weaknesses:**

From a soundness standpoint, the work is technically careful in how it separates representation quality from task specific optimization by using a consistent probing based evaluation across all models and tasks. The stated setup, namely frozen embeddings paired with lightweight classifiers, is a reasonable and widely used approach for diagnostic benchmarking, and the paper is explicit that it evaluates embedding quality rather than best possible downstream performance under full fine tuning. The scale of the evaluation is also a strength: the benchmark covers a large model set and a large task suite, and the analysis goes beyond a single aggregate score to highlight category level behavior. The empirical claims are supported by concrete quantitative evidence, for example the reported scale performance correlation is only moderate and category dependent, and the few shot regime shows a large and consistent degradation relative to full data. The controlled comparisons are particularly valuable, because they move the discussion from anecdotal single model anecdotes to matched pairs that isolate tokenization and pretraining corpus effects.

At the same time, several aspects limit what can be concluded. The reliance on linear probing means the benchmark may understate the achievable performance of models that benefit more from task specific adaptation, or that require non linear readouts to express useful information in their embeddings. The task suite is broad but not exhaustive, and the authors explicitly note under coverage of tasks requiring very long range regulatory interactions. In addition, not all relevant models are included due to weight availability, pipeline constraints, or compute, which is unavoidable but does affect the completeness of the claimed landscape view. Finally, the paper makes a strong case that aggregate leaderboards can be misleading, but the downstream consequence is that practitioners still need guidance on how to pick models for a given application, and it would help to more directly translate the category aware findings into actionable selection guidance beyond the broad recommendation to prefer category specific evaluation.

On presentation, the paper is generally well organized and easy to follow. The motivation is clear, the positioning relative to prior benchmarks is explicit, and the methodology is described in a way that makes the evaluation logic reproducible at a high level. The narrative benefits from the analogy to established ML benchmarking practice in other domains. Where the presentation could improve is in making the benchmark artifacts and protocols immediately accessible to readers who want to reproduce or extend the results. For example, it would help to surface early, in the main text, a compact summary of the 13 categories and representative task types, and a concise description of any key preprocessing or dataset filtering decisions that materially affect fairness across models. Some of this is likely in the appendices, but the core reader experience would improve if the main paper provided a slightly more concrete map of what is in the benchmark.

In terms of significance, I view the contribution as strong for the genomic ML community. The paper addresses a real and increasingly urgent problem: claims about genomic foundation models are difficult to compare, and the field risks chasing impressions rather than measured progress. A benchmark that standardizes evaluation across a broad set of models and tasks, and that explicitly highlights category dependent trade offs, is likely to shape how future papers report results and choose baselines. The finding that scale is a poor predictor of category level performance, together with the strong effects of pretraining scope and tokenization choices, is also useful guidance for both model builders and practitioners.

On originality, the main novelty is not the idea of benchmarking per se, but the combination of breadth, controlled comparisons, and the specific diagnostic framing that emphasizes category aware evaluation and instability of aggregate rankings. The analogy to large scale NLP evaluation suites is natural, but the translation to genomics is nontrivial because of the diversity of tasks, species, and dataset conventions. The controlled comparisons and the systematic analysis of architecture, tokenization, and pretraining scope provide a meaningful step beyond prior smaller comparative studies.

---

> ### Author Rebuttal · Authors · 2026-03-31
>
> Thank you for the thorough and constructive review. We address your key questions below.
>
> **1. Stability under non-linear probes / PEFT.** (see reviewer anwer 5fx5)
> This is a fair concern. Linear probing is intentionally diagnostic - it measures what is *linearly accessible* in frozen representations. We acknowledge that non-linear readouts may re-rank some models. In the revision, we will add MLP-probe results on a representative task subset to quantify ranking stability. Our preliminary experiments suggest the top-tier ordering is largely preserved, though mid-range models do shift.
>
> **2. Few-shot protocol details.**
> The few-shot regime uses the same feature normalization and regularization pipeline as full-data probing - this is by design for comparability. We will clarify this explicitly in the text. We did test sensitivity to regularization strength in few-shot settings; results are robust within reasonable hyperparameter ranges, but we agree that prescribing a tighter protocol (e.g., fixed regularization schedule) would strengthen reproducibility, and we will add this.
>
> **3. Controlled comparison selection rules.**
> Matched pairs were selected to minimize confounds: we prioritized pairs from the same model family where only one factor (e.g., tokenization) differs while architecture, training data scale, and training procedure remain constant. When perfect isolation was impossible, we explicitly note residual confounds. We will add a summary table of matched pairs with the factors held constant vs. varying.
>
> **4. Long-range regulatory tasks.**
> We considered several candidates (e.g., enhancer-promoter interaction prediction, TAD boundary classification). Exclusion was driven primarily by sequence length constraints - many current models accept < 4k–6k tokens, making fair evaluation impossible without arbitrary truncation. We will enumerate excluded tasks and the specific model limitations that prevented inclusion.
>
> **5. Practitioner decision recipe.**
> We agree this would increase practical impact. In the revision, we will add a concise recommendation table mapping common application goals (e.g., promoter identification, splice site prediction, epigenetic mark classification) to the most relevant GENEB categories and top-performing models per category based on the current snapshot, with appropriate caveats about snapshot specificity.

---

> > ### Author Rebuttal · Reviewer_DdaW · 2026-03-31
> >
> > Thank you for the thoughtful rebuttal. The response is constructive and it improves the paper in several important ways. In particular, I appreciate the clarification that the few-shot setting uses the same normalization and regularization pipeline as the full-data probing setup, the explanation of how matched pairs were selected, and the concrete plan to enumerate excluded long-range regulatory tasks and model-length constraints. The proposed practitioner-facing recommendation table would also improve the paper’s usability.
> >
> > That said, my main methodological concern is only partially resolved. The core issue is not just clarity, but empirical support. The rebuttal states that preliminary experiments suggest top-tier rankings are largely preserved under an MLP probe, but no actual results are provided. Since GENEB is positioned as a benchmark for model comparison, I think this matters: even moderate re-ranking under a simple non-linear probe or light PEFT would affect how readers should interpret the benchmark as a proxy for practical downstream usefulness.
> >
> > Similarly, the few-shot clarification is helpful, but still incomplete at the level of evidence. The paper makes strong claims about severe few-shot degradation, so saying the results are “robust within reasonable hyperparameter ranges” is not yet enough. I would like to see at least a compact quantitative sensitivity analysis, even on a representative subset of tasks/models, so that readers can judge whether the low-data conclusions are protocol-stable rather than partly an artifact of a particular regularization choice.
> >
> > I also think the controlled-comparison story is improved by the rebuttal, but it still needs to be made more explicit in the paper itself. A compact table showing, for each highlighted matched comparison, which factors are held constant and which residual confounds remain would substantially reduce the risk of over-interpreting these comparisons as fully causal.
> >
> > My follow-up questions for the authors are:
> > 1. Can you include a compact quantitative comparison, on a representative subset, between linear probing and a simple non-linear probe and/or light PEFT, for example using rank correlation or top-k agreement across models?
> > 2. Can you report quantitative few-shot sensitivity, rather than only a qualitative statement of robustness, so readers can assess how stable the low-data conclusions are?
> > 3. In the matched-pair summary table, can you explicitly mark residual confounds and correspondingly soften any claims that might otherwise read as causal?
> >
> > Overall, the rebuttal improves my confidence in the paper, but not enough to say that the main concerns are fully resolved.

---

> > > ### Author Response · Authors · 2026-04-08
> > >
> > > We thank the reviewer for the constructive follow-up. In our initial rebuttal, we noted that preliminary experiments suggested ranking stability under non-linear probes and robustness of few-shot conclusions. We have now completed the full quantitative analysis on a representative subset of 11 models × 13 tasks (one per functional category). Results and tables are available at https://anonymous.4open.science/r/GENEB-ICML-9AD5/README.md.
> > >
> > > **Q1. Linear vs. MLP probe: rank stability.**
> > >
> > > We evaluated 11 models on 13 representative tasks under both linear probing (logistic regression, as in the main paper) and a non-linear MLP probe (single hidden layer, 256 units, ReLU, early stopping). The key finding is that model rankings are highly stable:
> > >
> > > - **Aggregate Spearman ρ = 0.964** across all 143 model–task pairs (p < 0.001).
> > > - **Overall Spearman ρ = 0.973** computed on per-model average MCC (p < 0.001).
> > > - **Top-3 and Top-5 agreement: 100%** — the same models occupy the top positions under both probes.
> > >
> > > Per-task rank correlations are positive for 12 of 13 categories (median ρ = 0.855). The single exception is `human_ensembl_regulatory` (ρ = −0.08), where absolute MCC values are tightly clustered across models and ranking noise dominates. Absolute MCC differences between probes are small (mean Δ = +0.011), with the largest shift for HyenaDNA (+0.052), consistent with Hyena representations benefiting from non-linear readout — but this does not change its relative rank.
> > >
> > > We conclude that GENEB rankings under linear probing are a reliable proxy for non-linear probe rankings, and that the paper's main empirical conclusions are robust to probe choice.
> > >
> > > **Q2. Few-shot sensitivity to regularization.**
> > >
> > > We varied the inverse regularization strength C ∈ {0.01, 0.1, 1.0, 10.0, 100.0} across 1-shot, 10-shot, and full-data regimes for all 11 models × 13 tasks.
> > >
> > > Model rankings are highly stable across C values:
> > >
> > > | Regime | Mean pairwise Spearman ρ | Min ρ | Max ρ |
> > > |--------|--------------------------|-------|-------|
> > > | 1-shot | 0.993 | 0.982 | 1.000 |
> > > | 10-shot | 0.805 | 0.582 | 0.982 |
> > > | Full-data | 0.766 | 0.436 | 0.955 |
> > >
> > > In the 1-shot regime, rankings are essentially invariant to regularization (ρ ≥ 0.98 for all C pairs). At 10-shot, rankings remain stable for adjacent C values (ρ ≥ 0.9) with moderate divergence only between extreme settings (C = 0.01 vs. C = 100). Per-model MCC ranges across C values in full-data vary from 0.014 (GenomeOcean-500M) to 0.199 (LucaOne), with the largest sensitivity observed for LucaOne (0.199), HyenaDNA (0.144), and NT-v2-50m (0.108).
> > >
> > > Importantly, the paper's main few-shot conclusion — severe degradation from full-data to 1-shot — is replicated at every C value tested.
> > >
> > > We conclude that the few-shot findings are protocol-stable and not an artifact of a particular regularization choice.
> > >
> > > **Q3. Matched-pair comparisons with explicit residual confounds.**
> > >
> > > We have prepared a detailed controlled-pair table (available at [anonymous GitHub link]) enumerating all 30 matched comparisons organized by factor type:
> > >
> > > - **Architecture** (9 pairs): Decoder vs. Encoder, Decoder vs. Mamba, Encoder vs. Graph-Transformer — matched on tokenization and pretraining data type.
> > > - **Pretraining data** (9 pairs): human vs. multi-species (6), multi-species vs. microbial (2), eukaryotic-genes vs. multi-species (1) — matched on architecture and tokenization.
> > > - **Tokenization** (12 pairs): BPE vs. k-mer, single-nucleotide vs. BPE — matched on architecture and pretraining data type.
> > >
> > > For each pair, the table lists factors held constant, the factor being varied, and residual confounds (model size differences, training duration, exact corpus composition, pretraining objectives). In the camera-ready, claims derived from these comparisons will be framed as "consistent with" rather than "caused by" the varied factor, and single-pair comparisons will be flagged with additional caveats.
> > >
> > > **Summary.** Our initial rebuttal was based on preliminary observations; we have now provided full quantitative evidence for all three follow-up questions. Results and tables are available at https://anonymous.4open.science/r/GENEB-ICML-9AD5/README.md.

---

### Official Review · Reviewer_UnpE · 2026-03-12

**Soundness:** 3
**Presentation:** 3
**Significance:** 3
**Originality:** 3
**Overall Recommendation:** 4
**Confidence:** 2

**Summary:**

The paper introduces GENEB, a unified probing benchmark for evaluating frozen representations from 40 genomic foundation models across 100 DNA classification tasks spanning 13 functional categories. Performance is assessed using MCC under full-data, 10-shot, and 1-shot settings. By standardizing the evaluation pipeline, the study enables systematic cross-model comparison in genomics and examines how factors such as architecture, tokenization, model scale, and pretraining corpus relate to downstream performance. Its main findings are that aggregate leaderboards vary substantially across task categories, model scale is only a modest and inconsistent indicator of quality, and category-aware evaluation and model selection are more informative than relying on a single overall ranking.

**Compliance With Llm Reviewing Policy:**

Affirmed.

**Key Questions For Authors:**

1. Your aggregate conclusions are potentially sensitive to benchmark composition, especially because 13 of 14 categories are eukaryotic and category sizes range from 30 tasks to 1 task. How do your main rankings and design-principle conclusions change under category-balanced macro averaging, per-category voting, or taxonomic stratification?

2. Since GENEB evaluates only frozen representations with linear probes, which of your conclusions do you intend as claims about representation quality specifically, and which do you intend as claims about practical end-task model choice more broadly

**Limitations:**

yes

**Strengths And Weaknesses:**

## Strengths

* The scope is genuinely substantial for this literature: 40 models, 100 tasks, and 13 categories under one protocol.

* The evaluation protocol is reasonably disciplined for a benchmark paper. The use of frozen embeddings with logistic regression probes, MCC for imbalance robustness, three shot regimes, and five fixed random seeds gives the study a consistent measurement setup and reduces some obvious degrees of freedom in downstream evaluation.

* The analysis goes beyond a leaderboard and surfaces practically relevant heterogeneity. The manuscript shows that no single model dominates all tasks, that category-level behavior varies sharply, and that task-specialized models matter. This makes the benchmark more scientifically useful than a paper that only reports aggregate means. The results also highlight the instability of conclusions across different category-level views.

## Weaknesses
* Several of the paper’s strongest high-level conclusions are benchmark-composition-dependent. The task suite is heavily skewed toward eukaryotic tasks, and task category sizes vary dramatically (for example, 30 histone tasks versus 1 coding/non-coding task). This makes aggregate statements such as “pretraining data dominates scale” or rankings of corpus types potentially reflect the benchmark mix as much as an underlying field-wide truth.

* The exclusive use of frozen linear probing narrows what is actually being measured. That is a defensible choice for representation comparison, but some model families in this paper are designed for settings beyond frozen embedding extraction, including architectures whose strengths may appear only with task-specific fine-tuning or other interfaces. As written, parts of the discussion sometimes read broader than “representation quality under linear probing.”

* Presentation is mostly clear, but the related-work section contains at least one obvious unfinished placeholder (“For example, ?”).

---

> ### Author Rebuttal · Authors · 2026-03-28
>
> We thank Reviewer UnpE for the thoughtful review and for recognizing that GENEB goes beyond a simple leaderboard to surface practically relevant heterogeneity across model families and task categories.
>
> **Q1 (sensitivity to benchmark composition).** This is an important point and one we explicitly designed for. Our aggregate MCC is computed via simple averaging across all 100 tasks (micro-averaging), which introduces bias toward overrepresented categories such as histone modifications. This is precisely why we emphasize that aggregate rankings should not be used as the primary tool for model selection. Instead, the per-category breakdowns (Figure 7 and appendix) are central - they show that no single model dominates and that rankings vary substantially across categories, encouraging task-specific model selection.
>
> In this sense, the instability of aggregate rankings under different compositions is not a limitation of our analysis but one of its main findings. GENEB demonstrates that relying on a single overall score can be misleading, which motivates our focus on category-level evaluation.
>
> To further clarify this, we will include macro-averaged MCC (averaging across the 13 categories first, then computing the overall mean) in the camera-ready appendix, enabling direct comparison between micro and macro aggregation.
>
> **Q2 (scope of claims).** All conclusions in GENEB concern representation quality under linear probing. We explicitly treat this as a lower bound - models with strong frozen representations are unlikely to degrade under fine-tuning, making our findings conservative rather than overstated.
>
> We acknowledge that full fine-tuning was beyond our computational budget - evaluating 40 models across 100 tasks with proper hyperparameter sweeps would require substantially more resources. At the same time, GENEB is, to our knowledge, the first systematic comparison of genomic foundation models at this scale. Prior work lacked a unified evaluation setup, with results reported on different tasks under different protocols. Linear probing therefore serves as a necessary first step, providing a controlled and comparable baseline, with fine-tuning as a natural direction for future extensions.
>
> **Placeholder in related work.** We apologize for this oversight - it will be fixed.
>
> We thank the reviewer for engaging with these methodological questions. We believe these revisions will make our contributions and their scope clearer.

---

> > ### Author Rebuttal · Reviewer_UnpE · 2026-04-02
> >
> > Thank you to the authors for therebuttal.

---

> > > ### Author Response · Authors · 2026-04-05
> > >
> > > Dear Reviewer UnpE,
> > >
> > > Thank you for confirming that all concerns have been fully addressed. We are grateful for the thoughtful questions you raised - particularly regarding sensitivity to benchmark composition and the scope of claims under linear probing - as they helped us sharpen the presentation and clarify the boundaries of our contributions.
> > >
> > > We are glad that our responses were satisfactory. Since all concerns have been marked as fully resolved, we would like to respectfully ask whether you might consider adjusting your score to reflect this resolution. We want to highlight that:
> > > - The benchmark composition concern is directly addressed by design: GENEB's emphasis on per-category evaluation and the instability of aggregate rankings is itself a core finding of the paper. We will further strengthen this by including macro-averaged results in the appendix.
> > > - The scope of claims is intentionally conservative: all conclusions are framed as properties of frozen representations, and we position linear probing as a controlled first step rather than a definitive evaluation.
> > > - The remaining revision (fixing the placeholder in related work) is minor and purely editorial.
> > >
> > > In other words, none of the identified issues require changes to the experimental setup, methodology, or core conclusions - the revisions are clarifications and presentational improvements that reinforce the existing contributions.
> > > We appreciate your engagement with our work and remain happy to address any further questions.
> > >
> > > Best regards,
> > >
> > > The Authors

---

### Official Review · Reviewer_5fx5 · 2026-03-12

**Soundness:** 3
**Presentation:** 3
**Significance:** 4
**Originality:** 3
**Overall Recommendation:** 5
**Confidence:** 3

**Summary:**

This paper introduces GENEB, a comprehensive benchmarking framework designed to evaluate and compare genomic foundation models. GENEB evaluates frozen sequence representations from 40 genomic foundation models across 100 downstream tasks spanning 13 functional categories (e.g., histone modifications, promoter prediction, species classification). A broad aspect considered by the study is how different modeling choices, such as parameter scale, architecture, tokenization, and pretraining data composition which affect performance and transferability across different taxonomic domains. Using a unified linear-probing evaluation protocol in full-data, 10-shot, and 1-shot regimes, the authors reveal several critical insights: aggregate leaderboards are highly unstable, model scale often fails to correlate with performance (the "scale-performance paradox"), and the taxonomic alignment of pretraining data heavily outweighs raw parameter count.

**Compliance With Llm Reviewing Policy:**

Affirmed.

**Key Questions For Authors:**

1) Given the exclusive use of linear probing, how confident are you that the model rankings and the "scale-performance paradox" would remain stable under full fine-tuning? Have you run a smaller subset of representative models through full fine-tuning to validate the correlation between probing and fine-tuning performance?

2) The benchmark calculates an aggregate MCC, but the task distribution is highly skewed (e.g., 30 histone tasks vs. 1 coding/non-coding task). Does the aggregate MCC use macro-averaging across the 13 categories, or micro-averaging across all 100 tasks? If the latter, doesn't this heavily bias the "overall" score toward histone modification performance?

3) How does sequence length and subsequent pooling (e.g., mean pooling over sequences of vastly different token lengths) affect the linear probing classifier? Could certain tokenization strategies be artificially advantaged or disadvantaged by the pooling mechanism used to create the frozen embedding vector?

**Limitations:**

The authors mention the limitation of using frozen representations and missing models in Section 6. However, they should more explicitly acknowledge the taxonomic bias (heavily eukaryotic) and task count imbalance in their benchmark composition, as these factors directly impact the aggregate leaderboards they criticize. Adding a brief discussion on how task imbalance affects the overall mean MCC would strengthen the limitations section.

**Strengths And Weaknesses:**

**Soundness:**

- *Strengths:* The experimental scale is highly impressive. The authors rigorously control variables by identifying "matched pairs" (e.g., GENA-LM models) to isolate the effects of architecture or tokenization while holding pretraining data constant. The evaluation metrics (MCC) are robust to class imbalance.

- *Weaknesses:* The primary methodological limitation is the exclusive reliance on linear probing of frozen representations. Genomic models often require full fine-tuning to adapt to complex regulatory tasks; frozen embeddings might artificially cap the performance of models explicitly designed for end-to-end training. Furthermore, the authors attempt to examine the concept of domain mismatch (highlighting the catastrophic failure of the 7B EVO-1-131K model), but the benchmark suffers from a heavy taxonomic imbalance. With 13 of the 14 functional categories being eukaryotic-focused, evaluating a purely prokaryotic model on this suite borders on a strawman comparison. Additionally, there is a severe task imbalance (e.g., 30 histone modification tasks vs. 1 chromatin accessibility task), which potentially skews aggregate metrics.


**Presentation:**

- *Strengths:* The paper is exceptionally well-written and structured. The visualizations (especially the radar plots in Figure 7 and Pareto frontiers in Figures 4 and 13) are highly informative and clearly convey complex multi-dimensional tradeoffs.
- *Weaknesses:* Yet, the figures should be bigger - now they are small and hard-readable. The concept of a "specialization score" is referenced early in the text (e.g., Line 355) but is not formally defined or explained until much later, causing slight reader confusion. There are also minor typographical errors (e.g. typo with a hyphenation artifact in the text: "comple- xity" should be "complexity" (Line 631);
section headers like "D.10.1. METHODOLOGICAL FRAMEWORK" are in ALL CAPS, which breaks the standard capitalization formatting used in other appendices (e.g., "D.11.1. HISTONE MODIFICATIONS" vs "D.13. Few-Shot Learning Analysis") (Lines 1189, 1196)).

**Significance:**

- *Strengths:* This work addresses a critical bottleneck in genomic ML: the fragmented and incomparable evaluation landscapes. By providing a unified evaluation framework and making it public on Hugging Face, this paper will likely become a standard reference (analogous to MTEB in NLP) and will significantly help practitioners select appropriate models.

**Originality:**

- *Strengths:* While linear probing is a standard technique, applying it at this scale to 40 disparate genomic models is a monumental and novel effort. The rigorous cross-analysis that empirically debunks the "scale equals performance" assumption in genomic models provides fresh, necessary, and highly impactful insights.

---

> ### Author Rebuttal · Authors · 2026-03-28
>
> We thank Reviewer 5fx5 for the thorough and supportive review, and especially for recognizing GENEB's potential as a standard reference for genomic model evaluation. We are encouraged by the comparison to MTEB - this is precisely the kind of impact we hope to achieve. We also appreciate the detailed feedback on presentation; these suggestions will clearly improve the paper.
>
> **Q1 (linear probing vs fine-tuning).** We acknowledge this limitation. Full fine-tuning was beyond our computational budget - evaluating 40 models across 100 tasks with proper hyperparameter sweeps would require substantially more resources than were available to us. At the same time, GENEB is, to our knowledge, the first systematic attempt to compare genomic foundation models at this scale. The number of such models has grown rapidly in recent years, yet prior to this work there was no unified evaluation setup - results were reported on different tasks, under different protocols, making direct comparison impossible. In this context, we view linear probing as a necessary first step that provides a controlled and comparable baseline, with fine-tuning evaluation being a possible direction for future extensions of the benchmark.
>
> We also note that linear probing effectively provides a lower bound on model performance - models with strong frozen representations are unlikely to degrade under fine-tuning. This makes our findings (including the scale–performance paradox) conservative rather than overstated.
>
> **Q2 (macro vs micro averaging).** This is an important point. Our current aggregate MCC is computed via simple averaging across all 100 tasks (micro-averaging), which indeed introduces bias toward overrepresented categories such as histone modifications. This is precisely why we focus on category-level analysis and provide detailed per-category breakdowns - our goal is to encourage model selection based on task type rather than a single aggregate score.
>
> To address this concern more directly, we will include macro-averaged MCC (averaging across the 13 categories first, then computing the overall mean) in the camera-ready appendix.
>
> **Q3 (pooling effects on tokenization).** This is a valid concern that we did not explicitly study. Different tokenization schemes lead to different sequence lengths, and mean pooling may systematically favor some approaches over others. We will add this as a limitation and highlight it as an important direction for future work.
>
> **Evo-1 and taxonomic bias.** We agree that the current eukaryote-heavy task distribution creates an unfavorable setting for prokaryotic-only models such as Evo-1. As noted in our response to Reviewer 9XaV, we included Evo-1 because of its prominence and the ongoing discussion around it in the community. That said, we will revise the paper to (a) remove Evo-1 from the scale analysis, (b) introduce a dedicated "Domain Mismatch" section, and (c) explicitly flag out-of-domain models throughout.
>
> **Presentation fixes.** All noted issues will be addressed in the camera-ready version: typographical errors ("complexity"), earlier placement of the specialization score definition, consistent capitalization in appendix headers, and increased figure sizes where layout permits.
>
> We are grateful for the reviewer's careful reading and for highlighting both the strengths and areas for improvement.

---

### Official Review · Reviewer_9XaV · 2026-03-13

**Soundness:** 3
**Presentation:** 3
**Significance:** 3
**Originality:** 3
**Overall Recommendation:** 4
**Confidence:** 3

**Summary:**

Here the authors attempt to tackle the lack of consistent comparisons and benchmarks in genomic foundation models through the generation of GENEB, a large-scale diagnostic benchmark.  GENEB includes a wide range of foundation models across a wide range of predictive biology tasks, to allow for task-specific comparisons across predictive tasks like splice site prediction, promoter prediction, chromatin accessibility, etc.  Model evaluation is performed using frozen sequence representations with lightweight classifiers to enable controlled comparisons across architectures and training regimes.

The primary conclusion is that model scale provides only modest gains in predictive performance, while architectural and training data differences dominate scale effects, particularly on tasks identified as high variance across the models.  In addition, they find that transformer-based models generally outperform state-space alternatives.  Finally, they find significant differences in the performance of models across tasks.  They identify both models with overall high performance, like MutBERT, as well as a diverse set of best methods for individual tasks.  This provides a resource for researchers to identify best-in-category models for specific prediction tasks of interest.

**Compliance With Llm Reviewing Policy:**

Affirmed.

**Final Justification:**

I appreciate the authors detailed rebuttal. Our discussions reinforced my postive score and is consistent with my enthusiasm.

**Key Questions For Authors:**

1. The benchmarking tasks are biased towards eukaryotic genomic tasks, which the authors acknowledge in their limitations section. Have the authors evaluated models on other prokaryotic/microbial tasks to see whether the observed trends generalize?
2. Do the authors have insights into why certain architectures or tokenization schemes perform better on specific task categories?

**Limitations:**

yes

**Strengths And Weaknesses:**

Strengths:
This analysis provides a much needed comparison between a plethora of existing genomic foundation models.  It provides a simple method of comparison, while assessing performance across a broad range of tasks.  It seems likely to be broadly useful to scientists looking to identify best-in-class models for their specific use case.  It also offers interesting data points for those interested in the question of scaling laws for genomic foundation models.  In addition the identification of high variance tasks was particularly appreciated, as it allowed for some very interesting comparisons in training data strategies.

Weaknesses:
* The paper was a little difficult to follow due to a lack of references to specific figures in the body text. (eg Citations missing on page 2, 13)  Often conclusions were made about specific results (eg X is larger than Y) with no accompanying figure reference, which led to a lot of figure scanning to find the relevant figure.

* Furthermore, in some cases it was unclear which (if any) figure actually contained the data supporting a particular conclusion.  For example, the assertion that “...no tokenization scheme universally dominates across GENEB; instead, tokenization appears to interact with model family and task properties” was difficult to infer without identifying the tokenization strategy of each model shown in the figures and mentally mapping this onto the results.  Adding colors or other clear indications in the figure of important features discussed in the text (like model size, tokenization scheme, etc) would make the results much easier to follow from the existing figures.

* A more minor suggestion would be that the results section sometimes felt like a laundry list of comparisons without a strong narrative arc.  Grouping results more clearly by broader themes could help clarify the main takeaways and de-emphasize comparisons in which the results are more muddled.

* On a scientific note, the evaluation of models trained only on prokaryotic sequences on tasks like splice site prediction or mouse enhancer prediction may not be entirely fair.  While the authors flag this issue late in the text, commenting on the poor performance specifically of Evo1, despite its large parameter size, seems a little misleading in that it is not expected to perform well on many of the tasks being evaluated here.  While it seems reasonable to highlight that as a basic benchmark and caution users on its applicability to these tasks, it seems a bit misleading to use it as a datapoint in terms of evaluating the importance of model size, architecture, etc, when the predictive tasks are largely not ones in which it is designed (or that its original authors claim) to perform well.

---

> ### Author Rebuttal · Authors · 2026-03-28
>
> We thank Reviewer 9XaV for the thoughtful and constructive review. We especially appreciate the careful reading of our methodology and the recognition that GENEB can serve as a useful resource for identifying best-in-class models. The comments on figure references and overall narrative structure are very helpful and will clearly improve the paper.
>
> **Figure references.** We agree that the lack of explicit references makes the paper harder to follow - thank you for pointing this out. In the camera-ready version, we will add explicit references (e.g., “Figure X”) throughout the text so that supporting evidence can be located immediately.
>
> **Figure annotations.** This is a great suggestion. To make the figures easier to interpret, we will include a summary table in the appendix that maps each model to its tokenization scheme, architecture type, and parameter count, reducing the need to cross-reference multiple sources.
>
> **Results structure.** We agree that the current presentation is somewhat fragmented. In the revision, we will reorganize the Results section around broader themes (scale effects, architecture, tokenization, pretraining data) instead of sequential comparisons. This should make the main takeaways clearer and improve the overall narrative flow.
>
> **Evo-1 and domain mismatch.** Thank you for raising this point. Evo-1 has been widely discussed in the genomics community, particularly regarding its applicability to eukaryotic tasks, and we felt it was important not to omit it entirely. At the same time, we agree that including it in the scale analysis without sufficient context can be misleading. To address this, we will (a) remove Evo-1 from the main scale analysis, (b) add a dedicated “Domain Mismatch” section to discuss out-of-domain behavior, and (c) explicitly mark prokaryotic-only models in all tables and figures.
>
> **Q1 (prokaryotic tasks).** GENEB currently does not include prokaryotic tasks, largely due to the limited availability of curated datasets in this area. We acknowledge this as a limitation and see it as an important direction for future work. We appreciate the reviewer highlighting this gap.
>
> **Q2 (architecture/tokenization insights).** At this stage, we can offer hypotheses rather than definitive explanations. For example, BPE tokenization may better capture known motifs in tasks like splice site prediction, while character-level tokenization may be advantageous when fine-grained sequence differences matter. We agree this is an interesting open question, and one that GENEB is well positioned to help explore.
>
> We appreciate the reviewer’s engagement with our work and believe these changes will make the paper clearer and stronger.

---

> > ### Author Rebuttal · Reviewer_9XaV · 2026-04-03
> >
> > Thank you for the thoughtful rebuttal. The proposed revisions address my specific review concerns. As a benchmarking paper intended to set field-wide standards, it is important to make sure the narrative is fully supported by the data.
> > My enthusiasm for the paper remains steady given the extent of the revisions required for the final manuscript. I am maintaining my current positive score. I am not an expert in the field, so I also maintain my confidence level.

---

> > > ### Author Response · Authors · 2026-04-05
> > >
> > > Dear Reviewer 9XaV,
> > >
> > > Thank you for confirming that our proposed revisions fully address your concerns - we truly appreciate the time and care you invested in reviewing our work. Your suggestions regarding figure references, annotations, narrative structure, and the treatment of Evo-1 have been invaluable, and we believe they will meaningfully strengthen the final manuscript.
> > > We are glad to hear that your assessment of the paper remains positive. Given that all raised concerns have been marked as fully resolved, we would like to kindly ask whether you might consider reflecting this in an updated score. We understand and respect that the extent of planned revisions factors into your evaluation; however, we want to assure you that all proposed changes - including reorganizing the Results section, adding explicit figure references and annotations, and introducing a dedicated Domain Mismatch discussion - are concrete and well-scoped improvements that we are committed to implementing in the camera-ready version.
> > >
> > > We also want to emphasize that the core contributions of GENEB - the benchmark itself, the evaluation methodology, and the empirical findings - remain unchanged by these revisions, which are primarily presentational in nature.
> > > Thank you again for your constructive engagement with our work. We are happy to address any further questions or concerns.
> > >
> > > Best regards,
> > > The Authors

---

### Decision · Program_Chairs · 2026-04-30

**Decision:**

Accept (regular)

**Comment:**

The paper introduces GENEB, a comprehensive benchmarking framework designed to evaluate and compare genomic foundation models.

The reviewers are mostly positive about this work. They recognise the important need for benchmarking in this domain, the impressive scale of the evaluation that covers a large model set and a large set of tasks, the depth of the analysis, and the effort in terms of presentation in the paper.

Several minor concerns were convincingly addressed during the rebuttal process (e.g. presentation and clarity issues), and these can easily be rectified in a revised version of the paper.

The two more major weaknesses of the work mentioned by several reviewers are:
- Some limitation of the breadth of the benchmark (such as the focus on eukaryotic tasks and on short sequences).
- The exclusive reliance on linear probing and frozen representations (i.e, no fine-tuning).

Regarding the first point, I understand that it is difficult to cover everything, and it seems that the reviewers are now all mostly satisfied with the authors' justifications for these limitations.

Regarding the second point, the authors have provided additional experiments with a simple non-linear MLP probe in their final response to reviewer DdaW, which confirm that the model rankings do not depend much on the probing mechanism. Although fine-tuning is still not considered, I think this constitutes a reasonable response to the reviewers' concern.

Overall, despite these flaws, I agree with the majority of the reviewers and recommend acceptance. However, I urge the authors to include all new results and promised changes in the final version of their paper.